# A high-resolution transcriptomic and spatial atlas of cell types in the whole mouse brain

The mammalian brain consists of millions to billions of cells that are organized into many cell types with specific spatial distribution patterns and structural and functional properties[1–3]. Here we report a comprehensive and high-resolution transcriptomic and spatial cell-type atlas for the whole adult mouse brain. The cell-type atlas was created by combining a single-cell RNA-sequencing (scRNA-seq) dataset of around 7 million cells profiled (approximately 4.0 million cells passing quality control), and a spatial transcriptomic dataset of approximately 4.3 million cells using multiplexed error-robust fluorescence in situ hybridization (MERFISH). The atlas is hierarchically organized into 4 nested levels of classification: 34 classes, 338 subclasses, 1,201 supertypes and 5,322 clusters. We present an online platform, Allen Brain Cell Atlas, to visualize the mouse whole-brain cell-type atlas along with the single-cell RNA-sequencing and MERFISH datasets. We systematically analysed the neuronal and non-neuronal cell types across the brain and identified a high degree of correspondence between transcriptomic identity and spatial specificity for each cell type. The results reveal unique features of cell-type organization in different brain regions—in particular, a dichotomy between the dorsal and ventral parts of the brain. The dorsal part contains relatively fewer yet highly divergent neuronal types, whereas the ventral part contains more numerous neuronal types that are more closely related to each other. Our study also uncovered extraordinary diversity and heterogeneity in neurotransmitter and neuropeptide expression and co-expression patterns in different cell types. Finally, we found that transcription factors are major determinants of cell-type classification and identified a combinatorial transcription factor code that defines cell types across all parts of the brain. The whole mouse brain transcriptomic and spatial cell-type atlas establishes a benchmark reference atlas and a foundational resource for integrative investigations of cellular and circuit function, development and evolution of the mammalian brain.

The mammalian brain is extraordinarily complex, and controls a wide variety of the organism's activities including vitality, homeostasis, sleep, consciousness, sensation, innate behaviour, goal-directed behaviour, emotion, learning, memory, reasoning and cognition. These activities are governed by highly specialized yet intricately integrated neural circuits composed of many cell types with diverse molecular, anatomical and physiological properties. To understand how the variety of brain functions emerge from this complex system, it is essential to gain comprehensive knowledge about the cell types and circuits that constitute the molecular and anatomical architecture of the brain.

The anatomical architecture of the mammalian brain has been defined by its developmental plan and cross-species evolutionary ontology[4–6]. The entire brain is composed of telencephalon, diencephalon, mesencephalon (also known as midbrain) and rhombencephalon (also known as hindbrain). Telencephalon consists of five major brain structures: isocortex, hippocampal formation (HPF), olfactory areas (OLF), cortical subplate (CTXsp) and cerebral nuclei (CNU). The first four brain structures—isocortex, HPF, OLF and CTXsp—constitute the developmentally derived pallium structure and are also collectively called cerebral cortex, whereas CNU derives from subpallium and is further divided into striatum (STR) and pallidum (PAL). Diencephalon consists of thalamus (TH) and hypothalamus (HY). Together telencephalon and diencephalon are also collectively referred to as forebrain. Hindbrain is divided into pons (P), medulla (MY) and cerebellum (CB). Within each of these major brain structures, there are multiple regions and subregions, each comprising many cell types.

Single-cell transcriptomics by single-cell RNA sequencing (scRNA-seq) or single-nucleus RNA sequencing (snRNA-seq) provides unprecedented profiling depth and scalability, enabling comprehensive quantitative analysis and classification of cell types at scale[2,3,7–9]. Transcriptomically defined cell types have been shown to exhibit concordant morphological and physiological properties[10,11]. Single-cell transcriptomics has been used to categorize cell types from many different regions of the mouse nervous system and increasingly in human and non-human primate brains[2,12]. The BRAIN Initiative Cell Census Network (BICCN) and the Human Cell Atlas (HCA) are representative community efforts that use single-cell transcriptomics to create cell-type atlases for the brain and body of human and other mammals[8,13–16].

An essential next step is to create a comprehensive and high-resolution transcriptomic cell-type atlas for the entire adult brain from

a single mammalian species. The mouse (*Mus musculus*) is the most widely used mammalian model organism and is therefore a natural first choice for a comprehensive definition of mammalian brain composition and architecture. To define the anatomical context for cell types, another critical requirement is to characterize the precise spatial location of each cell type using single-cell-level spatial transcriptomics analysis[17–20] covering the entire mouse brain. In addition to describing a complete, brain-wide cell-type atlas of a mammalian brain, this analysis will enable us to address questions on how the brain-wide transcriptomic landscape of cell types relates to the anatomical and circuit organization and its ontology rooted in development and evolution, and how coordinated gene expression specifies cell-type identity and functional properties.

## Creation of the mouse brain cell-type atlas

As part of the BICCN, we set out to build a comprehensive, high-resolution transcriptomic cell-type atlas for the entire adult mouse brain. We systematically generated two types of large-scale, single-cell-resolution transcriptomic datasets for all mouse brain regions, using scRNA-seq and MERFISH[21]. We used the scRNA-seq data to generate a transcriptomic cell-type taxonomy, and the MERFISH data to visualize and annotate the spatial location of each cluster in this taxonomy, based on the Allen Mouse Brain Common Coordinate Framework version 3 (CCFv3)[22] (Supplementary Table 1 provides the anatomical ontology with full names and acronyms of all brain regions).

We first generated 781 scRNA-seq libraries (using 10x Genomics Chromium v2 (referred to as 10xv2) or v3 (10xv3)) from anatomically defined, CCFv3-guided (Supplementary Table 1) tissue microdissections (Methods), resulting in a dataset of around 7.0 million single-cell transcriptomes (Supplementary Tables 2 and 3), representing approximately 5% of the cells in a mouse brain. We developed a set of stringent quality control (QC) metrics guided by pilot clustering results that informed us on characteristics of low-quality single-cell transcriptomes (Extended Data Fig. 1a–c, Supplementary Table 4 and Methods). We then conducted iterative clustering analysis on around 4.3 million QC-qualified cells using custom software (scrattch.bigcat package developed in-house). The 10xv3 and 10xv2 cells were first clustered separately and then integrated with methods we developed previously[23], resulting in an initial joint transcriptomic cell-type taxonomy with 5,283 clusters (Extended Data Fig. 1a).

By performing all pairwise cluster comparisons in this initial transcriptomic taxonomy, we derived 8,460 differentially expressed genes (DEGs) (Supplementary Table 5) differentiating all pairs of clusters. We then designed two gene panels for the generation of MERFISH data, with each gene panel containing a selected set of marker genes with the greatest combinatorial power to discriminate among all clusters. The first gene panel contained 1,147 genes and was used by the X.Z. laboratory to generate MERFISH datasets from several male and female mouse brains using a custom imaging platform[24]. The second gene panel contained 500 genes (Supplementary Table 6 and Methods) and was used to generate a MERFISH dataset from one male mouse brain at the Allen Institute for Brain Science (AIBS) using the Vizgen MERSCOPE platform (Extended Data Fig. 2). The AIBS MERFISH dataset contained 59 serial full coronal sections at 200-μm intervals spanning the entire mouse brain, with a total of around 4.3 million segmented and QC-passed cells (Extended Data Fig. 2), subsequently registered to the Allen CCFv3 (Methods).

To hierarchically organize the transcriptomic cell-type taxonomy and delineate the relationship between clusters, we first computed Pearson correlations of gene expression between each pair of clusters using all or a subset of DEGs as a measure of similarity between clusters (Extended Data Fig. 3). We found that clusters have different degrees of similarities between them and can be grouped into smaller or larger categories. Furthermore, transcription factor marker genes provide the lowest correlation values across the brain compared with functional marker genes, adhesion molecules and all marker genes, and can best resolve the global relationships among clusters. Therefore, we used transcription factor marker genes to computationally build a cell-type hierarchy, grouping the clusters into putative classes, subclasses and supertypes (Methods).

We used the AIBS MERFISH dataset and one of the MERFISH datasets from the X.Z. laboratory to annotate the spatial location of each subclass, supertype and cluster. To do this, we developed a hierarchical mapping approach (Methods) to map each MERFISH cell to the transcriptomic taxonomy and assign the best matched cluster identity along with a correlation score to each MERFISH cell. The spatial location of each cluster was subsequently obtained by the collective locations of majority of the cells assigned to that cluster with high correlation scores. We annotated each subclass with its most representative anatomical regions and incorporated these annotations into subclass nomenclature for easier recognition of their identities. In this way, the high-level distribution of cell types across the entire mouse brain is described. As the anatomical annotations at subclass level are largely consistent between the X.Z. laboratory and the AIBS MERFISH datasets, the AIBS MERFISH dataset is used to illustrate our results and findings in the subsequent sections of this manuscript.

To finalize the transcriptomic cell-type taxonomy and atlas, we conducted detailed annotation and analysis of all the subclasses, supertypes and clusters on the basis of molecular and spatial relationships among these cell types. During this process, we identified and removed an additional set of 'noise' clusters (usually doublets or mixed debris; Methods) that had escaped the initial QC process, resulting in a final set of around 4.0 million high-quality single-cell transcriptomes (Extended Data Fig. 1a,d,e). We further refined the clustering (Methods) and identified cell types in midbrain and hindbrain that were depleted in our scRNA-seq dataset. These cell types were supplemented with 10x Multiome snRNA-seq data (10xMulti), with a total of 1,687 10xMulti nuclei across 33 clusters added to the taxonomy (Extended Data Fig. 1a and Methods).

Thorough analysis revealed extraordinarily complex relationships among transcriptomic clusters and their associated regions. We further fine-tuned and adjusted class, subclass and supertype memberships of a small fraction of clusters to reach the final definition (Extended Data Fig. 4 and Methods). To organize the complex molecular relationships, we present a high-resolution transcriptomic and spatial cell-type atlas for the whole mouse brain with four nested levels of classification: 34 classes, 338 subclasses, 1,201 supertypes and 5,322 clusters or types (Fig. 1, Extended Data Fig. 5e and Extended Data Table 1). We also grouped the classes into seven neighbourhoods for more in-depth analyses of related subsets of cell types. The neighbourhoods recapitulate to a great extent the molecular and anatomical relatedness among cell types, but they are not part of the cell-type hierarchy because they do not strictly follow the distance relationship among cell types and they contain partially overlapping memberships.

Supplementary Table 7 provides the cluster annotation, including the neighbourhood, class, subclass and supertype assignment for each cluster, as well as the anatomical annotations, marker genes and various metadata information. We provide several representations of this atlas for further analysis: (1) a dendrogram at subclass resolution along with bar graphs displaying various metadata information (Fig. 1a and Extended Data Fig. 5e); (2) uniform manifold approximations and projections (UMAPs) at single-cell resolution coloured with different types of metadata information (Fig. 1b–e); and (3) a constellation diagram at subclass resolution to depict multidimensional relationships among different subclasses (Extended Data Fig. 6).

The high quality of the scRNA-seq and snRNA-seq data included in the final taxonomy is indicated by the high gene counts and unique molecular identifier (UMI) counts across cell types (Extended Data Fig. 5a–d). To test the robustness of the clustering results, we first

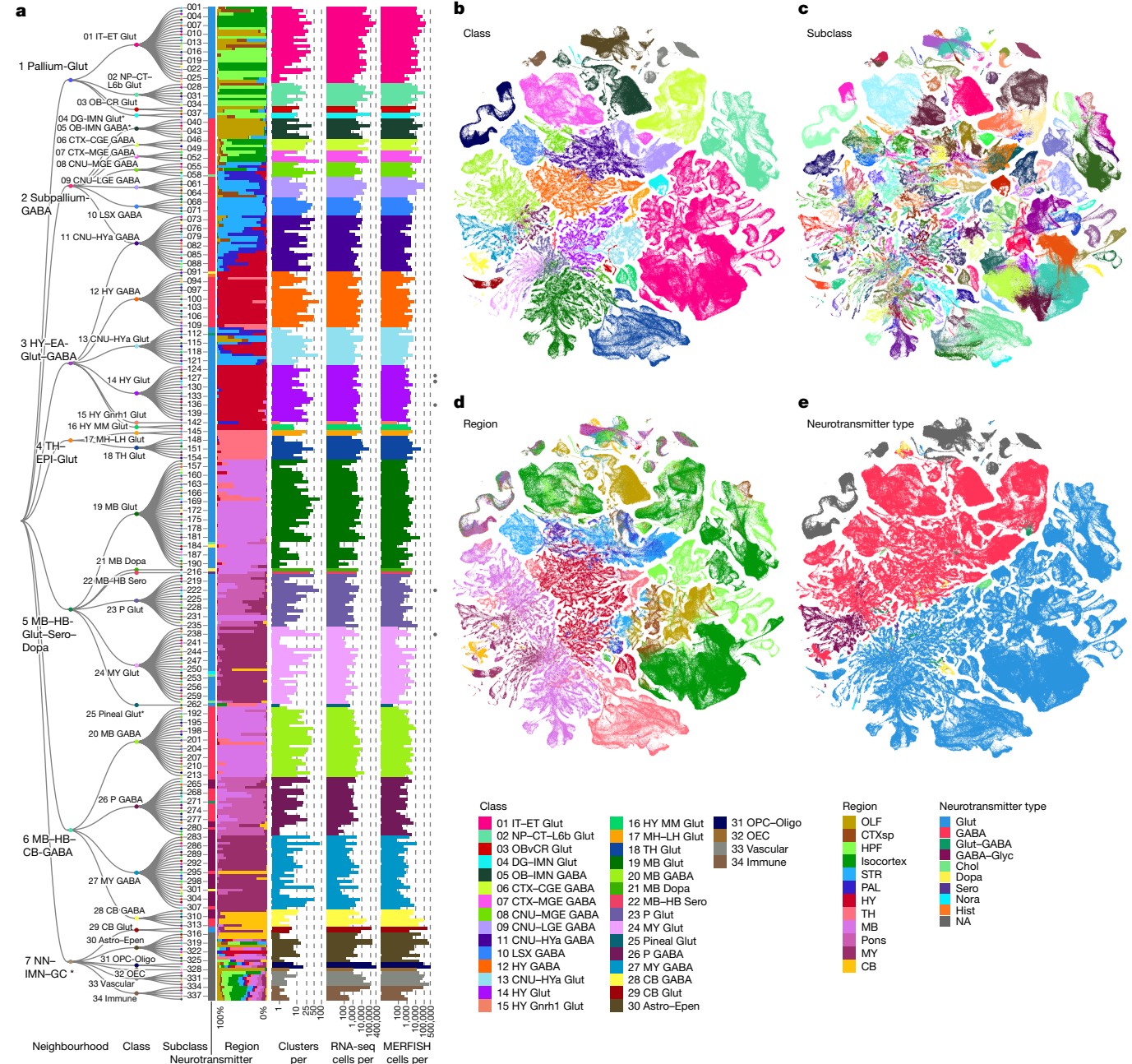

**Fig. 1 | Transcriptomic cell-type taxonomy of the whole mouse brain. a**, Left, the transcriptomic taxonomy tree of 338 subclasses organized in a dendrogram (10xv2: *n* = 1,699,939 cells; 10xv3: *n* = 2,341,350 cells; 10x Multiome: *n* = 1,687 nuclei). The neighbourhood and class levels are marked on the taxonomy tree. Classes marked with asterisks are included in the NN–IMN–GC neighbourhood. The IDs of every third subclass are shown to the right of the dendrogram. Full subclass names are provided in Supplementary Table 7. Following subclass IDs, bar plots represent (left to right): major neurotransmitter type, region distribution of profiled cells, number of clusters per subclass, number of RNA-seq cells analysed per subclass, and number of cells analysed by MERFISH per subclass. Subclasses marked with grey dots contain sex-dominant clusters. Sex-dominant clusters within a subclass are identified by calculating the odds and log *P* value for male/female distribution per cluster. Clusters with odds < 0.2 and log$_{10}$(*P* value) < −10 are considered to be sex-dominant. **b**–**e**, UMAP representation of all cell types coloured by class (**b**), subclass (**c**), brain region (**d**) and major neurotransmitter type (**e**). Colour schemes for **a**–**e** are shown in

the key at the bottom right of the figure. Astro, astrocyte; CB, cerebellum; CGE, caudal ganglionic eminence; CNU, cerebral nuclei; CR, Cajal–Retzius; CT, corticothalamic; CTX, cerebral cortex; CTXsp, cortical subplate; DG, dentate gyrus; EA, extended amygdala; Epen, ependymal; EPI, epithalamus; ET, extratelencephalic; GC, granule cell; HB, hindbrain; HPF, hippocampal formation; HY, hypothalamus; HYa, anterior hypothalamic; IMN, immature neurons; IT, intratelencephalic; L6b, layer 6b; LGE, lateral ganglionic eminence; LH, lateral habenula; LSX, lateral septal complex; MB, midbrain; MGE, medial ganglionic eminence; MH, medial habenula; MM, medial mammillary nucleus; MY, medulla; NN, non-neuronal; NP, near-projecting; OB, olfactory bulb; OEC, olfactory ensheathing cells; OLF, olfactory areas; Oligo, oligodendrocytes; OPC, oligodendrocyte precursor cells; P, pons; PAL, pallidum; STR, striatum; TH, thalamus. Neurotransmitter types: Chol, cholinergic; Dopa, dopaminergic; GABA, GABAergic; Glut, glutamatergic; Glyc, glycinergic; Hist, histaminergic; Nora, noradrenergic; Sero, serotonergic; NA, not applicable (no neurotransmitter detected).

performed five-fold cross-validation using all 8,460 markers as features for classification, to assess how well the cells could be mapped to the cell types they were originally assigned to. The median classification accuracy was $0.87 \pm 0.10$ (median ± s.d.) and $0.98 \pm 0.03$ for all clusters and all subclasses, respectively. Next, we evaluated the integration between 10xv2, 10xv3, 10xMulti and MERFISH transcriptomes (Extended Data Fig. 7a–c). The median correlation between 10xv2 and 10xv3 is $0.89 \pm 0.09$ and that between 10xv3 and MERFISH data is $0.91 \pm 0.20$ (Extended Data Fig. 7d), suggesting that most marker genes show consistent relative expression levels at cluster level across platforms. The MERFISH dataset can resolve the vast majority of clusters owing to strong correlation of DEG expression between 10xv3 and MERFISH clusters (Extended Data Fig. 7e–g).

To further integrate the transcriptomic and spatial profiles of each cell type and even each single cell, we computationally imputed the 10xv3 scRNA-seq data into the MERFISH space by searching for the $k$-nearest neighbours (KNNs) among 10xv3 cells for each MERFISH cell, using the 500 MERFISH genes (Methods). To test the accuracy of MERFISH imputation, we excluded one gene from the gene panel at a time from the KNN computation and compared its imputed gene expression with its original gene expression. High correlations between imputed expression and the original MERFISH expression, as well as the reference 10xv3 expression for each gene were observed (Extended Data Fig. 8a). The imputed spatial expression patterns were consistent with the actual expression patterns by both MERFISH and in situ hybridization from the Allen Brain Atlas[25] for genes that are on the 500 MERFISH gene panel (*Calb2*, *Baiap3* and *Lypd1*) and those that are not (*Foxp2*) (Extended Data Fig. 8b–f). *Foxp2* is expressed with $\log_2$(counts per million mapped reads (CPM)) > 3 in 1,340 clusters among 177 subclasses and 27 classes, which is exemplary of the overall accuracy of MERFISH gene imputation.

## An interactive online platform for the atlas

To facilitate the wide dissemination of data and utilization of the comprehensive mouse whole-brain cell-type atlas, we have developed the Allen Brain Cell Atlas. This platform, accessible at https://portal.brain-map.org/atlases-and-data/bkp/abc-atlas, is designed to visualize extensive scRNA-seq, snRNA-seq and MERFISH datasets, organized according to the whole-brain cell-type taxonomy, along with accompanying metadata. The Allen Brain Cell Atlas leverages a service-oriented architecture and is hosted on Amazon Web Services, ensuring efficient access and robust performance.

The Allen Brain Cell Atlas enables researchers to explore the landscape of cell types across various hierarchical levels and brain regions. Users can delve into specific cell types, examine their spatial distributions, study gene expression patterns, explore co-expression relationships, or investigate the composition of cell types within distinct brain regions. Additionally, the Allen Brain Cell Atlas provides valuable links to related resources, including an open source project repository for data download, complete with comprehensive documentation and a Jupyter Notebook that illustrates data retrieval and analysis techniques (available at https://alleninstitute.github.io/abc_atlas_access/intro.html). To foster a supportive research community, we offer a dedicated community forum where users can find a user guide, seek assistance and exchange knowledge. This forum, which is monitored by members of the Allen Brain Cell Atlas team, can be accessed at https://community.brain-map.org/c/how-to/abc-atlas/19/l/top.

Furthermore, we have developed the MapMyCells tool (https://portal.brain-map.org/atlases-and-data/bkp/mapmycells), which enables researchers to upload and use our cell-type mapping solution based on the hierarchical mapping tools that we have developed (https://github.com/AllenInstitute/scratch.mapping). This tool facilitates integrating and comparing their scRNA-seq and/or snRNA-seq data with the reference taxonomy of cell types in whole brain of mouse, including high-quality single-cell transcriptomes. By doing so, researchers can gain valuable insights into their data mapped against a reference and accelerate their investigations.

## Neuronal cell types across the mouse brain

Neuronal cell types constitute a large proportion of the whole-brain cell-type atlas, including 6 neighbourhoods, 29 classes (85%), 315 subclasses (93%), 1,156 supertypes (96%) and 5,205 clusters (98%) (Extended Data Table 1 and Supplementary Table 7). Neuronal types have high regional specificity and exhibit highly variable degrees of similarities and differences. To further investigate the neuronal diversity within each major brain structure, we generated re-embedded UMAPs (in 2D and 3D) for the neighbourhoods of neuronal types described above, to reveal fine-grained relationships between neuronal types within and between brain regions in conjunction with the MERFISH data. The results shown in Fig. 2 reveal a marked correspondence between transcriptomic specificity and relatedness and spatial specificity and relatedness among the different neuronal subclasses.

Glutamatergic neurons from all pallium structures, including isocortex, HPF, OLF and CTXsp, form a distinct Pallium-Glut neighbourhood that includes subclasses 1–38 and a total of 517 clusters (Figs. 1a and 2a,b, Extended Data Table 1, Extended Data Fig. 6 and Supplementary Table 7). Here, each neuronal subclass exhibits layer and/or region specificity (Fig. 2a,b). We found that the parallel relationships of the different subclasses of glutamatergic neurons between isocortex and HPF that we had reported previously[23] extend to other pallium structures—that is, OLF and CTXsp. We also observed that the NP–CT–L6b-like (NP, near-projecting; CT, corticothalamic; L6b, layer 6b) subclasses emerge as a group highly distinct from the IT–ET-like (IT, intratelencephalic; ET, extratelencephalic) subclasses[13,23,26,27], forming two distinct classes, IT–ET Glut and NP–CT–L6b Glut. In addition, we uncovered relatedness between the Cajal–Retzius (CR) cells mostly found in HPF (subclass 036 HPF CR Glut) and the olfactory bulb (OB) glutamatergic subclass, 035 OB Eomes Ms4a15 Glut, which are likely mitral and tufted cells[28], and grouped them into the OB–CR Glut class (Extended Data Fig. 6). Finally, this neighbourhood includes the DG–IMN Glut class which contains both the dentate gyrus (DG) granule cells and the immature neurons found in DG and the piriform cortex (PIR) that are involved in adult neurogenesis (see below).

A set of developmental subpallium-derived GABAergic (γ-aminobutyric acid-producing) neuronal subclasses, including all GABAergic neurons found in pallium structures and those in the subpallial CNU, including dorsal STR (STRd) and ventral STR (STRv), lateral septal complex (LSX), and dorsal PAL (PALd), ventral PAL (PALv) and medial PAL (PALm), form a Subpallium-GABA neighbourhood (Figs. 1a and 2c,d and Extended Data Fig. 6). On the basis of the molecular signature and regional specificity of each subclass, the Subpallium-GABA neighbourhood (subclasses 39–90, total 1,051 clusters) was divided into 7 classes that are likely related to their distinct developmental origins[29,30] (Fig. 2c,d, Extended Data Table 1 and Supplementary Table 7): CTX–CGE GABA (containing cortical/pallial GABAergic neurons derived from the caudal ganglionic eminence), CTX–MGE GABA (containing cortical/pallial GABAergic neurons derived from the medial ganglionic eminence), CNU–MGE GABA (containing striatal/pallidal GABAergic neurons derived from MGE), CNU–LGE GABA (containing striatal/pallidal GABAergic neurons derived from the lateral ganglionic eminence), LSX GABA (containing lateral septum GABAergic neurons derived from the embryonic septum[31]), CNU–HYa GABA (containing striatal/pallidal and anterior hypothalamic GABAergic neurons potentially derived from the embryonic preoptic area (POA)), and OB–IMN GABA (containing olfactory bulb GABAergic neurons potentially derived from LGE, as well as the olfactory bulb-destined immature neurons involved in adult neurogenesis (see below)).

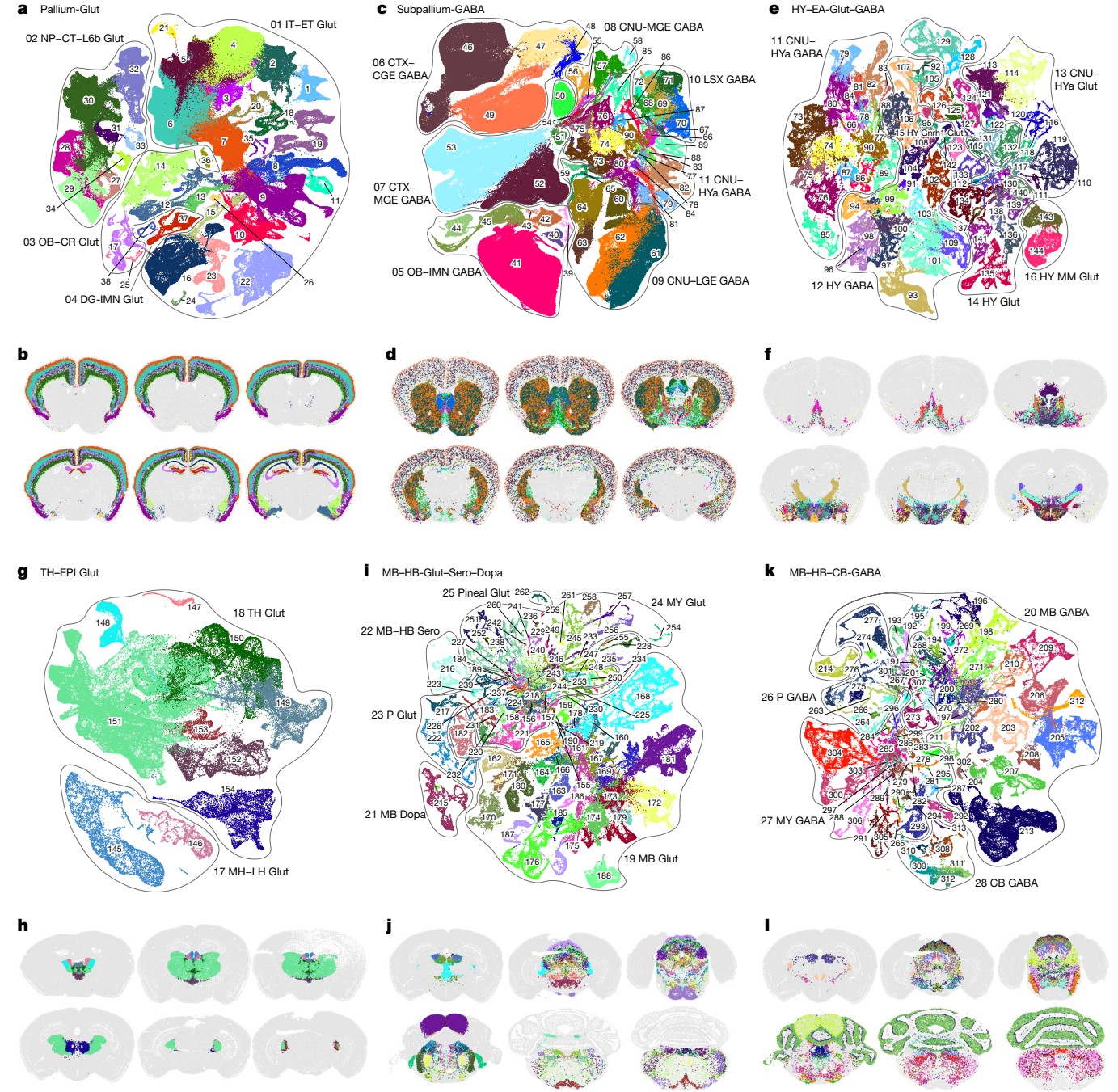

**Fig. 2 | Neuronal cell-type classification and distribution across the brain.**
**a–l**, UMAP representation (**a**,**c**,**e**,**g**,**i**,**k**) and representative MERFISH sections
(**b**,**d**,**f**,**h**,**j**,**l**) of Pallium-Glut (**a**,**b**), Subpallium-GABA (**c**,**d**), HY–EA-Glut–GABA
(**e**,**f**), TH–EPI-Glut (**g**,**h**), MB–HB-Glut–Sero–Dopa (**i**,**j**) and MB–HB-CB-GABA

(**k**,**l**) neighbourhoods coloured by subclass. Each subclass is labelled with its ID
and shown in the same colour in UMAP representations and MERFISH sections.
**a**,**c**,**e**,**g**,**i**,**k**, Outlines in UMAP representations show cell classes. For full
subclass names, see Supplementary Table 7.

The HY–EA-Glut–GABA neighbourhood (including subclasses 66 and
73–144, total 1,404 clusters) contains a set of closely related neuronal
subclasses from the entire hypothalamus[32,33], as well as the striatum-like
amygdalar nuclei (sAMY) and caudal PAL regions of CNU that are also
known as the extended amygdala (Figs. 1a and 2e,f and Extended Data
Fig. 6). Both glutamatergic and GABAergic neuronal subclasses in this
neighbourhood exhibit a gradual anterior-to-posterior transition, and
thus were grouped into six classes (Fig. 2e,f, Extended Data Table 1
and Supplementary Table 7): CNU–HYa GABA, HY GABA, CNU–HYa
Glut, HY Glut, HY Gnrh1 Glut and HY MM Glut (MM, medial mammillary
nucleus). Neuronal types in the most anterior part of hypothalamus,

the POA, are highly similar to neuronal types in sAMY and PAL. Thus,
the CNU–HYa GABA class is also included in the Subpallium-GABA
neighbourhood described above to show their relatedness and con-
tinuity with the striatal/pallidal types (Fig. 2c,d). The more posterior
HY GABA class also includes GABAergic neurons from the thalamic
reticular nucleus (RT) (subclass 93) and the ventral part of the lateral
geniculate complex (LGv) (subclass 109), which are closely related to
zona incerta (ZI) neurons in hypothalamus (subclass 101), revealing a
relationship of GABAergic types between hypothalamus and thalamus
that is consistent with their developmental origins. Both RT and ZI
neurons may have originated from the prethalamus or the zona limitans

intrathalamica (ZLI)[34–38]. The HY Gnrh1 Glut class is the hypothalamic Gnrh1 neuronal type developmentally originated from the embryonic olfactory epithelium[39].

The fourth neuronal neighbourhood, TH–EPI-Glut (subclasses 145–154, total 148 clusters), contains all glutamatergic neuronal subclasses located in the thalamus, as well as the medial habenula (MH) and lateral habenula (LH), which collectively compose the epithalamus (EPI) (Figs. 1a and 2g,h and Extended Data Fig. 6). These subclasses were grouped correspondingly into TH Glut and MH–LH Glut classes.

The fifth neuronal neighbourhood, MB-HB-Glut–Sero–Dopa, contains all glutamatergic, serotonergic and dopaminergic neuronal types in midbrain (MB) and hindbrain (HB) (Figs. 1a and 2i,j and Extended Data Fig. 6). The neighbourhood, the largest and most complex, includes 6 classes, 84 subclasses and 1,431 clusters (Fig. 2i,j, Extended Data Table 1 and Supplementary Table 7). MB Glut, P Glut and MY Glut are the three largest classes, containing 37, 18 and 26 subclasses, respectively. By contrast, the MB Dopa, MB–HB Sero and Pineal Glut classes each contains a single subclass. Note that we did not include the CB Glut class in this neighbourhood but placed it in the NN–IMN–GC neighbourhood instead (see below), because CB Glut contains the cerebellar granule cells that are highly distinct from the midbrain or hindbrain neuronal types.

The sixth and final neuronal neighbourhood, MB–HB–CB-GABA, contains all GABAergic subclasses located in midbrain, hindbrain and cerebellum (Figs. 1a and 2k,l and Extended Data Fig. 6). This neighbourhood includes 4 classes (MB GABA, P GABA, MY GABA and CB GABA), 75 subclasses and 1,040 clusters (Fig. 2k,l, Extended Data Table 1 and Supplementary Table 7).

We found more transitional cell types across brain structures, which again may be owing to unique developmental origins. For example, both glutamatergic and GABAergic subclasses from the cerebellar nuclei (CBN), 250 CBN Neurod2 Pvalb Glut and 295 CBN Dmbx1 Gaba, are more closely related to those from the medulla than those from the cerebellar cortex (CBX), and they are included in MY Glut and MY GABA classes, respectively. Glutamatergic subclass 168 SPA–SPFm–SPFp–POL–PIL–PoT Sp9 Glut and GABAergic subclass 203 LGv-SPFp-SPFm Nkx2-2 Tcf7l2 Gaba belong to MB Glut and MB GABA classes, respectively, but they are both located in various posterior thalamic nuclei, suggesting potential migration of these neurons from midbrain pretectal area to thalamus[40] (SPA, subparafascicular area; SPFm, subparafascicular nucleus, magnocellular part; SPFp, subparafascicular nucleus, parvicellular part; POL, posterior limiting nucleus of the thalamus; PIL, posterior intralaminar thalamic nucleus; PoT, posterior triangular thalamic nucleus).

## Neurotransmitter and neuropeptide expression

We systematically assigned neurotransmitter identity to each cell cluster on the basis of the co-expression of canonical neurotransmitter transporter genes and synthesizing enzymes and considering alternative neurotransmitter release mechanisms (Figs. 1e and 3, Extended Data Figs. 5e and 9, Supplementary Table 7 and Methods).

These marker genes indicate that the majority of neuronal clusters release a single neurotransmitter—either glutamate or GABA. Many GABAergic neuronal clusters in midbrain and hindbrain co-release glycine. We identified 62 clusters with glutamate–GABA dual transmitters (Glut–GABA), most of which express the glutamate transporter genes *Slc17a6* or *Slc17a8* (Extended Data Fig. 9). These clusters are widely distributed in different parts of the brain. They include four clusters in the isocortex and hippocampus and three clusters in globus pallidus, internal segment (GPi), which probably correspond to previously well-characterized glutamate–GABA co-releasing neuronal types in these regions[41,42]. They also include a few clusters each in STRv, PALv, several hypothalamus areas including arcuate hypothalamic nucleus (ARH) and supramammillary nucleus (SUM), several midbrain areas including ventral tegmental area (VTA), pedunculopontine nucleus

(PPN) and interpeduncular nucleus (IPN), areas in pons such as superior central nucleus raphe (CS), nucleus raphe pontis (RPO), nucleus incertus (NI), posterodorsal tegmental nucleus (PDTg), and others. Notably, except for the three Glut–GABA clusters that form an exclusive subclass in GPi (subclass 112), the other Glut–GABA clusters are present in subclasses that also contain closely related single-neurotransmitter (glutamate or GABA) clusters (Extended Data Fig. 9 and Supplementary Table 7).

We also systematically identified all clusters that produce modulatory neurotransmitters (Fig. 3 and Supplementary Table 7). Cholinergic neurons[43,44] are found mainly in subclass 58 in the ventral PAL (11 clusters), but also include 2 clusters in LSX, 8 clusters in MH, 3 clusters in PPN, 5 clusters in dorsal motor nucleus of the vagus nerve (DMX) and nucleus of the solitary tract (NTS), and approximately 13 clusters scattered in other medulla nuclei. We also found *Slc18a3* and *Chat* expression in several clusters in the Vip GABA subclass in isocortex, but its expression at cluster level did not cross our threshold to label these clusters as cholinergic. Cholinergic neurons often co-release glutamate (24 clusters out of 48), sometimes GABA (7 clusters), both glutamate and GABA (3 clusters), or dopamine (1 cluster in DMX).

Dopaminergic neurons[45] are found predominantly in subclass 215 (containing 43 clusters), which is the sole member of the MB Dopa class, as well as an additional 28 clusters spread across 14 subclasses. Subclass 215, located in substantia nigra, compact part (SNc), VTA and midbrain raphe nuclei (RAmb) areas, displays the most heterogeneous neurotransmitter content. It contains 39 dopaminergic clusters and 4 dual Glut–GABA clusters. Most (35) of the 39 dopaminergic clusters also co-release glutamate (11 clusters) or GABA (10 clusters), or both glutamate and GABA (14 clusters). We identified clusters that correspond anatomically to all classically defined dopaminergic neuron groups—A8–A16—across the brain; there were four clusters in the A16 main olfactory bulb (MOB) group, two clusters in the A15 rostral hypothalamus group, eight clusters in the A14 periventricular hypothalamus group, one cluster in the A13 ZI group, five clusters in the A12 ARH group, two clusters in the A11 posterior hypothalamus group, ten clusters in the A10 VTA group, nine clusters in the A9 SNc group, and two clusters in the A8 retrorubral group[46–50] (Supplementary Table 7). Beyond these groups, we also found dopaminergic neuronal types in other brain regions, including many clusters (22) in RAmb and periaqueductal gray (PAG) as well as 3 clusters in dorsomedial nucleus of the hypothalamus (DMH).

Serotonergic neurons[51] all belong to the single subclass 216, which solely comprises the distinct MB–HB Sero class. This subclass consists of 20 serotonergic clusters and 12 glutamatergic (marked by *Slc17a8*) clusters that are all closely related to each other. The serotonergic neurons often co-release glutamate (8 clusters), GABA (4 clusters), or glutamate and GABA (7 clusters). All of these clusters reside in the various raphe nuclei within midbrain or medulla. Thus, the serotonergic neuron class and/or subclass is highly heterogeneous in both neurotransmitter content and spatial localization. Of note, even though many serotonergic and dopaminergic clusters are colocalized in the RAmb areas, they are well segregated in the gene expression space, and we found no clusters that could co-release serotonin and dopamine, as no clusters co-express the key synthesis genes *Th* and *Tph2*.

Noradrenergic neurons[52,53] are found exclusively in subclass 251. This subclass contains 12 noradrenergic clusters and 14 glutamatergic clusters, with the noradrenergic clusters also co-releasing glutamate (marked by *Slc17a6*; 10 clusters), or GABA (1 cluster), or glutamate and GABA (1 cluster). All but four clusters in this subclass are located in NTS; of the remaining clusters, one glutamatergic cluster is located in locus ceruleus (LC) and one noradrenergic cluster is located in both locus ceruleus and subceruleus nucleus (SLC).

Histaminergic neurons are found exclusively in the tuberomammillary nucleus, dorsal part (TMd) and ventral part (TMv), of hypothalamus (5 clusters in subclass 92), and all co-release GABA[54].

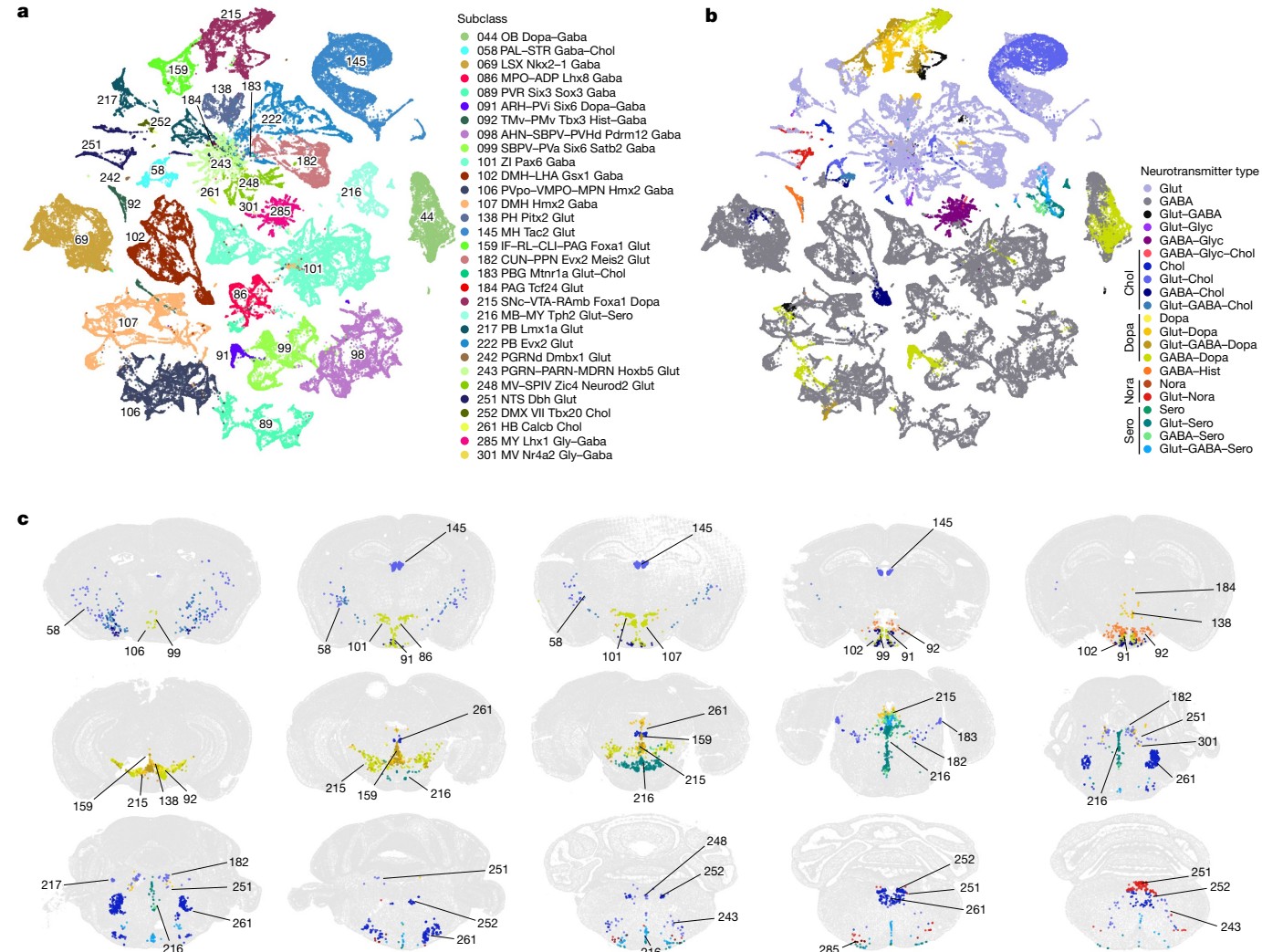

**Fig. 3 | Modulatory neurotransmitter types and their distribution throughout the brain. a,b,** Neuronal subclasses containing clusters that release modulatory neurotransmitters and their various co-release combinations with glutamate and/or GABA. UMAPs are coloured by subclass (**a**) and neurotransmitter type (**b**). **c,** Representative MERFISH sections showing the location of neuronal types expressing modulatory neurotransmitters. Cells are coloured by neurotransmitter type and labelled by subclass ID. See Supplementary Table 7 for detailed neurotransmitter assignment for each cluster. ADP, anterodorsal preoptic nucleus; AHN, anterior hypothalamic nucleus; ARH, arcuate hypothalamic nucleus; CLI, central linear nucleus raphe; CUN, cuneiform nucleus; DMH, dorsomedial nucleus of the hypothalamus; DMX, dorsal motor nucleus of the vagus nerve; IF, interfascicular nucleus raphe; LHA, lateral hypothalamic area; MDRN, medullary reticular nucleus; MPN, medial preoptic nucleus; MPO, medial preoptic area; MV, medial vestibular nucleus; NTS, nucleus of the solitary tract; PAG, periaqueductal grey; PARN, parvicellular reticular nucleus; PB, parabrachial nucleus; PBG, parabigeminal nucleus; PGRN, paragigantocellular reticular nucleus; PGRNd, paragigantocellular reticular nucleus, dorsal part; PH, posterior hypothalamic nucleus; PMv, ventral premammillary nucleus; PPN, pedunculopontine nucleus; PVa, periventricular hypothalamic nucleus, anterior part; PVHd, paraventricular hypothalamic nucleus, descending division; PVi, periventricular hypothalamic nucleus, intermediate part; PVpo, periventricular hypothalamic nucleus, preoptic part; PVR, periventricular region; RAmb, midbrain raphe nuclei; RL, rostral linear nucleus raphe; SBPV, subparaventricular zone; SNc, substantia nigra, compact part; SPIV, spinal vestibular nucleus; TMv, tuberomammillary nucleus, ventral part; VII, facial motor nucleus; VMPO, ventromedial preoptic nucleus; VTA, ventral tegmental area; ZI, zona incerta.

Overall, a pattern emerged where nearly all subclasses with a dominant modulatory neurotransmitter contain clusters transmitting glutamate and/or GABA only, as well as various patterns of co-transmission, indicating a high degree of heterogeneity in neurotransmitter release and co-release among closely related neuronal types that may have common developmental origins. Our QC process excluded the possibility of doublet or low-quality cell contamination accounting for the heterogeneity. Although many of these neurotransmitter co-release patterns had been documented previously[55–57], our study defined a comprehensive set of cell types with unique and differing neurotransmitter content that can be identified through combinations of marker genes.

Neuropeptides are also major agents for intercellular communications in the brain[58,59]. We examined cell-type-specific expression patterns of dozens of main neuropeptide genes and their receptors in our datasets (Supplementary Table 7). We measured the cell-type specificity of expression of these genes using the Tau score[60] and found a wide range of variation (Extended Data Fig. 10a,b). Some neuropeptides are widely expressed in many cell types or clusters and at high levels (for example, *Cck, Pnoc, Adcyap1, Penk, Sst* and *Tac1*), some are expressed at high levels in a moderate number of clusters (for example, *Cartpt, Nts, Pdyn, Gal, Tac2, Grp, Vip, Crh, Trh* and *Cort*), and others are highly expressed specifically in only one or few clusters (for example, *Avp, Agrp, Pomc, Pmch, Oxt, Rln3, Npw, Nps, Ucn, Hcrt, Gnrh1, Gcg* and

*Pyy*) (Extended Data Fig. 10c–f). About 79% of all clusters express at least one neuropeptide gene, and there are numerous co-expression combinations of different neuropeptides in many clusters, with high degrees of variations within subclasses (Supplementary Table 7). Our datasets provide a rich resource for the exploration of neuropeptide ligand–receptor interactions across the entire brain.

## Non-neuronal and immature neuronal cell types

Unlike the six neuronal neighbourhoods defined above, the seventh and final neighbourhood, NN–IMN–GC, contains a mixed collection of highly distinct non-neuronal cell types, immature neuronal types and granule cell types (Fig. 4a). It has nine classes, including five non-neuronal classes (Astro–Epen, OPC–Oligo, OEC, Vascular and Immune) and four granule and immature neuronal classes (DG–IMN Glut, OB–IMN GABA, Pineal Glut and CB Glut).

All non-neuronal cell types across the mouse brain are classified into 5 classes, 23 subclasses, 45 supertypes and 117 clusters (Figs. 1a, 4a, Extended Data Table 1 and Supplementary Table 7), which can be distinguished by highly specific marker genes at all levels of hierarchy (Extended Data Fig. 11a–f). The Astro–Epen class is the most complex, containing ten subclasses, five of which represent astrocytes that are specific to different brain regions: Astro-OLF, Astro-TE (for telencephalon), Astro-NT (for non-telencephalon), Astro-CB and Bergmann glia, whereas the other five subclasses are ependymal cell types: astroependymal cells, ependymal cells, tanycytes, hypendymal cells and choroid plexus (CHOR) cells (Fig. 4a–c). The OPC–Oligo class contains two subclasses, oligodendrocyte precursor cells (OPC) and oligodendrocytes (Oligo). The Oligo subclass is further divided into four supertypes corresponding to different stages of oligodendrocyte maturation: committed oligodendrocyte precursors (COP), newly formed oligodendrocytes (NFOL), myelin-forming oligodendrocytes (MFOL), and mature oligodendrocytes (MOL) (Extended Data Fig. 11i). The OEC class corresponds to olfactory ensheathing cells (OEC). The Vascular class consists of 5 subclasses: arachnoid barrier cells (ABC), vascular leptomeningeal cells (VLMC), pericytes (Peri), smooth muscle cells (SMC) and endothelial cells (Endo). The Immune class consists of 5 subclasses: microglia, border-associated macrophages (BAM), monocytes, dendritic cells (DC) and lymphoid cells, which contains B cells, T cells, natural killer (NK) cells and innate lymphoid cells (ILC).

We identified transcription factors that potentially serve as master regulators for many of these non-neuronal cell types (Extended Data Fig. 11d), many of which were well documented[61–66]. For example, *Sox2*, a well-known radial glia marker, is widely expressed in astrocytes and oligodendrocytes. *Sox9* is specific to the Astro–Epen class, *Sox10* is specific to the OPC–Oligo class, *Foxd3* and *Hey2* are specific to OEC, *Foxc1* is specific to the Vascular class, and *Ikzf1* is specific to the Immune class. Within each class, additional transcription factors mark finer groupings (Extended Data Fig. 11d–f). For example, Astro-TE cells express *Foxg1* and *Emx2*, which are key regulators of neurogenesis in the telencephalon. Similarly, Astro-CB cells express *Pax3*, which is also highly expressed in GABAergic neurons in the cerebellum. These observations are consistent with the notion that astrocytes and neurons are derived from common regionally distinct progenitors and share common transcription factors for spatial patterning[66,67]. Among other astrocyte-related subclasses, *Nkx2-2* is specific to Bergmann glia, *Rax* is specific to tanycytes, *Myb* is specific to ependymal cells, *Spdef* is specific to hypendymal cells, and *Lef1* is specific to CHOR cells.

The spatial distribution of all non-neuronal cell types in the mouse brain was confirmed and further refined by the MERFISH data. We observed an inside-out spatial gradient in MOB among the four OEC clusters (Extended Data Fig. 11g). In addition to being widely distributed across the brain, oligodendrocytes are also highly concentrated in white matter fibre tracts; by contrast, the 1180 OPC NN_2 supertype is found mostly in grey matter areas (Extended Data Fig. 11i–k).

Of all the non-neuronal cell types, the Astro–Epen class exhibits the most diverse spatial patterns[68,69]. Region-specific astrocytes Astro-OLF, Astro-TE, Astro-NT and Astro-CB are arranged in the UMAP in an anterior-to-posterior order (Fig. 4b), consistent with their spatial patterning. Many astrocyte clusters exhibit further subregion specificity: Astro-TE cluster 5228 is specific to hippocampal region and CTXsp, 5227 is specific to STRd, 5226 is specific to LSX and midline cortical areas, 5225 is specific to isocortex/OLF, 5223 and 5222 are specific to dentate gyrus; Astro-NT cluster 5215 is specific to thalamus, and 5217 is specific to CBN, dorsal cochlear nucleus (DCO) and ventral cochlear nucleus (VCO). Astro-TE clusters 5229 and 5230 and clusters in the Astro-OLF subclass match the path of the rostral migratory stream[70–72] (RMS; see below). Astro-TE cluster 5219 is located at the pia of telencephalon (Fig. 4b) and has high expression of *Gfap* (Extended Data Fig. 11e), consistent with the definition of interlaminar astrocytes (ILAs)[73]. Other clusters (5208, 5209, 5210 and 5211) in the Astro-NT subclass are also localized at the pia with high expression of *Gfap*, which we hypothesize to be ILAs outside telencephalon.

The five ependymal subclasses—Astroependymal, Ependymal, Tanycyte, Hypendymal and CHOR—line different parts of the ventricles throughout the brain, and the clusters within them exhibit exquisite spatial specificity (Fig. 4c). Circumventricular organs (CVOs) are specialized structures located around the third and fourth ventricles that mediate communications between brain, blood and cerebrospinal fluid[74,75] (CSF). They are highly vascularized and are lined with ependymal cells and tanycytes that act as a selective barrier between brain and blood and/or CSF. Tanycytes are specialized ependymal cells that line the third ventricle (V3) and the median eminence (ME) in the hypothalamus[76]. They are classified into four subtypes, and we identified clusters corresponding to each: clusters 5245/5246, 5247, 5249 and 5250 as α1, α2, β1 and β2 tanycytes, respectively (Fig. 4e). We also identified tanycyte-like ependymal cell clusters that are specifically located in other CVOs (Fig. 4c): cluster 5243 in the subfornical organ (SFO), 5244 in the vascular organ of the lamina terminalis (OV (also known as OVLT)), 5240 in area postrema (AP), and the hypendymal cell cluster 5263 (marked by *Sspo*) (Extended Data Fig. 11e) in the subcommissural organ[77] (SCO). Many of these clusters express radial glia marker genes such as *Gfap*, *Sox2*, *Nkain4* and *Pax6*, suggesting that these types have neurogenic potential, which corroborates findings indicating the existence of neural stem cells in the OVLT, SFO, ME and AP[78–81].

VLMC types[66,82] also show highly specific spatial and colocalization patterns. Clusters 5296–5299 are located at the pia, in contrast to clusters 5300 and 5301 which are scattered widely in the brain (Fig. 4d). Notably, we found highly specific spatial colocalization between VLMC cluster 5303 and Tanycyte clusters (Fig. 4e), between VLMC cluster 5302 and Ependymal and CHOR clusters (Fig. 4f), and between pia-specific VLMC clusters and ILAs (Extended Data Fig. 11h). Marker genes for VLMC clusters are enriched in extracellular matrix components and transmembrane transporters, including collagens and solute carriers with distinct cell-type specificity (Extended Data Fig. 11f). The tanycyte-interacting VLMC cluster 5303 does not express many markers present in other VLMC types but has specific expression of transmembrane genes *Tenm4* and *Tmtc2*. Interactions between various VLMC and ependymal cell clusters, together with ABCs, likely regulate the movement of molecules and cells across the barriers between brain and blood or CSF[82].

Cell proliferation and neuronal differentiation continue in adulthood only in restricted areas of the brain[83]. The two main adult neurogenic niches are the dentate gyrus and the subventricular zone (SVZ) lining the lateral ventricles. The first gives rise to the excitatory DG granule cells, whereas the second produces migrating cells that follow the RMS and in the olfactory bulb differentiate into inhibitory OB granule cells[72,84,85]. We identified two subclasses of immature neurons, 38 DG–PIR Ex IMN grouped with glutamatergic granule cells in DG to form the DG–IMN Glut class, and 45 OB–STR–CTX Inh IMN grouped with

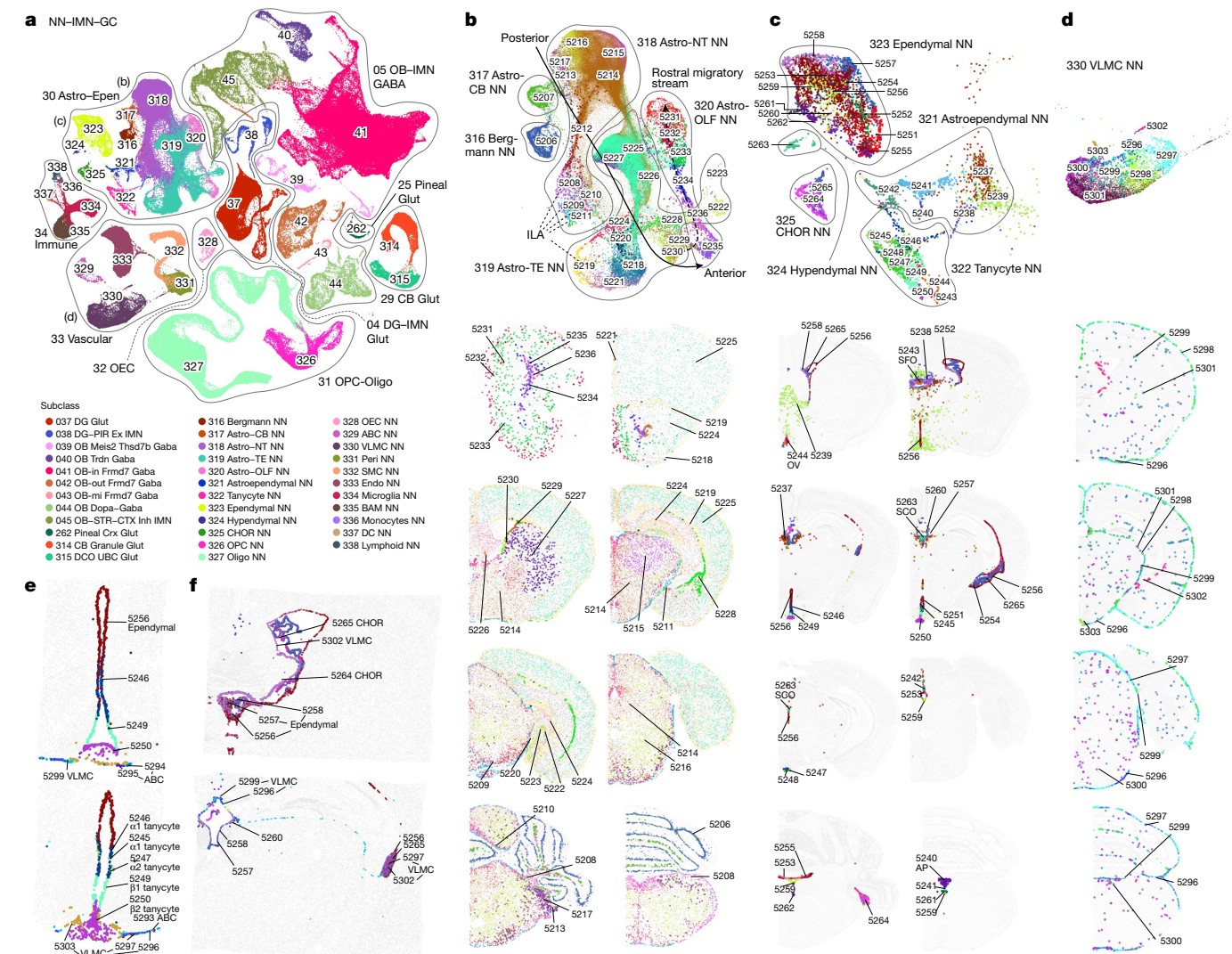

**Fig. 4 | Non-neuronal cell types and immature neuronal types. a**, UMAP representation of the NN–IMN–GC neighbourhood coloured by subclass. Outlines show cell classes. **b**–**d**, Three subpopulations indicated in **a** are highlighted and further investigated: astrocytes (**b**), ependymal cells (**c**) and VLMCs (**d**). UMAP representation and representative MERFISH sections of astrocytes (**b**), ependymal cells (**c**) and VLMCs (**d**) are coloured and numbered by cluster. **b**,**c**, Outlines in UMAPs show subclasses. **e**, Colocalization of tanycyte, ependymal cell and VLMC clusters around V3 and ME, as shown in selected MERFISH sections. **f**, Colocalization of VLMC, CHOR and ependymal cell clusters in various ventricles, as shown in selected MERFISH sections. ABC, arachnoid barrier cells; BAM, border-associated macrophages; CHOR, choroid plexus; DC, dendritic cells; DCO, dorsal cochlear nucleus; Endo, endothelial cells; NT, non-telencephalon; Peri, pericytes; PIR, piriform cortex; SMC, smooth muscle cells; TE, telencephalon; UBC, unipolar brush cells; VLMC, vascular leptomeningeal cells.

GABAergic neuron subclasses in OB[28] to form the OB-IMN GABA class (Fig. 4a and Supplementary Table 7).

The scRNA-seq data show a trajectory from immature neurons to mature neurons in DG, and the MERFISH data corroborate that the immature neurons are located in the subgranular zone (SGZ) of DG, whereas the mature neurons reside in the dentate granular cell layer (Extended Data Fig. 12a,b). Immature neurons in the SGZ, SVZ and RMS have a shared gene expression pattern that includes the expression of immature neuron markers such as *Draxin, Prox1, Mex3a* and *Dcx* (Extended Data Fig. 12e). Besides the shared gene expression patterns in DG and OB trajectories, distinct gene expression patterns include more lineage-specific genes, such as *Rbfox3* and *Frmd7* for more mature OB neurons, and *C1ql2* and *Smad3* for mature DG neurons (Extended Data Fig. 12f,g).

The migrating neurons in the RMS are separated from the parenchyma by astrocytes that form tunnels through which the cells migrate[71,86]. Astro-TE clusters 5229 and 5230, located in the lateral ventricle bordering rostral dorsal STR, and clusters belonging to the Astro-OLF subclass (5231–5236) match the path of the RMS[70–72] (Extended Data Fig. 12h); the trajectory of these astrocyte clusters on the UMAP matches well with the corresponding spatial gradients (Fig. 4b). Our data showed two main neuronal populations arising from RMS into olfactory bulb, clusters that populate the inner granule and mitral cell layers (Extended Data Fig. 12a,c, trajectory c) and clusters that populate the outer glomerular layer (Extended Data Fig. 12a,d, trajectory d). Immature neurons in the SVZ and RMS are marked by the expression of cell cycle-associated genes like *Top2a* and *Mki67* (Extended Data Fig. 12e,f). As the immature OB neurons exit the RMS, they express markers such as *Sox11* and *S100a6*[87], whereas the mature OB neurons are marked by the expression of *Frmd7*. Astrocytes that follow the same trajectory as the immature neurons in the RMS also show changes in gene expression along the trajectory that are similar to the gene expression changes in the IMN population. There are 290 genes that are differentially expressed along the RMS astrocyte

trajectory (Extended Data Fig. 12i), of which 93 genes are also differentially expressed in the OB IMN.

## Transcription factors in defining cell types

Transcription factors are considered key regulators of cell-type identity[88,89]. To evaluate the correspondence of transcription factor expression to transcriptomic cell types, we calculated the number of differentially expressed transcription factors between each pair of neuronal versus non-neuronal classes, classes, subclasses or pairs of clusters within a subclass (Fig. 5a). We then compared cross-validation accuracy of subclass and cluster recalls using classifiers built based on all 8,460 DEGs, 534 transcription factor marker genes, 541 functional genes, genes coding for adhesion molecules, and 534 randomly selected DEGs (Fig. 5b; see Supplementary Table 5 for full lists of marker genes used). The median cluster recall accuracy of cross-validation with transcription factors is between that of all DEGs and the random subset of DEGs. The cross-validation accuracy of subclass recall with transcription factors is 0.94, which is close to the accuracy with all DEGs (0.98), whereas the accuracy using functional genes, 857 or 534 adhesion molecule encoding genes, or the random subset of DEGs is lower (accuracy of 0.90, 0.88, 0.81 or 0.75, respectively). The Pearson correlation of gene expression between a pair of cell types computed using all or a subset of DEGs is a measure of the similarity between the two cell types. We compared the pairwise correlation values among all clusters computed using all 8,460 DEGs with those computed using the adhesion, functional or transcription factor marker gene sets (Extended Data Fig. 3e–g). We found that transcription factor marker genes show the lowest correlations in gene expression among all clusters compared with functional genes, adhesion molecules and all DEGs (Fig. 5c), suggesting that transcription factors have the greatest capability to differentiate cell types. These results quantify the major roles transcription factors can have in defining cell-type identities.

We identified a large set of transcription factor co-expression modules (52 modules) (Methods and Supplementary Table 8) that are selectively expressed in specific groups of cell types at all hierarchical levels and hence may define identities of these groups of cell types (Fig. 5d). A pallium glutamatergic-specific module includes *Tbr1* and *Satb2*, which also show differential expression in different subclasses. Immediate early genes *Egr3* and *Nr4a1* are highly expressed in pallium glutamatergic neurons, whereas *Fos* and *Fosb* have more uniform expression. The bHLH transcription factors including *Neurod1*, *Neurod2*, *Neurod6* and *Bhlhe22* are widely expressed in many types of neurons but have highest expression in pallium glutamatergic cells. The *Dlx1*, *Dlx2*, *Dlx5*, *Dlx6*, *Arx*, *Sp8* and *Sp9* module is specific to GABAergic neurons in telencephalon, whereas the *Gata3*, *Gata2* and *Tal1* module is specific to GABAergic neurons in midbrain and pons. Of note, the latter gene module is best known as master regulator of haematopoietic development[90], and is an example of repurposing the same transcription factor module for specifying cell types in different systems. *Gbx2*, *Shox2* and *Tcf7l2* are highly expressed in thalamus glutamatergic neurons[91,92], whereas *Shox2* and *Tcf7l2* are also expressed in midbrain. Hox genes are specific to medulla GABAergic and glutamatergic neurons, whereas *Pax2* and *Pax8* distinguish medulla GABAergic neurons from medulla glutamatergic neurons. We also identified a transcription factor module for the Astro–Epen cell class, including *Sox9*, *Gli2*, *Gli3* and *Rfx4*, and several distinct modules for other non-neuronal cell subclasses.

For most other modules, each module consisted of a few transcription factors that are homologues—for example, *Nfia*, *Nfib* and *Nfix*, the Zic family, the Irx family, the Ebf family, *En1* and *En2*, *Lhx6* and *Lhx8*, *Six3* and *Six6*, and *Pou4f1*, *Pou4f2* and *Pou4f3*. Some of these homologues are located next to each other on the same chromosome, such as *Dlx1* and *Dlx2*, *Dlx5* and *Dlx6*, *Irx1* and *Irx2*, *Irx3* and *Irx5*, *Zic1* and *Zic4*, *Zic2* and *Zic5*, and *Hoxb2–8*. These homologues are likely located within the same chromatin domains, are regulated by the same enhancers, and

have highly similar expression patterns. Many co-expressed homologues show subtle but interesting distinctions. Consistent with the well-studied roles of Hox genes in regulating anterior–posterior axis in development[93], *Hoxb2* and *Hoxb3* have broader expression than *Hoxb4* and *Hoxb5*, and *Hoxb8* has the most restricted expression pattern in posterior lateral medulla, in the order that is consistent with their locations on the chromosome. Although their loci are not very near to each other, *Nfia*, *Nfib* and *Nfix* regulate cell-type differentiation in many tissues[94–96], function as homo- or heterodimers, and bind to largely common targets[97]. Similar interactions between homologues have been reported for many other transcription factor families, such as Ebf[98] and Irx[99]. Finally, we identified a set of transcription factors such as *Meis1* and *Meis2*, and *Nr2f1* and *Nrf2f2*, that are widely expressed but delineate closely related subclasses and clusters and show local spatial gradients.

Although many transcription factor homologues are co-expressed (Fig. 5d), they can also show distinct expression patterns. We examined the expression patterns of several transcription factor families (Extended Data Fig. 13), including forkhead box (Fox), Krüppel-like factor (Klf), LIM homeobox (Lhx), NKX-homeodomain (Nkx), nuclear receptor (Nr), Paired box (Pax), POU domain (Pou), positive regulatory domain (Prdm), SRY-related HMG-box (Sox), and T-box (Tbx), all of which have been shown to have important roles in spatial patterning, cell-type specification and differentiation during development[100–107]. In each family, only the transcription factor markers identified in this study are included. Members of the same transcription factor family evolved from common ancestors, have strong sequence conservation, and very similar DNA binding motifs. Revealing their distinct cell-type specificity provides deeper insights into the evolution of these transcription factor families.

Particularly intriguing is the LIM homeobox family, which can be split into multiple groups with complementary expression patterns that together cover most neuronal types in the brain. *Lhx2* and *Lhx9* are co-expressed in thalamus and midbrain glutamatergic types, but *Lhx2* is also specifically expressed in the pallium IT–ET types[108,109]. *Lhx6* and *Lhx8* are co-expressed in some CNU and hypothalamus GABAergic types[110,111], but *Lhx6* is also specifically expressed in MGE types. *Lhx1* and *Lhx5* are co-expressed in HY MM, as well as in midbrain and hindbrain cell types where they are more widely expressed in GABAergic than glutamatergic types. *Lmx1a* and *Lmx1b* are co-expressed in hindbrain glutamatergic and midbrain dopaminergic cell types[112], and *Lmx1b* is also specifically expressed in serotonergic types[113,114]. *Lhx3* and *Lhx4* are co-expressed in very specific glutamatergic types in hindbrain and pineal gland. *Isl1* is widely expressed in hypothalamus and CNU, and more highly in GABAergic than glutamatergic cell types[115]. The grouping of *Lhx* members based on their gene expression patterns exactly matches their phylogeny tree based on their coding sequences[106] and aligns with the sub-family definition.

## Brain region-specific cell-type features

The results presented above showed that each cell type has a specific spatial localization. To compare the global spatial distribution patterns of all cell types and the relationship between transcriptomic similarity and spatial proximity, we quantified the brain-wide spatial distribution patterns of all cell subclasses against all mid-ontology level brain regions (Supplementary Table 9) using the CCFv3-registered whole-brain MERFISH dataset (Fig. 6a). The result showed that all neuronal subclasses are restricted to a particular brain region, whereas non-neuronal subclasses are more widely distributed. Transcriptomically more similar cell types are located closer to each other spatially—for example, neuronal subclasses within the same class are mostly colocalized within the same major brain region. Conversely, transcriptomically more distant cell types are spatially further apart from each other. Each major brain region has its own specific sets of

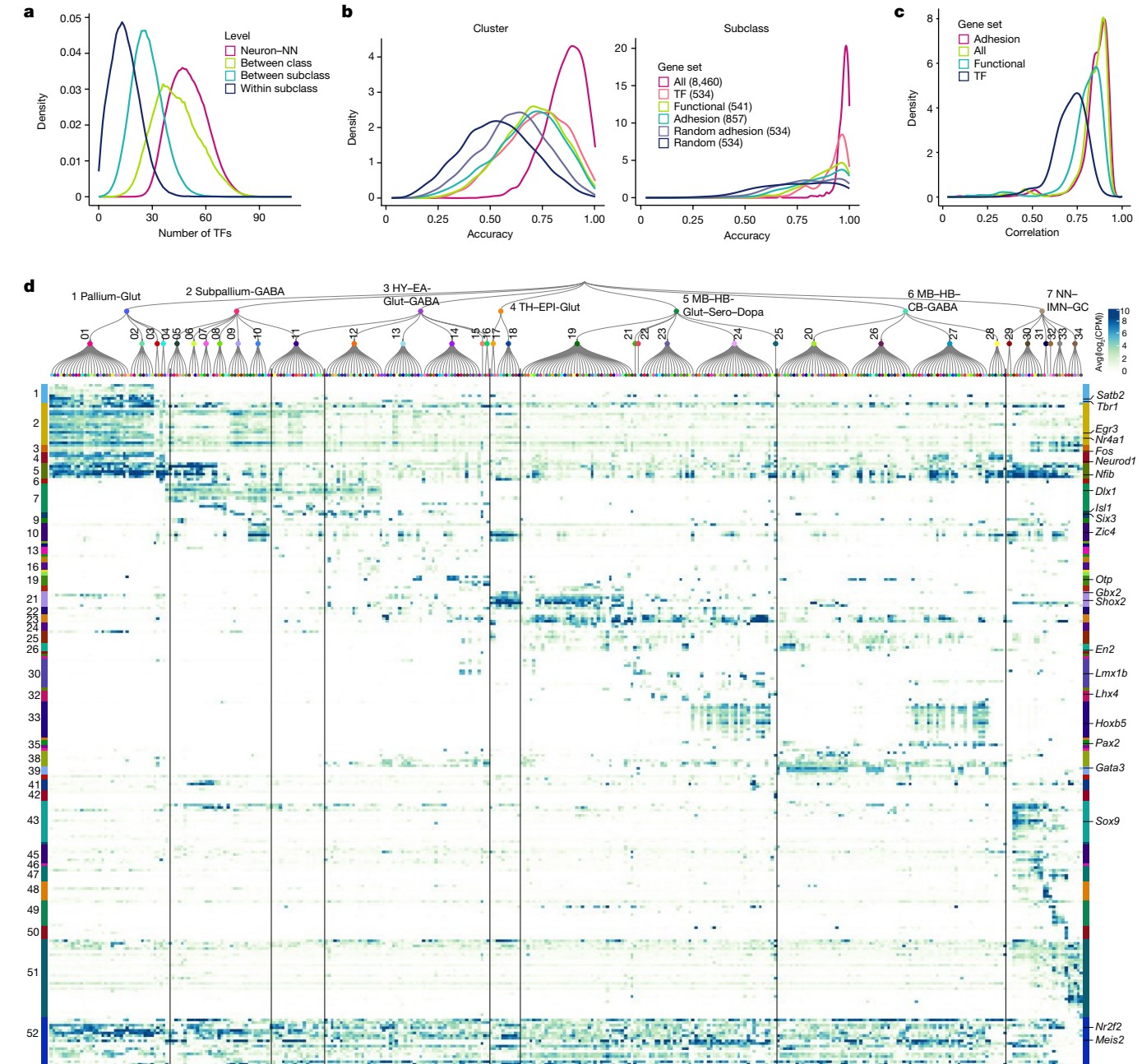

**Fig. 5 | Transcription factor modules across the whole mouse brain.**
**a**, Distribution of the number of differentially expressed transcription factors (TFs) between neuronal and non-neuronal classes, between classes, between subclasses, and within subclasses. **b**, Cross-validation accuracy for each cluster (left) or subclass (right) using classifiers built based on all 8,460 marker genes (all), 534 transcription factor marker genes (TF), 541 functional marker genes, 857 marker genes encoding adhesion molecules (adhesion), 534 randomly selected adhesion marker genes (random adhesion), or 534 randomly selected marker genes (random). **c**, Density plot showing distribution of correlation of

marker gene expression between clusters using all markers, adhesion marker genes, functional genes and transcription factors. Correlation values are derived from full correlation matrices shown in Extended Data Fig. 3. **d**, Expression of key transcription factors for each subclass in the taxonomy tree, organized in transcription factor co-expression modules shown as colour bars on both sides of the heat map. Module IDs are shown on the left, exemplary transcription factor genes are shown on the right. For a full list of transcription factor genes in each module (in the same order as in this heat map), see Supplementary Table 8. Avg, mean.

both glutamatergic and GABAergic neuronal subclasses that are mostly colocalized. Although not illustrated here, such high correspondence between transcriptomic and spatial specificity extends to supertype level. We further used the Gini coefficient and Shannon diversity index to measure the extent of variation in spatial distribution among subclasses (Fig. 6a; also see Extended Data Fig. 14 for Gini coefficient), and both reveal very high inequality (that is, highly localized patterns) in spatial distribution of each neuronal subclass.

We further evaluated the correspondence between transcriptomic identity and spatial specificity by computing their mutual predictability (Methods) using imputed whole-transcriptomic profiles in the MERFISH space (Extended Data Fig. 8). As glutamatergic and GABAergic neurons colocalize in many brain regions, which would confound the space-to-transcriptome prediction, we performed the analysis in two separate groups, one with the GABA classes and the other with Glut, Dopa and Sero classes. We found high predictability

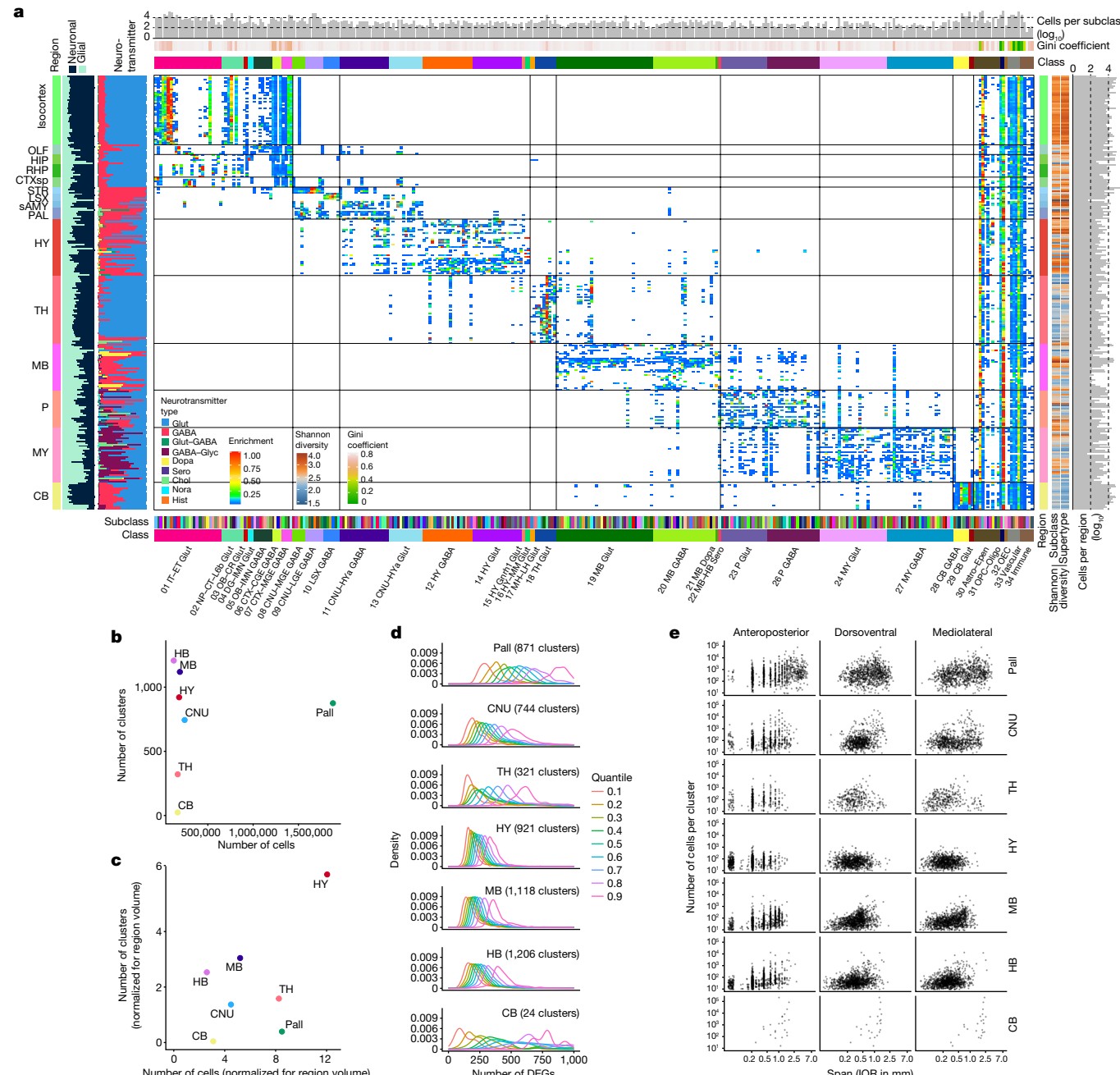

**Fig. 6 | Brain region-specific features of cell types. a**, Heat map showing the CCFv3 region distribution (*y* axis) in each subclass (*x* axis) for MERFISH cells. Bar graphs on the left show the broad CCFv3 regions, proportion of neuronal versus glial cells per region of interest (ROI), and proportion of neurotransmitter types per ROI. Bar graphs on the right show broad CCFv3 regions, Shannon diversity per subclass and supertype, and number of cells per ROI. Bar graphs on the top show number of cells per subclass, Gini coefficient and class assignment. Bar graphs on the bottom show subclass and class annotations. **b**, Scatter plot showing the number of neuronal clusters identified per major brain region versus the number of neuronal cells profiled by scRNA-seq in the corresponding region. Each neuronal cluster is assigned to the most dominant region. **c**, As in **b**, except numbers of clusters and profiled cells are normalized by the region volume. **d**, Distribution of the number of DEGs (identified in scRNA-seq data) between every pair of neuronal clusters within each major brain region, split into indicated quantiles. The curves show the spread of the number of DEGs between more similar types at 0.1 quantile versus the more distinct types at 0.9 quantile. **e**, Scatter plot showing the number of cells mapped to a given neuronal cluster versus the span (as measured by IQR) of their 3D coordinates along the anteroposterior, dorsoventral and mediolateral axes based on the MERFISH dataset, stratified by the major brain regions. Note that both axes are in log scales. The plot shows how localized the clusters are within each region along each spatial axis. IQR, inter-quantile range (the difference between 75% quantile and 25% quantile). Pall, pallium.

from 3D coordinates in CCFv3 for transcriptomic classes and subclasses (Extended Data Fig. 15), with confusions seen only among a few closely related subclasses. Similarly, there was a high degree of predictability from transcriptomic identities to the location of cell types in CCFv3 subregions, with confusions mostly confined to neighbouring subregions (Extended Data Fig. 16). This analysis

indicates that most CCFv3 structures contain distinct neuronal cell types. Notably, the prediction of GABAergic subclasses and their spatial location in both directions appears to have more confusions, especially in the Subpallium-GABA neighbourhood, consistent with the more widespread distribution across multiple cortical areas of many GABAergic cell types[23,26].

We found that the numbers of clusters from different regions do not correlate with the numbers of cells profiled by scRNA-seq even when corrected for brain region volumes (Fig. 6b,c); rather, region-specific characteristics dominate. The hypothalamus, midbrain and hindbrain regions contain the largest numbers of clusters, indicating a high degree of cell-type complexity, consistent with these broad regions having many small and heterogeneous subregions. By contrast, despite orders of magnitude more cells profiled in the pallium owing to the many subregions contained within it (including isocortex, HPF, OLF and CTXsp, each containing multiple subregions) and its overall 4 to 15 times larger volume compared with other major brain structures (Supplementary Table 1), we found an intermediate number of clusters for the entire pallium, similar to the other telencephalic structure, the subpallial CNU. Overall, after volume normalization, pallium, CNU, thalamus and cerebellum contain smaller numbers of clusters, suggesting lower complexity than hypothalamus, midbrain and hindbrain.

We calculated the number of DEGs between each pair of clusters within a brain region, divided the numbers into nine quantiles based on similarities (that is, higher similarity yields fewer number of DEGs) and plotted their distribution by quantiles (Fig. 6d). In regions with larger numbers of clusters—that is, hypothalamus, midbrain and hindbrain—the clusters are more similar to each other within each region, suggesting that cell types in these regions have lower level of transcriptomic differences and are less hierarchical. By contrast, in regions with smaller numbers of clusters—that is, cerebellum, thalamus and pallium—there are large differences in numbers of DEGs between clusters; thus, cell types in these regions appear to be more diverse and hierarchical. CNU exhibits an intermediate level of diversity.

We calculated the 3D spatial span of each cluster based on the MERFISH dataset and aggregated the spans of all clusters within each brain region (Fig. 6e). Again, different regions show differential characteristics, with clusters in pallium, CNU and cerebellum having much larger spans suggesting more widespread distributions, and clusters in hypothalamus, midbrain and hindbrain having much smaller spans suggesting more confined localization. Consistent with this, when quantifying the number of clusters in each subregion, we observed more clusters in individual cortical areas than in many hypothalamus, midbrain and hindbrain nuclei (Extended Data Fig. 17a), suggesting that there are more cell types intermixed in each cortical area than in hypothalamus, midbrain and hindbrain subregions. Furthermore, cluster sizes (tha is, the number of cells in each cluster) also vary among major brain regions (Extended Data Fig. 17b), with hypothalamus, midbrain and hindbrain containing more smaller clusters.

We investigated sex differences in the whole mouse brain transcriptomic cell-type atlas. We identified 26 clusters across 11 subclasses with a skewed distribution of cells derived from the two sexes (Fig. 1a and Supplementary Table 7). Of these, 5 are small, sex-specific clusters: clusters 211, 1299, 2470 and 2472 are male-specific and cluster 2293 is female-specific. The 21 sex-dominant clusters include 1301, 1891, 1895, 1898, 1915, 1916, 2251, 2282, 2290 and 4246, which contain mostly cells from female donors; and clusters 1293, 1304, 1306, 1562, 1685, 1881, 1890, 1913, 2247, 2281 and 4088, which contain mostly cells from male donors. On the basis of the MERFISH data, these clusters mostly reside in specific regions of PAL, sAMY, hypothalamus and hindbrain.

Within the whole mouse brain scRNA-seq dataset, we also collected a complete subset of data covering all brain regions from the dark phase of the circadian cycle (Supplementary Table 2, total 1,121,542 10xv3 cells). All the dark-phase transcriptomes were included in the overall clustering analysis. In all but one subclass, they are found commingled with the corresponding light-phase transcriptomes (the exception being subclass 282, with only 22 cells that are all from the light phase) (Extended Data Fig. 3e and Supplementary Table 7). Out of all 5,322 clusters, there are 335 clusters that do not contain dark-phase cells, whereas none contain dark-phase cells only. Detailed gene expression analysis at class and subclass levels revealed widespread expression differences

of canonical circadian clock genes between the light and dark phases (Extended Data Fig. 18). Across many neuronal and non-neuronal classes and subclasses throughout the brain, nearly all clock genes show consistently higher expression levels in the dark phase than the light phase, except for *Arntl*, which displays an opposite pattern. The 262 Pineal Crx Glut subclass, located in the dorsal part of the third ventricle and on top of superior colliculus (SC) in the MERFISH data, which probably represents the pinealocytes that evolved from photoreceptor cells and secret melatonin[116], has particularly strong circadian gene expression fluctuations (Extended Data Fig. 18b,c). Furthermore, in the 094 SCH Six6 Cdc14a Gaba subclass, which is specific to the suprachiasmatic nucleus (SCH), the circadian pacemaker of the brain, most clock genes (for example, *Per1*, *Per3*, *Dbp*, *Nr1d1* and *Nr1d2*) have higher levels of expression in the light phase than the dark phase, suggesting that the pacemaker cells are at a different phase of the circadian cycle of gene expression from the rest of the brain, consistent with previous findings[117] (Extended Data Fig. 18b,c). Of note, the vascular 329 ABC NN subclass also displays a similar phase shift. These results suggest that our whole mouse brain transcriptomic cell-type atlas also captured circadian state-dependent gene expression changes. Although supervised analysis can reveal these changes, our cell-type classification is not significantly affected by the different circadian states.

## Discussion

In this study, we created a comprehensive, high-resolution transcriptomic cell-type atlas for the whole adult mouse brain based on the combination of whole-brain-scale scRNA-seq and MERFISH datasets. The cell-type atlas was hierarchically organized into four nested levels: 34 classes, 338 subclasses, 1,201 supertypes and 5,322 clusters (Fig. 1). The neuronal cell-type composition in each major brain region was systematically analysed (Fig. 2) and distinct features in different brain regions were identified (Fig. 6). We identified many sets of neuronal types with varying degrees of similarity with each other, including highly distinct neuronal types as well as transitional neuronal types across regions. We also systematically analysed all classes of non-neuronal cell types as well as immature neuronal types and identified their unique spatial distribution and spatial interaction patterns (Fig. 4 and Extended Data Fig. 12). Finally, we characterized cell-type-specific expression of neurotransmitters, neuropeptides and transcription factors (Figs. 3 and 5 and Extended Data Fig. 10) and identified unique characteristics for each, as discussed below. This large-scale study enabled us to delineate several principles regarding cell-type organization across the whole brain. It provides a reference cell-type atlas as a resource for the community that will enable many more discoveries in the future.

One of the most notable findings from our study is the high degree of correspondence between transcriptomic identity and spatial specificity (Figs. 2–4 and 6). Every subclass (and all supertypes and many clusters within each) has a unique and specific spatial localization pattern within the brain. The relative relatedness between transcriptomic types is strongly correlated with the spatial relationship between them (Fig. 6a and Extended Data Figs. 3, 15 and 16). Transcriptomically similar cell types are often found in the same region, or in some cases in related regions that have a common developmental origin. Transitional cell types in the transcriptomic space are also found crossing regional boundaries. The strong correspondence between transcriptomic and spatial specificity and relatedness indicates the importance of anatomical specialization of cell types and lends strong support to the robustness and validity of our transcriptome-derived cell-type classification.

Another notable finding is the distinct features of cell-type organization in different major brain structures (Fig. 6). The anterior and dorsal brain regions, including OLF, isocortex, HPF, STR, thalamus and cerebellum, contain cell classes and types that are highly distinct from the other parts of the brain. Cell types in these regions tend to be more widely distributed, and are often shared between neighbouring regions

or subregions. By contrast, cells from the ventral part of the brain—including ventral PAL, extended amygdala, hypothalamus, midbrain, pons and medulla—form many small clusters that are closely related to each other. These cell types often have restricted spatial localization that likely underlies the small nuclei characteristic of these regions. This dichotomy between the roughly dorsal and ventral parts of the brain may reflect the different evolutionary histories of these brain structures. We hypothesize that the ventral part of the brain mainly carries out the survival function of the organism (such as feeding, reproduction and metabolic homeostasis) and is thus more ancient and subject to more evolutionary constraints; as such, there are many dedicated cell types and circuits in this part of the brain, and they have not changed markedly during evolution. Conversely, the dorsal part of the brain mainly carries out the adaptive function of the organism (such as sensorimotor specialization and cognition), and its structure, function and underlying cell types have expanded and diversified more rapidly during evolution.

While neuronal types constitute the vast majority of cell types in the brain and exhibit high regional specificity, non-neuronal cell types are generally more widely distributed, except for astrocytes and ependymal cells, which have multiple subclasses with regional specificity. However, at the cluster level, we also observed a great degree of spatial specificity in non-neuronal cell types, especially for astrocytes, ependymal cells, tanycytes and VLMCs, indicating specific neuron–glia and glia–vasculature interactions (Fig. 4). We also identified several groups of immature neuronal types and could infer their trajectories to mature neuronal types in olfactory bulb and dentate gyrus on the basis of their spatial localization and transitioning gene signatures (Extended Data Fig. 12).

We examined neurotransmitter and neuropeptide expression in cell types across the brain. We found a diverse set of neuronal clusters with glutamate–GABA co-transmission from many brain regions (Extended Data Fig. 9). We identified all cell types expressing different modulatory neurotransmitters and found that they often co-release glutamate and/or GABA. The neuromodulatory cell types often have closely related glutamatergic and/or GABAergic clusters within the same subclass, showing a high degree of heterogeneity in neurotransmitter content in these cell populations (Fig. 3). Of note, our assignment of neurotransmitter types based on synthesizing enzymes and transporter genes is conservative; there may be even more diversity in neurotransmitter co-release patterns if other unconventional transmitter release routes are considered[57,118]. Similarly, there is a wide spectrum of expression patterns among the different neuropeptide genes—some are widely expressed in many cell types, whereas others are highly specific to one or few cell types (Extended Data Fig. 10). Furthermore, there are numerous co-expression combinations of two or more neuropeptide genes in many neuronal clusters (Supplementary Table 7). These results support the extraordinary diversity in intercellular communications in the brain.

Transcription factors are known to have major roles in patterning brain regions, defining neural progenitor domains and specifying cell-type identities during development. Here we found that in the adult brain, transcription factors also are major determinants in defining cell types across all regions of the brain. Comparison of gene expression correlation matrices among all pairs of clusters showed that transcription factors have the greatest overall power to distinguish cell types (Fig. 5 and Extended Data Fig. 3). We identified transcription factor genes and co-expression modules that are specific at different hierarchical levels (Fig. 5d, Extended Data Fig. 13 and Supplementary Table 8). We observed several different modes of coordination among transcription factors. The first mode is the coordinated expression of different transcription factors (often pairs of transcription factors) within the same transcription factor gene family in specific cell types. The second is the combination of transcription factors at different hierarchical branch levels to collectively define the identity of the leaf-node cell types. The third represents the intersection between different sets of transcription factors that define molecular identity or spatial specificity, respectively, within a cell type. These findings reveal how transcription factors form a combinatorial code that lays out the highly complex cell-type landscape.

The above findings of the high correspondence between transcriptomic identities and spatial distribution patterns of cell types and the prominent roles of transcription factors in defining both transcriptomic and spatial specificity paint a unified picture of the brain architecture—that is, different anatomical regions contain highly diverse sets of cell types that are defined by a master plan of transcription factors. Prior knowledge informed us that the transcription factor master plan is played out during development to generate cell types and brain regions in a stepwise manner. Therefore, studying cell types in the developing brain will be extremely informative for gaining a mechanistic understanding of the formation of the brain architecture (from which brain functions emerge). This understanding will further enable the refinement and revision of existing anatomical ontologies, which are based on the current limited knowledge about brain development at cellular level, and lead to new cell-type-based brain atlases that integrate developmental and adult brain circuit information.

Consistent with the principle of hierarchical cell-type organization identified in previous studies[2], here we have defined a hierarchical taxonomy of cell types across the entire mouse brain, with four levels of classification: class, subclass, supertype and cluster (or type). This classification scheme is analogous to the Linnaean classification system of species and continues to be a useful general framework for defining cell types, since—like species—cell types are the product of evolution[2], evolving from singular to multiple with the genetic linkages to their ancestors and siblings stored in their genomes, epigenomes and transcriptomes. At the same time, the transcriptomic profile of each cell is multidimensional, containing not only information about the cell-type identity, but also information about many other aspects of cellular properties such as connectivity, function or a particular cell state. Therefore, the transcriptomic relationships between cell types are both hierarchical and multidimensional.

We also emphasize the great technical challenges we encountered when analysing these large and highly complex datasets and the two main caveats for the results presented here. First, owing to the difficulty in dissociating and isolating intact cells from the adult brain tissue—especially in highly myelinated areas—our scRNA-seq dataset contained many types of low-quality cells, including damaged cells and mixed debris of various cell-type combinations. These low-quality transcriptomes could be mistaken as real cell types, part of a cell-type continuum, or transitional cell types. They could also drive substantial wrong mapping of MERFISH cells, as we discovered in our analysis. To generate a high-quality transcriptomic cell-type atlas with precise spatial annotation, we developed a set of QC metrics that are more stringent than those widely used in the field, and therefore we disqualified a high proportion of cells from our scRNA-seq dataset (Extended Data Fig. 1). During this process, it is likely that some cell types were more selectively depleted than others, especially large neurons that are more vulnerable to damage during tissue dissociation—such as Purkinje cells and large motor neurons in the midbrain and hindbrain. Thus, cell types in the midbrain and hindbrain may not be fully represented or fully resolved in our atlas. To compensate for this, we performed more refined clustering on the particularly messy midbrain and hindbrain cell types present in our scRNA-seq data and further supplemented it with a small set of high-quality snRNA-seq transcriptomes that we generated separately to make our cell-type taxonomy more complete in this part of the brain.

Second, although we used only the selected high-quality single-cell transcriptomes to construct the cell atlas, the relationships between the large number of cell types across the entire brain are still highly complex and cannot be fully captured by a one-dimensional hierarchical tree or two-dimensional UMAPs. We conducted extensive

iterative clustering to try to resolve all dimensions of variation at the cluster level. Thus, not every cluster may represent a true cell type; our categorization scheme may also need to be revised in the future with better computational methods and/or more experimental evidence (especially developmental data). Lastly, owing to the sheer scale of the atlas, we have not extensively incorporated the vast amount of existing data and knowledge about cell types in many parts of the brain to help better annotate the cell-type atlas. Moving forward, it will be critical to engage the neuroscience community to collectively annotate, refine and enhance this whole mouse brain cell-type atlas, and we hope that the online platform we have provided will facilitate this effort.

In conclusion, the transcriptomic and spatial cell-type atlas of the whole mouse brain establishes a foundation for deep and integrative investigations of cellular and circuit function, development and evolution of the brain, akin to the reference genomes for studying gene function and genomic evolution. The atlas provides baseline gene expression patterns that enable investigation of the dynamic changes in gene expression and cellular function in different physiological and diseased conditions. It enables the creation of cell-type-targeting tools for labelling and manipulating specific cell types to probe and modify their functions in vivo. The atlas provides a foundational framework for organizing and integrating the vast knowledge about the brain structure and function, facilitating the extraction of new principles from the extraordinarily complex cell-type and circuit landscape. It provides a guidepost for generating similarly comprehensive and detailed cell-type atlases for other species as well as across developmental times, facilitating cross-species comparative studies and gaining mechanistic insights into the genesis of cell types and circuits in the mammalian brain. Understanding the conservation and divergence of cell types between human and model organisms will have profound implications for the study of human brain function and diseases.

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

Zizhen Yao[1]✉, Cindy T. J. van Velthoven[1], Michael Kunst[1], Meng Zhang[2], Delissa McMillen[1], Changkyu Lee[1], Won Jung[2], Jeff Goldy[1], Aliya Abdelhak[1], Matthew Aitken[1], Katherine Baker[1], Pamela Baker[1], Eliza Barkan[1], Darren Bertagnolli[1], Ashwin Bhandiwad[1], Cameron Bielstein[1], Prajal Bishwakarma[1], Jazmin Campos[1], Daniel Carey[1], Tamara Casper[1], Anish Bhaswanth Chakka[1], Rushil Chakrabarty[1], Sakshi Chavan[1], Min Chen[3], Michael Clark[1], Jennie Close[1], Kirsten Crichton[1], Scott Daniel[1], Peter DiValentin[1], Tim Dolbeare[1], Lauren Ellingwood[1], Elysha Fiabane[1], Timothy Fliss[1], James Gee[3], James Gerstenberger[1], Alexandra Glandon[1], Jessica Gloe[1], Joshua Gould[4], James Gray[1], Nathan Guilford[1], Junitta Guzman[1], Daniel Hirschstein[1], Windy Ho[1], Marcus Hooper[1], Mike Huang[1], Madie Hupp[1], Kelly Jin[1], Matthew Kroll[1], Kanan Lathia[1], Arielle Leon[1], Su Li[1], Brian Long[1], Zach Madigan[1], Jessica Malloy[1], Jocelin Malone[1], Zoe Maltzer[1], Naomi Martin[1], Rachel McCue[1], Ryan McGinty[1], Nicholas Mei[1], Jose Melchor[1], Emma Meyerdierks[1], Tyler Mollenkopf[1], Skyler Moonsman[1], Thuc Nghi Nguyen[1], Sven Otto[1], Trangthanh Pham[1], Christine Rimorin[1], Augustin Ruiz[1], Raymond Sanchez[1], Lane Sawyer[1], Nadiya Shapovalova[1], Noah Shepard[1], Cliff Slaughterbeck[1], Josef Sulc[1], Michael Tieu[1], Amy Torkelson[1], Herman Tung[1], Nasmil Valera Cuevas[1], Shane Vance[1], Katherine Wadhwani[1], Katelyn Ward[1], Boaz Levi[1], Colin Farrell[1], Rob Young[1], Brian Staats[1], Ming-Qiang Michael Wang[1], Carol L. Thompson[1], Shoaib Mufti[1], Chelsea M. Pagan[1], Lauren Kruse[1], Nick Dee[1], Susan M. Sunkin[1], Luke Esposito[1], Michael J. Hawrylycz[1], Jack Waters[1], Lydia Ng[1], Kimberly Smith[1], Bosiljka Tasic[1], Xiaowei Zhuang[2] & Hongkui Zeng[1]✉

[1]Allen Institute for Brain Science, Seattle, WA, USA. [2]Howard Hughes Medical Institute, Department of Chemistry and Chemical Biology, Department of Physics, Harvard University, Cambridge, MA, USA. [3]University of Pennsylvania, Philadelphia, PA, USA. [4]Genentech, San Francisco, CA, USA. ✉e-mail: zizheny@alleninstitute.org; hongkuiz@alleninstitute.org

# Methods

## Mouse breeding and husbandry

All experimental procedures related to the use of mice were approved by the Institutional Animal Care and Use Committee of the AIBS, in accordance with NIH guidelines. Mice were housed in a room with temperature (21–22 °C) and humidity (40–51%) control within the vivarium of the AIBS at no more than five adult animals of the same sex per cage. Mice were provided food and water ad libitum and were maintained on a regular 14:10 h light:dark cycle, or on a reversed 12:12 h light:dark cycle. Mice were maintained on the C57BL/6J background. We excluded any mice with anophthalmia or microphthalmia.

We used 95 mice (41 female, 54 male) to collect 2,492,084 cells for 10xv2 and 222 mice (112 female, 110 male) to collect 4,466,283 cells for 10xv3. Animals were euthanized at postnatal day (P)53–59 ($n = 141$), P50–52 ($n = 3$), or P60–71 ($n = 173$). No statistical methods were used to predetermine sample size. All donor animals used for scRNA-seq data generation are listed in Supplementary Table 2.

Transgenic driver lines were used for fluorescence-positive cell isolation by fluorescence-activated cell sorting (FACS) to enrich for neurons. Most cells were isolated from the pan-neuronal *Snap25-IRES2-cre* line (RRID:IMSR_JAX:023525) crossed to the *Ai14*-tdTomato reporter[119,120] (RRID:IMSR_JAX:007914) (279 out of 317 donors, Supplementary Table 2). A small number of *Gad2-IRES-cre/wt;Ai14/wt* (6 donors) and *Slc32a1-IRES-cre/wt;Ai14/wt* mice (4 donors) (*Gad2-IRES-cre*: RRID:IMSR_JAX:028867; *Slc32a1-IRES-cre*: RRID:IMSR_JAX:028862) were used for fluorescence-positive cell isolation to enrich for the sampling of GABAergic neurons in hippocampal region, OLF and cerebellum. For unbiased sampling without FACS, we used either *Snap25-IRES2-cre/wt;Ai14/wt or Ai14/wt* mice.

The number of mice contributing to each cluster varies between 2 and 266, with an average of 19 and median of 14. There are 23 clusters that have fewer than 4 donor animals each. Thus, individual mouse variability should not affect cell-type identities (Extended Data Fig. 5).

For cell collection during the dark phase of the circadian cycle, mice were randomly assigned to circadian time groups at time of weaning and housed on the reversed 12:12 h light:dark cycle. Brain dissections for all groups took place in the morning. From 267 donors, 5,836,825 cells were collected during the light phase of the light:dark cycle. For 50 donors, 1,121,542 cells across the whole brain were collected during the dark phase of the light:dark cycle (Supplementary Table 2).

## scRNA-seq

**Single-cell isolation.** We used CCFv3 (RRID: SCR_002978) ontology[22] (http://atlas.brain-map.org/, Supplementary Table 1) to define brain regions for profiling and boundaries for dissection. We covered all regions of the brain using sampling at top-ontology level with judicious joining of neighbouring regions (Extended Data Fig. 1d,e and Supplementary Table 3). These choices were guided by the fact that microdissections of small regions were difficult. Therefore, joint dissection of neighbouring regions was sometimes necessary to obtain sufficient numbers of cells for profiling. Comparison with subsequently generated MERFISH data showed that our CCFv3-based microdissections were largely accurate at cell subclass and major brain region levels (Extended Data Fig. 2h).

Single cells were isolated following a cell-isolation protocol developed at AIBS[23,121]. The brain was dissected, submerged in artificial cerebrospinal fluid (ACSF), embedded in 2% agarose, and sliced into 350-μm coronal sections on a compresstome (Precisionary Instruments). Block-face images were captured during slicing. ROIs were then microdissected from the slices and dissociated into single cells as previously described[23]. Fluorescent images of each slice before and after ROI dissection were taken at the dissection microscope. These images were used to document the precise location of the ROIs using annotated coronal plates of CCFv3 as reference.

Dissected tissue pieces were digested with 30 U ml⁻¹ papain (Worthington PAP2) in ACSF for 30 min at 30 °C. Due to the short incubation period in a dry oven, we set the oven temperature to 35 °C to compensate for the indirect heat exchange, with a target solution temperature of 30 °C. Enzymatic digestion was quenched by exchanging the papain solution three times with quenching buffer (ACSF with 1% FBS and 0.2% BSA). Samples were incubated on ice for 5 min before trituration. The tissue pieces in the quenching buffer were triturated through a fire-polished pipette with 600-μm diameter opening approximately 20 times. The tissue pieces were allowed to settle and the supernatant, which now contained suspended single cells, was transferred to a new tube. Fresh quenching buffer was added to the settled tissue pieces, and trituration and supernatant transfer were repeated using 300-μm and 150-μm fire-polished pipettes. The single-cell suspension was passed through a 70-μm filter into a 15-ml conical tube with 500 μl of high-BSA buffer (ACSF with 1% FBS and 1% BSA) at the bottom to help cushion the cells during centrifugation at 100*g* in a swinging-bucket centrifuge for 10 min. The supernatant was discarded, and the cell pellet was resuspended in the quenching buffer. We collected 1,508,284 cells without performing FACS. The concentration of the resuspended cells was quantified, and cells were immediately loaded onto the 10x Genomics Chromium controller.

To enrich for neurons or live cells, cells were collected by fluorescence-activated cell sorting (FACS, BD Aria II running FACSdiva v8) using a 130-μm nozzle, following a FACS protocol developed at AIBS[122]. Cells were prepared for sorting by passing the suspension through a 70-μm filter and adding Hoechst or DAPI (to a final concentration of 2 ng ml⁻¹). Sorting strategy with example images[122] was as previously described[23], and most cells were collected using the tdTomato-positive label. Around 30,000 cells were sorted within 10 min into a tube containing 500 μl of quenching buffer. We found that sorting more cells into one tube diluted the ACSF in the collection buffer, causing cell death. We also observed decreased cell viability for longer sorts. Each aliquot of sorted 30,000 cells was gently layered on top of 200 μl of high-BSA buffer and immediately centrifuged at 230*g* for 10 min in a centrifuge with a swinging-bucket rotor (the high-BSA buffer at the bottom of the tube slows down the cells as they reach the bottom, minimizing cell death). No pellet could be seen with this small number of cells, so we removed the supernatant and left behind 35 μl of buffer, in which we resuspended the cells. Immediate centrifugation and resuspension allowed the cells to be temporarily stored in a high-BSA buffer with minimal ACSF dilution. The resuspended cells were stored at 4 °C until all samples were collected, usually within 30 min. Samples from the same ROI were pooled, cell concentration quantified, and immediately loaded onto the 10x Genomics Chromium controller.

**Single-nucleus isolation.** Some neuronal types are difficult to isolate using a cell-isolation procedure. We collected additional single-nucleus 10x Multiome data in midbrain and hindbrain regions to supplement cell types lost due to technical limitations.

Mice were anaesthetized with 2.5–3% isoflurane and transcardially perfused with cold, pH 7.4 HEPES buffer containing 110 mM NaCl, 10 mM HEPES, 25 mM glucose, 75 mM sucrose, 7.5 mM MgCl₂, and 2.5 mM KCl to remove blood from brain[123]. Following perfusion, the brain was dissected quickly, frozen for 2 min in liquid nitrogen vapour and then moved to −80 °C for long term storage following a freezing protocol developed at AIBS[124].

Frozen mouse brains were sectioned using a cryostat with the cryochamber temperature set at −20 °C and the object temperature set at −22 °C. Brains were securely mounted by the cerebellum or by the olfactory region onto cryostat chucks using OCT (Sakura FineTek 4583). Tissue was trimmed using a thickness of 20–50 μm and once at the desired location slices with thickness of 300 μm were generated to dissect out ROI(s) following reference atlas. Images were taken while leaving the dissection in the cutout section. Nuclei were isolated

using the RAISINs method[125] with a few modifications as described in a nuclei isolation protocol developed at AIBS[126]. In short, excised tissue dissectates were transferred to a 12-well plate containing CST extraction buffer. Mechanical dissociation was performed by chopping the dissectate using spring scissors in ice-cold CST buffer for 10 min. The entire volume of the well was then transferred to a 50-ml conical tube while passing through a 100-μm filter and the walls of the tube were washed using ST buffer. Next the suspension was gently transferred to a 15-ml conical tube and centrifuged in a swinging-bucket centrifuge for 5 min at 500 rcf and 4 °C. Following centrifugation, the majority of supernatant was discarded, pellets were resuspended in 100 μl 0.1× lysis buffer and incubated for 2 min on ice. Following addition of 1 ml wash buffer, samples were gently filtered using a 20-μm filter and centrifuged as before. After centrifugation most of the supernatant was discarded, pellets were resuspended in 10 μl chilled nuclei buffer and nuclei were counted to determine the concentration. Nuclei were diluted to a concentration targeting 5,000 nuclei per μl.

**cDNA amplification and library construction.** For 10xv2 processing, we used Chromium Single Cell 3′ Reagent Kit v2 (120237, 10x Genomics). We followed the manufacturer's instructions for cell capture, barcoding, reverse transcription, cDNA amplification, and library construction[127]. We loaded 11,870 ± 4,146 (mean ± s.d.) cells per port. We targeted sequencing depth of 60,000 reads per cell; the actual average achieved was 54,379 ± 34,845 (mean ± s.d.) reads per cell across 299 libraries.

For 10xv3 processing, we used the Chromium Single Cell 3′ Reagent Kit v3 (1000075, 10x Genomics). We followed the manufacturer's instructions for cell capture, barcoding, reverse transcription, cDNA amplification and library construction[128]. We loaded 13,404 ± 2,798 cells per port. We targeted a sequencing depth of 120,000 reads per cell; the actual average achieved was 83,190 ± 85,142 reads per cell across 482 libraries.

For 10x Multiome processing, we used the Chromium Next GEM Single Cell Multiome ATAC + Gene Expression Reagent Bundle (1000283, 10x Genomics). We followed the manufacturer's instructions for transposition, nucleus capture, barcoding, reverse transcription, cDNA amplification and library construction[129]. For the snRNA-seq libraries, we loaded 16,007 ± 692 nuclei per port and targeted a sequencing depth of 120,000 reads per nucleus. The actual average achieved, for the nuclei included in this study, was 157,023 ± 68,484 reads per nucleus across 1,687 nuclei.

**Sequencing data processing and QC.** Processing of 10x Genomics scRNA-seq libraries was performed as described previously[23]. In brief, libraries were sequenced on the Illumina NovaSeq6000, and sequencing reads were aligned to the mouse reference transcriptome (M21, GRCm38.p6) using the 10x Genomics CellRanger pipeline (version 6.1.1) with default parameters.

10x Genomics Multiome (10xMulti) libraries were sequenced on the Illumina NovaSeq6000, and sequencing reads were aligned to the mouse references downloaded from 10x Genomics, which includes ensembl GRCm38 (v98) fasta and gencode (vM23) gtf file, using the 10x Genomics CellRanger Arc (v2.0) workflow with default parameters.

To remove low-quality cells, we developed a stringent QC process. Cells were first classified into broad cell classes after mapping to an existing, preliminary version of taxonomy, and cell quality was assessed based on gene detection, QC score, and doublet score. The QC score was calculated by summing the log-transformed expression of a set of genes whose expression level is decreased significantly in poor quality cells. These are housekeeping genes that are strongly expressed in nearly all cells with a very tight co-expression pattern that is anti-correlated with the nucleus localized gene *Malat1* (Supplementary Table 4). Out of the 62 such genes chosen, 30 are annotated as mitochondrial inner membrane category based on GO ontology cellular component, although they are not located on the mitochondrial chromosome. Some evidence suggests the mRNAs of some of these genes or their homologues are translocated to the mitochondrial surface[130,131]. We used this QC score to quantify the integrity of cytoplasmic mRNA content, which tended to show bimodal distribution. Cells at the low end were very similar to single nuclei, which we removed for downstream analysis. Doublets were identified using a modified version of the DoubletFinder algorithm[132] (available in scratch.hicat, https://github.com/AllenInstitute/scratch.hicat, v1.0.9) and removed when doublet score >0.3. Using QC score and gene-count thresholds that were tailored to different cell classes, we filtered out 43% and 29% of cells and kept 2,546,319 cells and 1,769,304 cells for 10xv3 and 10xv2 data, respectively (Extended Data Fig. 1). Threshold parameters and number of cells filtered are summarized in Supplementary Table 4. For example, for neurons (excluding granule cells) we used gene counts cutoff of 2,000 and QC score cutoff of 200.

We adopted a similar strategy to filter low-quality nuclei for the 10xMulti snRNA-seq dataset. Nuclei were first classified into broad cell classes after mapping to an existing, preliminary version of taxonomy, and cell quality was assessed based on gene detection, QC score, and doublet score. For 10xMulti snRNA-seq dataset, although the overall gene counts were lower compared to 10xv3 scRNA-seq dataset, they showed stronger bimodal distribution of QC metrics, so we could afford to keep the high cutoffs. For neurons (excluding granule cells), we applied the gene-count cutoff of 2,000, and QC score cutoff of 100.

**Clustering scRNA-seq data.** Clustering for both 10xv2 and 10xv3 datasets was performed independently using the in-house developed R package scratch.bigcat (v0.0.5, available via github https://github.com/AllenInstitute/scratch.bigcat), which is a scaled-up version of R package scratch.hicat[23,26] to deal with the increased size of datasets. Scratch.bigcat adopted the parquet file format for storing sparse matrix, which allows for manipulation of matrices that are too large to fit in memory through memory mapping to files on disk. The whole gene-count matrices were chunked to smaller parquet files with bin size of 50,000 for cells, and 500 for genes, which could be loaded efficiently and concurrently using the arrow package (v12.0.1, https://github.com/apache/arrow/, https://arrow.apache.org/docs/r/).

We provide utility functions to convert and concatenate sparse matrices in R to this format, and functions for conversion between this format and other commonly used file formats such as h5, h5ad and Zarr. We also provide a function that loads any sub-matrix into the memory given the cell IDs and gene IDs. The choice of parquet format is based on its great performance in R, which allows continual usage of our legacy codebase. The major functions of scratch.hicat package were rewritten and made available in scratch.bigcat. We used the automatic iterative clustering method, iter_clust_big, which performed clustering in top down manner into cell types of increasingly finer resolution without any human intervention, while ensuring that all pairs of clusters, even at the finest level, were separable by stringent differential gene expression criteria as follows: for 10v2, q1.th = 0.4, q.diff.th = 0.7, de.score.th = 150, min.cells = 10; for 10xv3, q1.th = 0.5, q.diff.th = 0.7, de.score.th = 150, min.cells = 4. These criteria translated to at least 8 binary DEGs between any pair of clusters (each DEG's contribution to de.score was capped at 20, so at least 8 genes were needed to exceed de.score.th of 150). Binary DEGs were defined as genes expressed in at least 40% cells in the foreground cluster in 10xv2, and 50% in 10xv3 (q1.th parameter), $|\log_2(FC)| > 1$, adj Pval <0.01, and difference between the fraction of cells expressing the gene in foreground and background divided by the foreground fraction was greater than 0.7 (q.diff.th parameter).

To enhance scalability, a randomly subsampled set of cells to be clustered were loaded into memory to compute high variance genes and perform principal component analysis (PCA), then projected to all the cells to obtain their reduced dimensions. Then Jaccard–Leiden clustering proceeded as before[23].

10xMulti snRNA-seq datasets were clustered using the same pipeline, using more relaxed threshold: q1.th = 0.4, de.score.th = 130, min.cells = 10.

**Differential gene expression analysis.** We performed differential gene expression both at the clustering step for each iteration, and after clustering between all pairs of clusters. In our original scrattch.hicat package, we applied limma package[133] to perform this analysis. Given the significant increase of data size and complexities of the taxonomy, we re-implemented this method that provides essentially identical results, but drastically improves performance and scalability. The method first scanned the whole log-transformed cell-by-gene matrix once to compute, for each cluster and each gene, the average expression, the fraction of cells expressing the gene, and the sum of square of gene expression of all the cells within the cluster. These cluster-level summary statistics were then used in the linear model equivalent to the one used in limma to compute the pvalue, adjusted pvalue, log fold change, and the contrast between foreground and background based on the fraction of cells expressing the gene. This process was massively parallelized. Clusters were grouped into bins, and the DEG analysis results were stored on disk in chunked parquet files, split based on which bin the foreground and background clusters belonged to. In this way, we were able to compute DEGs between ~13.5 million pairs of clusters within a day on a single Linux server. Using the arrow package, we were able to query DEGs between any pairs of clusters very efficiently.

**Excluding noise clusters.** Before proceeding with integration between 10xv2, 10xv3, and 10xMulti datasets, we first needed to remove noise clusters. The presence of such clusters can confuse the integration algorithm and reduce the cell-type resolution. There are two main categories of noise clusters: clusters with significantly lower gene detection due to extensive drop out, and clusters due to doublets or contamination.

We first identified doublet clusters based on the co-expression of any pair of broad class marker genes using find_doublet_by_marker function in scrattch.bigcat package. To identify other doublet clusters, we searched for triplets of clusters A, B and C, wherein A was the putative doublet cluster, such that up-regulated genes of A relative to B largely overlapped with up-regulated genes in C relative to B, and up-regulated genes in A relative to C largely overlapped with up-regulated genes of B relative to C. This criterion ensured that A included the most distinguished signature of B and C. To rule out the possibility that A was a transitional type between B and C, we required that B and C could not be closely related types based on the correlation of their average gene expression of marker genes. After we systematically produced the list of all the candidate triplet clusters, the final determination was an iterative process that involved setting different thresholds and manual inspection of borderline cases.

After removing all doublet clusters, we then identified clusters with lower gene detection. To do that, we identified pairs of clusters such that one cluster with at least 50% fewer UMIs or >100 lower QC score, smaller size, and no more than one up-regulated gene relative to another cluster was identified as the low-quality cluster. In these cases, one cluster was a degraded version of another cluster and therefore removed.

We identified 933 noise clusters with 153,598 cells in 10xv3, and 201 noise clusters with 38,073 cells in 10xv2. 10xv3 noise clusters were removed from integration analysis but 10xv2 noise clusters were included accidentally. Fortunately, most of the cells from 10xv2 noise clusters were excluded in further QC steps after integration.

**Joint clustering 10xv2 and 10xv3 datasets.** To provide one consensus cell-type taxonomy based on both 10xv2 and 10xv3 datasets of ~2 M cells each, we scaled up the integrative clustering method[23] and made it available via scrattch.bigcat package which extends the clustering pipeline described above to integrate datasets collected by different transcriptomic platforms. Analysis was performed as described before[23] with minor modifications. To build the common graph that incorporates samples from all the datasets, both 10xv2 and 10xv3 were used as the reference datasets. The key steps in the pipeline are: (1) select anchor cells for each reference dataset; (2) select high variance genes in each reference dataset, prioritizing shared high variance genes; (3) compute KNN both within modality and cross modality; (4) compute Jaccard similarity based on shared neighbours; (5) perform Leiden clustering based on Jaccard similarity; (6) merge clusters based on total number and significance of conserved DEGs across modality between similar cell types; (7) repeat steps 1–6 for cells within a cluster to gain finer-resolution clusters until no clusters can be found; (8) concatenate all the clusters from all the iterative clustering steps and perform final merging as in step 6. For step 6, if one cluster had fewer than the minimal number of cells in a dataset (4 cells for 10xv3 and 10 cells for 10xv2), then this dataset was not used for differentially expressed gene computation for all pairs involving the given cluster. This step allows detection of unique clusters only present in some data types.

Compared to the previous version, the key improvement is step 3 for computing KNN. We used BiocNeighbor package (v1.16.0, https://github.com/LTLA/BiocNeighbors) for computing KNN using Euclidean distance within modality and cosine distance across modality using the Annoy algorithm (v1.17.1, https://github.com/spotify/annoy). The Annoy index was built based on anchor cells for the reference dataset, and KNNs were computed in parallel for all the query cells. Due to significantly increased dataset sizes, the Jaccard similarity graph can be extremely large, impossible to fit in memory. The method down-sampled the datasets based on a user specified parameter, and if the cluster membership of each modality was provided as input for integration algorithm, we down-sampled cells by within-modality clusters, ensuring preservation of rare cell types. All the anchor cells were added to the down-sampled datasets. The Jaccard–Leiden clustering was performed on the down-sampled datasets, and the cluster membership of other cells were imputed based on KNNs computed in step 3.

The integration algorithm generated 5,283 clusters, which were used to build cell-type taxonomy. During this process, additional noise clusters were identified by manual inspection, which exhibited abnormal QC statistics, abnormal expression of canonical markers, or absence in MERFISH dataset. Most of these clusters were very small, likely doublets of damaged cells. After removing these additional noise clusters, we had 5,200 clusters with 4,041,289 cells.

After careful examination of spatial distribution of each cell types based on the MERFISH dataset, we realized that the some of the existing clusters in Astro–Epen class did not fully capture the rich spatial gradients present in the dataset. We also identified some hindbrain neuronal clusters that had high within-cluster heterogeneity. Several factors potentially contribute to presence of these heterogeneous clusters. First, sampling of these cell types was not comprehensive enough. They are likely very rare and, given the large fraction of non-neuronal cells present in these areas along with high level of myelination, these cells can be very difficult to collect. Second, some cell types in hindbrain are particularly vulnerable to tissue dissociation, making them even more difficult to profile, and the cells that survive tend to leak more transcripts. This is what we observed in Purkinje neuron population, which was very small in our dataset. After initial clustering, we identified a pair of Purkinje neuron clusters, with high or low gene count, respectively. The low gene-count cluster was discarded subsequently as a low-quality cluster in the post-processing pipeline. Finally, transcriptomic differences in hindbrain neuronal types appear to be subtler compared to neuronal types in other brain regions. These subtler differences make the hindbrain neuronal types more difficult to categorize. To address the problems stated above, we re-clustered cells from the Astro–Epen class and the few highly heterogeneous hindbrain neuronal clusters with more relaxed threshold: de.score.th = 80, min.cells = 8. The resulting more refined clusters were better mapped to the MERFISH

dataset, with distinct spatial distribution and more distinct expression of marker genes.

**Supplementing scRNA-seq taxonomy with Multiome snRNA-seq clusters.** Through mapping our newly generated 10xMulti snRNA-seq data with the scRNA-seq taxonomy (see 'Assigning cell-type identities'), we identified several 10xMulti clusters that have very few mapped scRNA-seq cells with clear transcriptional signature. The missing of these clusters in our scRNA-seq taxonomy led to poor mapping and 'holes' in our MERFISH cell-type annotation. We therefore added these and their neighbouring 10xMulti clusters to the existing taxonomy to enhance cell-type resolution of these depleted populations in the scRNA-seq datasets. Considering the overall lower cell-type resolution of the 10xMulti snRNA-seq dataset due to smaller number of cells and lower gene detection compared to the scRNA-seq datasets, we did not proceed with full-scale integration of the entire 10xMulti snRNA-seq dataset which could compromise the many high-resolution clusters already present in our scRNA-seq taxonomy. As such, our final cell-type taxonomy consists of 5,291 scRNA-seq dominated clusters with a total of 4,041,289 cells and 31 10xMulti snRNA-seq dominated clusters with a total of 1,687 nuclei.

**Marker gene selection.** For each pair of clusters, we computed conserved DEGs (at least significant in one dataset, and at least twofold change in the same direction in the other datasets). We selected the top 20 DEGs in each direction and pooled such genes from all pairwise comparisons to generate a total of 8,460 gene markers (Supplementary Table 5).

**MERFISH gene panel design.** To create the gene panel for MERFISH experiments, we prioritized choosing marker genes that separate pairs of distinct clusters with >100 DEGs, and pairs of MB/HB cell types with >20 DEGs. We also excluded any genes that performed poorly in previous MERFISH experiments. We started with a default set of well-established marker genes curated from previous studies and selected additional genes to choose a minimal set that includes at least 2 DEGs for all such pairs in each direction. To do that, we used a greedy algorithm to choose one gene at a time that separates as many unresolved pairs as possible while still considering its relative statistical significance, which is implemented in select_N_markers function from scrattch.bigcat package, and selected the top 400 genes (including the default genes) from this list. We then attempted to select one DEG in each direction for any remaining pairs of clusters not covered by the selected genes using the same function. Our goal was to build a solid gene panel with strong predictive power at subclass level and be opportunistic at resolving finer cell types. Except for the default gene set, the remaining genes were largely ordered with decreasing predictive power. We submitted a total of 700 genes to the Vizgen portal and selected the top 500 genes that passed the additional filters applied by Vizgen. The final gene set provided an overall cross-validation accuracy of 76.6% at cluster level and 97.2% at subclass level.

**Assessing concordance of joint clustering between 10xv2 and 10xv3.** We first compared the joint clustering result with the independent clustering result from each dataset. We then calculated the cluster means of marker genes for each dataset. For each marker gene, we computed the Pearson correlation between its average expression for each cluster across two different datasets to quantify the consistency of its expression at the cluster level between datasets (Extended Data Fig. 7d). We performed a similar analysis between 10xv3 and MERFISH datasets.

**Building cell-type hierarchy.** To make the cell-type complexity tractable at each level, we organized the 5,322 clusters into a hierarchy with 4 levels: class, subclass, supertype and cluster. After clusters were computed as described in 'Joint clustering' section, we first defined

subclasses by clustering the clusters. This was performed by Jaccard–Leiden clustering using the average expression of 534 transcription factor marker genes of all the cells in each cluster, using 5 KNNs, and varying the resolution index of Leiden algorithm at 0.1, 0.2, 1, 5 and 8. We tried clustering using either all 8,460 marker genes or 534 transcription factor marker genes only, and found the result based on transcription factors recapitulate existing knowledge of cell types including spatial distribution and lineage relationships better. The Leiden algorithm generated 48 groups at resolution index 0.2, which generated the initial version of 'classes', and 240 groups at resolution index 8, which generated the initial version of 'subclasses'.

The initial fully automatically generated versions of classes and subclasses were visualized together with all the other metadata on UMAPs and on MERFISH sections using the single-cell data visualization tool cirrocumulus (v1.1.56, https://cirrocumulus.readthedocs.io/en/latest/) for manual examination. We fine tuned the borderline cases, and further split or merged some putative subclasses to reach the final definition of subclasses. We applied a similar process to define classes, and to achieve strict hierarchy, assigned all the clusters in one subclass to the same class. Finally, we applied the same Jaccard–Leiden algorithm to all the clusters within each subclass separately to define supertypes, using the union of the top 20 DEGs between all pairs of clusters within the subclass as features. Again, they were adjusted based on manual inspection of 2D and 3D UMAPs and MERFISH sections after visualization on cirrocumulus to increase the consistency of supertype definitions between subclasses. Comparison of automatically computed and manually revised, final definitions of cell classes and subclasses are shown as confusion matrices in Extended Data Fig. 4.

**UMAP projection.** We performed PCA based on the imputed gene expression matrix of 8,460 marker genes using the 10xv3 reference. We down-sampled up to 100 cells per cluster, and further down-sampled up to 250,000 cells if the total exceeded this number, so that PCA could proceed without any memory issues. Again, the principal components based on sampled cells were projected to the whole datasets. We selected the top 100 principal components, then removing one PC with more than 0.7 correlation with the technical bias vector, defined as $\log_2$(gene count) for each cell. We used the remaining principal components as input to create 2D and 3D UMAPs[134], using parameters nn.neighbors = 25 and md = 0.4. To prevent some of the big clusters taking up too much space, we down-sampled up to 1,000 cells per cluster to build the UMAP and imputed the UMAP coordinates of the other cells based on KNN neighbours among the sampled cells in the PCA space.

**Constellation plot.** The global relatedness between cell types was visualized using a constellation plot (Extended Data Fig. 6). To generate the constellation plot, each transcriptomic subclass was represented by a node (circle), whose surface area reflected the number of cells within the subclass in log scale. The position of nodes was based on the centroid positions of the corresponding subclasses in UMAP coordinates. The relationships between nodes were indicated by edges that were calculated as follows. For each cell, 15 nearest neighbours in reduced dimension space were determined and summarized by subclass. For each subclass, we then calculated the fraction of nearest neighbours that were assigned to other subclasses. The edges connected two nodes in which at least one of the nodes had >5% of nearest neighbours in the connecting node. The width of the edge at the node reflected the fraction of nearest neighbours that were assigned to the connecting node and was scaled to node size. For all nodes in the plot, we then determined the maximum fraction of "outside" neighbours and set this as edge width = 100% of node width. The function for creating these plots, plot_constellation, is included in scrattch.bigcat.

**Assigning subclass, supertype and cluster names.** We first annotated each subclass with its most representative anatomical region(s)

and named the subclass using the combination of its representative region(s), major neurotransmitter, and in some cases one or two marker genes. We then ordered the subclasses based on the taxonomy tree and assigned subclass IDs accordingly. Supertype names within each subclass were defined by combining the subclass name and the grouping numbers of supertypes within the subclass. Supertype IDs were assigned sequentially based on the taxonomy tree order of subclasses and the group order of supertypes within each subclass. Cluster IDs were also assigned sequentially based on the ordering of subclasses and supertypes. And the final cluster names were assigned by combining each cluster's ID with the name of the supertype the cluster belongs to. Based on the Allen Institute proposal for cell-type nomenclature[135], we also assigned accession numbers to cell types, as included in Supplementary Table 7.

**Assigning cell-type identities within a modality for cross-validation and across modalities.** We performed fivefold cross-validation using different sets of marker genes: all 8,460 marker genes ('Marker gene selection'), 534 transcription factor marker genes, and 20 sets of 534 randomly sampled marker genes from the 8,460-marker list. We defined the cluster centroid in each modality as the average gene expression for all the training cells within the cluster and built the Annoy KNN indices based on user specified distance metrics (cosine by default) using the chosen marker list. For the testing cells in each modality, we assigned their cell-type identities by mapping them to the nearest cluster centroid using the corresponding Annoy index. This process is implemented in map_cells_knn_big function from scratch.bigcat package, and mapping can be performed very efficiently by massive parallelization.

We also developed a hierarchical version of this approach to assign cell-type identities for MERFISH, Multiome snRNA-seq, or any external datasets to the 10xv3 dataset as reference, using different gene lists based on the contexts. When mapping confidence was needed, we sampled 80% genes from the marker list randomly, and performed mapping 100 times. The fraction of times a cell is assigned to a given cell type is defined as the mapping probability.

For the hierarchical mapping, we first built a tree with root, classes, subclasses, and clusters. At each internal node, we selected markers that best discriminate the clusters from different child nodes and assign the query cells to the child node with the nearest cluster centroids based on the selected markers. The process was repeated at each level of the tree till the query cells were mapped to the leaf-level clusters. This algorithm is implemented in the scratch-mapping package and publicly accessible (v0.2, https://github.com/AllenInstitute/scratch.mapping).

**Imputation.** To facilitate direct comparisons of cells from different platforms, we projected gene expression of the 10xv2 or MERFISH dataset (query) to the 10xv3 dataset (reference). The basic idea is to compute the KNNs ($k = 15$ by default) among the reference cells for each query cell and use the average expression of these neighbours for each gene as the imputed values. One of the key decisions is selection of the distance metric used to compute the KNNs. We chose cosine metric due to overall conservation of gene expression at the cluster level (Extended Data Fig. 7d). On the other hand, the conservation is not perfect, and using too many genes to infer KNNs could make the inference more susceptible to platform differences. Therefore, we used the 500 MERFISH marker genes to compute KNNs, since they provide good predictive powers at all levels of the hierarchy and show high correlation at the cluster level between 10xv2 and 10xv3 platforms (median 0.945, Extended Data Fig. 7d). Although a good starting point, the imputation accuracy for separating finer cell types could be improved further by incorporating cluster-level DEGs, as fewer of them were included in the 500 MERFISH gene list and they are not completely binary. To solve this problem, we leveraged the established cell-type hierarchy and performed imputation iteratively, first at top level and then within each class and subclass. At the top level, we used the 500 MERFISH genes to compute KNNs and then imputed the expression of all 8,460 marker genes. For each subsequent iteration, we only computed the KNNs among the reference cells within the same class/subclass as the query cells using the top 10 DEGs between clusters within the given class/subclass; we then updated the imputed expression of the top 20 DEGs between clusters within the given class/subclass.

An alternative simpler strategy is to simply compute the KNNs of each query cell among the reference cells within the same cluster and/or the same subclass and impute the expression of all marker genes. However, the imputation values using this strategy could not preserve the transitions between clusters or subclasses, exaggerating the separations between cell types, especially at the finer level.

Recent benchmark studies indicate that for simpler integration problems with relatively low biological complexity and relatively small batch effects in scRNA-Seq datasets, linear methods outperform nonlinear, more complex methods[136,137], while for complicated integration tasks with large biological complexity and bigger batch effects, nonlinear methods such as scVI/scanVI outperform others[138]. In our case, the overall batch effects between 10xv2 and 10xv3 are relatively small, but biological complexities are huge. Therefore, to tackle this complexity in a divide and conquer manner, we used series of linear imputations with increasing resolutions to approximate the nonlinear relationship, as any nonlinear curve can be accurately approximated using a series of linear segments. This method provides the scalability/robustness to very large datasets while preserving fine-grained cell-type resolution. The method is implemented in the impute_cross_big function in scratch.bigcat, with predefined split of the datasets at various levels, the genes used for KNN inference, and the genes used for imputation as inputs.

In parallel, we have also tested nonlinear methods including scVI/scanVI. To make it work for this large dataset with high complexity, we down-sampled the datasets and increased the size of neuronal network model dramatically to achieve reasonable performance. There is a huge parameter space we need to explore to further optimize the performance, and this is an active area of further investigation.

We used the same strategy to impute the expression of 8,460 marker genes for the MERFISH dataset, except that only the DEGs present in the 500 MERFISH gene panel were used to compute KNNs at class and subclass levels. This strategy still helped to reduce the effect of contamination from neighbouring cells due to imperfect segmentation. Validation results are shown in Extended Data Fig. 8.

The imputation results for 10xv2, 10xMulti and MERFISH datasets, along with the 10xv3 dataset as the anchor, were used to generate the integrated UMAP shown in Extended Data Fig. 7a.

**Defining neurotransmitter types.** We systematically assigned neurotransmitter identity to each cell cluster based on the expression of canonical neurotransmitter transporter genes and synthesizing enzymes (Fig. 1e, Fig. 3, Extended Data Fig. 3e, Extended Data Fig. 9, Supplementary Table 7). Criteria used are:

Glutamatergic (Glut): *Slc17a6* (also known as *Vglut2*), *Slc17a7* (*Vglut1*) or *Slc17a8* (*Vglut3*).

GABAergic (GABA): (*Slc32a1* (*Vgat*) or *Slc18a2* (*Vmat2*)) and (*Gad1*, *Gad2* or *Aldh1a1*).

Glycinergic (Glyc): *Slc6a5*.

Cholinergic (Chol): *Slc18a3* (*Vacht*) and *Chat*.

Dopaminergic (Dopa): (*Slc6a3* (*Dat*) or *Slc18a2*) and (*Th* and *Ddc*).

Serotonergic (Sero): (*Slc6a4* (*Sert*) or *Slc18a2*) and (*Tph2* and *Ddc*).

Noradrenergic (Nora): (*Slc6a2* (*Net*) or *Slc18a2*) and *Dbh*.

Histaminergic (Hist): *Slc18a2* and *Hdc*.

We used a stringent expression threshold of $\log_2$(CPM) > 3 for these genes to assign neurotransmitter identity to each cluster.

These criteria are stringent as they require co-expression of both a neurotransmitter transporter and the corresponding key neurotransmitter synthesizing enzyme(s). They are also inclusive as alternative

neurotransmitter synthesizing and releasing genes are included. For example, we included the vesicular monoamine transporter *Slc18a2* (*Vmat2*) to all monoamine transmitters as well as GABA. It is known that in many midbrain dopamine neurons (in VTA and SNc), *Aldh1a1* is used for synthesizing GABA in the absence of *Gad1* or *Gad2*, and *Slc18a2* is used for co-release of dopamine and GABA in the absence of *Slc32a1*[57].

Of note, a reported unconventional mechanism[118] underlying co-transmission of dopamine and GABA by SNc dopaminergic neurons in the STR does not depend on cell-autonomous GABA synthesis but instead on presynaptic uptake from the extracellular space through the GABA transporter *Slc6a1* (*Gat1*) while still relying on *Slc18a2* for synaptic GABA release, which could make *Aldh1a1* unnecessary in these cells. However, because *Slc6a1* is widely expressed in all GABAergic neurons, as well as astrocytes, and even many subcortical glutamatergic neurons, it is unclear to us how widely applicable this unconventional mechanism (which bypasses all GABA-synthesizing enzymes) is. Therefore, we did not include *Slc6a1* in our criteria in order to minimize false positives, even at the risk of having some false negatives.

**Defining transcription factor co-expression gene modules.** To identify transcription factor gene modules that are involved in defining major cell types (Fig. 5d, Supplementary Table 8), we performed WGCNA analysis[139] on 534 transcription factor marker genes based on their average expression at the subclass level with power = 6 and TOMType = "signed", and detectCutHeight = 0.998. Genes in the 'grey' module were removed, which had poor correlation with all the other genes, and genes that were generally enriched in neurons were excluded. Genes in some modules clearly had distinct patterns and were thus further split, and they were re-ordered for better visualization.

## MERFISH

**Brain dissection and freezing.** Standard procedures were developed to isolate, cut, fix and pre-treat tissue to preserve macro and cellular morphology and to produce the best signal to noise ratio for MERFISH. Mice were transferred from the vivarium to the procedure room with efforts to minimize stress during transfer. If mouse body weight fell outside of the normal range (18.8 to 26.4 g), the brain was not used in the MERFISH process. Mice were anaesthetized with 0.5% isoflurane. A grid-lined freezing chamber was designed to allow for standardized placement of the brain within the block in order to minimize variation in sectioning plane. Chilled OCT was placed in the chamber, and a thin layer of OCT was frozen along the bottom by brief placement of the chamber in a dry ice ethanol bath. The brain was rapidly dissected and placed into the OCT. The orientation of the brain was adjusted using a dissecting scope, and the freezing chamber containing OCT and brain were frozen in a dry ice/ethanol bath. Brains were stored at −80 °C.

**Cryosectioning.** The fresh frozen brain was sectioned at 10 μm on Leica 3050 S cryostats. The OCT block containing a fresh frozen brain was trimmed in the cryostat until reaching the desired starting section. Sections were collected every 200 μm to evenly cover the brain from anterior to posterior and each section was mounted onto a functionalized 20-mm coverslip treated with yellow green (YG) fluorescent microspheres (VIZGEN, 2040003)

**Fixation and dehydration.** After air drying on the coverslips for 10–15 min, the tissue sections were loaded into a Leica Autostainer XL (Leica ST5010). They were washed in 1× PBS for 1 min, fixed in 4% PFA for 15 min, washed in 1× PBS for 5 min 3 times, washed in 70% ethanol and then stored in 70% ethanol at 4 °C. They were stored for at least one day and no more than 6 weeks before proceeding.

**Hybridization.** For staining the tissue with MERFISH probes a modified version of instructions provided by the manufacturer was used. All solutions were prepared according to the instruction provided by the manufacturer. For hybridization samples were removed from the 70% ethanol and washed in a petri dish containing VIZGEN sample prep buffer (VIZGEN, 20300001). Sample prep buffer was aspirated, and the samples were equilibrated with 5 ml of VIZGEN formamide wash buffer (VIZGEN, 20300002) in a humidified incubator at 37 °C for 30 min. Formamide wash buffer was removed via aspiration and a 50-μl droplet of MERSCOPE Gene Panel Mix was added onto the centre of the tissue section. Next, the tissue section was covered with parafilm and stored in a humidified 37 °C cell culture incubator for 36–48 h.

**Gel embedding.** Parafilm covering the sections was removed and 5 ml of the VIZGEN formamide wash buffer was immediately added. Sections were incubated at 47 °C for 30 min. Formamide wash buffer was aspirated and the previous step repeated. Sections were washed with VIZGEN sample prep wash buffer after the second formamide wash for 2 min. 110 μl of VIZGEN gel embedding solution (VIZGEN 20300004) with APS and TEMED was added onto the centre of a Gel Slick-coated microscope slide and any excess embedding solution was gently removed.

To allow for the gel to fully polymerize the sections were incubated at room temperature for 1.5 h. To clear the tissue the section was incubated in 5 ml of VIZGEN Clearing Solution (VIZGEN 20300003) with Proteinase K (NEB P8107S) according to the Manufacturer's instructions for at least 24 h or until it was clear in a humidified incubation oven at 37 °C.

**Imaging.** Following clearing, sections were washed twice for 5 min in Sample prep wash buffer (VIZGEN, 20300001). VIZGEN DAPI and PolyT Stain (VIZGEN, 20300021) was applied to each section for 15 min followed by a 10 min wash in formamide wash buffer. Formamide wash buffer was removed and replaced with sample prep wash buffer during MERSCOPE set up. 100 μl of RNAse Inhibitor (New England BioLabs M0314L) was added to 250 μl of Imaging Buffer Activator (VIZGEN, 203000015) and this mixture was added via the cartridge activation port to a pre-thawed and mixed MERSCOPE Imaging cartridge (VIZGEN, 1040004). Fifteen millilitres of mineral oil (Millipore-Sigma m5904-6X500ML) was added to the activation port and the MERSCOPE fluidics system was primed according to VIZGEN instructions. The flow chamber was assembled with the hybridized and cleared section coverslip according to VIZGEN specifications and the imaging session was initiated after collection of a 10× mosaic DAPI image and selection of the imaging area. For specimens that passed the minimum count threshold, imaging was initiated, and processing completed according to VIZGEN proprietary protocol.

**Data analysis.** Raw MERSCOPE data were decoded using Vizgen software (v231). Cell segmentation was performed as described previously[140]. In brief, cells were segmented based on DAPI and PolyT staining using Cellpose[141]. Segmentation was performed on a median *z*-plane (4th out of 7) and cell borders were propagated to *z*-planes above and below. The resulting cell-by-gene table was filtered to keep cells with a volume >100 μm$^3$ and <3,000 μm$^3$, that have at least 15 genes detected and contain a minimum of 40 but no more than 3,000 mRNA molecules (red dashed lines in Extended Data Fig. 2d,e) to remove low-quality cells and doublets that are outside of these ranges. Overall counts of genes were normalized by cell volume and log$_2$-transformed. To assign cluster identity to each cell in the MERFISH dataset, we mapped the MERFISH cells to the scRNA-seq reference taxonomy. For this, the 10xv3 scRNA-seq data was subsetted to only genes common to both datasets. Our mapping method (as described in 'Assigning cell-type identities') finds the nearest cluster centroid in the scRNA-seq reference dataset for a query data point with the correlation of shared genes as distance metric. The cluster label of the nearest neighbour was assigned as mapped label. Bootstrapping was conducted with 80% subsampling of marker genes to make label assignment robust.

**Registration to Common Coordinate Framework.** To facilitate alignment of MERFISH sections to the CCFv3, we assigned each cell from the

scRNA-seq dataset to one of these major regions: cerebellum, CTXsp, hindbrain, HPF, hypothalamus, isocortex, LSX, midbrain, OLF, PAL, sAMY, STRd, STRv, thalamus and hindbrain. This delineation was driven by the level of region-specific dissection for the scRNA-seq experiments as well as the cell-type specificity of regions. Because of the more gradient transition of cell-type composition between cortical regions, the specificity of cortical plate regions is limited to isocortex, OLF and HPF despite more granular dissection regions. Each cluster in the scRNA-seq dataset was assigned to the region the majority of cells were derived from. We identified anchor clusters we used for region annotation of the MERFISH data. These clusters were defined as (1) having more that 30% of all cells in one region, and (2) more than 20 cells in a MERFISH section. In addition to that we used ependymal and choroid plexus cells to label the ventricles and identified specific clusters of oligodendrocytes that were enriched in white matter tracts. To account for clusters that were found at low frequency in regions outside its main region we calculated for each cell its 50 nearest neighbours in physical space and reassigned each cell to the region annotation dominating its neighbourhood. Next, we used that same approach to assign each cell mapped to a non-anchor cluster to the region annotation dominating its immediate surrounding. The resultant label maps were used as input to our registration tool to find for each section its approximate location along the anterior to posterior axis of the brain as well as any offsets in pitch and yaw introduced during sectioning.

Registration was performed at 10-μm in-plane resolution. For each section, an anatomical reference image was created by aggregating the number of detected spots within a 10 × 10 μm grid for each gene probe. A single image was created across all probes by taking the maximum count for each grid unit. The midline was manually determined by annotating the most dorsal and most ventral point. These points were then used to compute a rigid transform to rotate the section upright and centre in the middle. This set of rectified images were stacked in sequential order to create an initial configuration for registration.

Alignment to the Allen CCFv3 was performed by matching the above-mentioned scRNA-seq derived region labels to their corresponding anatomical parcellation of the CCFv3. A label map was generated for each region by aggregating the cells assigned to that region within a 10×10 μm grid, transformed to the initial configuration using the computed rigid transforms. Using the corresponding anatomical labels, the ANTS registration framework was used to establish a 2.5D deformable spatial mapping between the MERFISH data and the CCFv3 via three major steps: (1) A 3D global affine (12 dof) mapping was performed to align the CCFv3 into the MERFISH space. This generated resampled sections from the CCFv3 that provided section-wise 2D target space for each of the MERFISH sections. Since the CCFv3 is a continuous label set with isotropic voxels, this avoids interpolation artifacts that can result if resampling is performed on the MERFISH data instead, which has large section gaps, and can contain missing sections. (2) After establishing a resampled CCFv3 section for each MERFISH section, 2D affine registrations were performed to align each MERFISH section to match the global anatomy of the CCFv3 brain. This addressed misalignments from the initial manual stacking of the MERFISH sections using the midline and provided a global mapping to initialize the local deformable mappings. (3) Finally, a 2D multi-scale, symmetric diffeomorphic registration (step size = 0.2, sigma = 3) was used on each section to map local anatomic differences between the corresponding MERFISH and CCFv3 structures in each section. Global and section-wise mappings from each of these registration steps were preserved and concatenated (with appropriate inversions) to allow point-to-point mapping between the original MERFISH coordinate space and the CCFv3 space.

After registration to CCFv3, we found that out of 554 terminal regions (grey matter only, Supplementary Table 1), there were only 7 small subregions completely missed in the MERFISH dataset: frontal pole, layer 1 (FRP1), FRP2/3, FRP5, accessory olfactory bulb, glomerular layer (AOBgl), accessory olfactory bulb, granular layer (AOBgr), accessory olfactory bulb, mitral layer (AOBmi) and accessory supraoptic group (ASO).

**Quantifying spatial distribution of cell types.** Based on the CCFv3 registration results, each MERFISH cell was assigned to a CCFv3 structure. For further quantification, we aggregated the CCFv3 at two levels of hierarchy (CCFv3_level1 and CCFv3_level2, Supplementary Table 9) focusing on grey matter structures only. Only 51/59 sections fell within the boundaries of the CCFv3. In addition, potentially misaligned cells were filtered out by excluding cells of a subclass in a CCFv3_level1 region if less than 5% of cells were present in that region. This limits the number of cells used for spatial analysis to 3,062,367 cells. We summed up all cells per subclass within a region and normalized by the max number of subclasses per region (Enrichment, Fig. 6a). Regions were ordered by graph order of the CCFv3 (Supplementary Table 9), and subclasses were order by class first and within class by subclass ID. Class order was slightly altered to have less emphasize on neurotransmitter type and more on region specificity. We did not include the class 25 Pineal Glut, since the pineal body is not part of the CCFv3. To calculate the neuron/glia ratio we assigned various classes as either neuronal or glial (Supplementary Table 7) and summed up the cells per CCFv3_level2 region. A similar approach was used to calculate the neurotransmitter composition per region based on the assigned neurotransmitter identity for each cluster (Supplementary Table 7).

**Gini coefficient.** To quantify the distribution patterns of cell types across brain regions we calculated the Gini coefficient for each subclass at the CCFv3_level2 region annotation (Fig. 6a and Extended Data Fig. 14). The Gini coefficient is a measure of inequality within a distribution. It's a number between 0 and 1, where 0 represents perfect equality (every region has the same relative number of cells of a type), and 1 represents maximum inequality (a cell type is found in only one region). Before calculating the Gini coefficient, the number of cells per subclass in each region was normalized by the total number of cells per region to account for difference in volume and hence total number of cells of individual brain regions. We used the Gini function from the R package DescTools to calculate the Gini coefficient for each subclass.

**Shannon diversity.** To measure complexity of brain regions with respect to cell-type composition we calculated the Shannon diversity index. The Shannon diversity index quantifies the uncertainty or entropy in a system by considering the distribution of cell types and their abundance. It combines richness (number of distinct cell types) and evenness to provide a measure of diversity, reflecting the information content or disorder in the system. Higher values indicate greater diversity and more uniform distribution of species. We used the diversity function from the R package vegan. We calculated the Shannon diversity index for CCFv3_level2 regions for both the composition of subclasses and supertypes (Fig. 6a).

**Mutual prediction between transcriptomic identity and spatial localization.** To examine the relationship between transcriptomic identity and spatial localization for neuronal cells, we tested to what extent transcriptomic cell types can be predicted based on spatial location and vice versa. As glutamatergic and GABAergic neurons colocalize in many brain regions, to simplify the problem, we first divided neurons based on their class assignment, with cells from the GABA classes in one group, and cells from Glut, Dopa and Sero classes in another group. For this analysis, our goal is to understand the relationship between transcriptome and spatial location within either group.

We used the MERFISH dataset for this test. To predict spatial localization based on transcriptome, we used the imputed MERFISH transcriptomes of all 8,460 marker genes. Within each group, we first computed the top 20 DEGs between all pairs of clusters within the group and pooled them. We then performed PCA on the imputed MERFISH transcriptomes using selected markers and selected the top 100 components to compute the KNNs (excluding itself) of each cell. Then for each cell, we predicted its CCFv3 region based on the

majority votes of the CCFv3 regions of its transcriptomic KNNs computed as described above. The confusion matrix between the actual CCFv3 region and the predicted CCFv3 region (Extended Data Fig. 16) indicates which CCFv3 regions share cells with homogeneous transcriptomic signatures.

Similarly, we computed KNNs of each cell based on their 3D spatial coordinates and predicted its cell-type identity at class, subclass and supertype levels based on the majority votes of the cell type of its spatial KNNs. The confusion matrix between the actual cell-type identity and the predicted cell type (Extended Data Fig. 15) indicates the cell types that colocalized in the same spatial location.

This analysis showed that for most of the CCFv3 regions, each region contains cells with distinct transcriptomic signatures from other regions and confusion usually only exists among neighbouring regions. One exception is cortical GABAergic cell types, which are shared across all cortical areas as we previously reported[23,26] (Extended Data Fig. 16b). We could still observe partial separation of upper and lower cortical layers due to enrichment of CGE and MGE neurons respectively. The analysis also showed that each subclass only colocalizes spatially with a few other subclasses within the same group, except for a couple of HB subclasses that are highly heterogenous (Extended Data Fig. 15).

Note that the precision of this analysis is highly dependent on the accuracy of CCFv3 registration. Any confusion arising from neighbouring CCFv3 regions might be due to registration inaccuracies, an aspect we are actively working to improve.

## Reporting summary

Further information on research design is available in the Nature Portfolio Reporting Summary linked to this article.

## Data availability

The data were generated under the BRAIN Initiative Cell Census Network (BICCN, www.biccn.org, RRID: SCR_015820) and are accessible through Neuroscience Multi-omic Data Archive (NeMO, https://nemoarchive.org/) and Brain Image Library (BIL; https://www.brain-imagelibrary.org/index.html). The AIBS 10x scRNA-seq datasets (FASTQ files) are available at NeMO (https://assets.nemoarchive.org/dat-qg7n1b0). The 10x scRNA-seq sequencing data are also uploaded to Gene Expression Omnibus (GEO) under accession code GSE246717, and to BioProject under accession code PRJNA1030397. The AIBS MERFISH dataset is available at BIL (https://doi.org/10.35077/g.610). Allen Brain Cell Atlas—mouse whole-brain cell-type atlas is accessible at https://portal.brain-map.org/atlases-and-data/bkp/abc-atlas, to visualize scRNA-seq, snRNA-seq and MERFISH datasets. Instructions for access to the processed 10x scRNA-seq data are available at https://github.com/AllenInstitute/abc_atlas_access/blob/main/descriptions/WMB–10X.md, and instructions for access to the processed MERFISH data are available at https://github.com/AllenInstitute/abc_atlas_access/blob/main/descriptions/MERFISH-C57BL6J-638850.md. Source data are provided with this paper.

## Code availability

Data analysis code used in the manuscript—R package scratch.bigcat—is available via github https://github.com/AllenInstitute/scratch.bigcat. Cell-type mapping code is available via github https://github.com/AllenInstitute/scratch.mapping.

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

**Acknowledgements** We are grateful to the Transgenic Colony Management, Lab Animal Services, Molecular Biology, Spatial Transcriptomics, and Histology teams at the Allen Institute for technical support. The research was funded by grants from National Institute of Mental Health, U19MH114830 to H.Z. and X.Z., and U24MH130918 to S. Mufti, M.J.H. and L.N., under the BRAIN Initiative of National Institutes of Health (NIH). The content is solely the responsibility of the authors and does not necessarily represent the official views of NIH and its subsidiary institutes. X.Z. is a Howard Hughes Medical Institute investigator. This work was also supported by the Allen Institute for Brain Science. The authors thank the Allen Institute founder, Paul G. Allen, for his vision, encouragement, and support.

**Author contributions** Conceptualization: H.Z. Data analysis lead and coordination: Z.Y. Data generation (scRNA-seq): Z.Y., C.T.J.v.V., D.M., A.A., E.B., D.B., D.C., T.C., M. Clark, K.C., L. Ellingwood, A.G., J. Gloe, J. Gray, N.G., J. Guzman, D.H., W.H., M. Kroll, K.L., R. McCue, E.M., T.N.N., T.P., C.R., N. Shapovalova, J.S., M.T., A.T., H.T., K. Wadhwani, K. Ward, B. Levi, N.D., K.S., B.T., H.Z. Data processing and analysis (scRNA-seq): Z.Y., C.T.J.v.V., C.L., J. Goldy, M.A., A.B.C., R.C., S.C., T.D., J. Gould, M. Hooper, R. McGinty, N. Mei, M.-Q.M.W., K.S., B.T., H.Z. Data generation (MERFISH): M. Kunst, M.Z., D.M., W.J., J. Campos, M. Huang, M. Hupp, J. Malone, N. Martin, A.R., N.V.C., J.W., X.Z., H.Z. Data processing and analysis (MERFISH): Z.Y., C.T.J.v.V., M. Kunst, M.Z., D.M., W.J., M.A., M. Chen, J. Close, S.D., J. Gee, M. Huang, A.L., B. Long, Z. Maltzer, N. Mei, C.S., J.W., L.N., X.Z., H.Z. Online visualization platform development: M.A., K.B., A.B., C.B., P. Bishwakarma, P.D., E.F., T.F., J. Gerstenberger, M. Huang, S.L., Z. Madigan, J. Malloy, R. McGinty, N. Mei, J. Melchor, S. Moonsman, S.O., R.S., L.S., N. Shephard, S.V., B.S., M.-Q.M.W. Project management: P. Baker, C.L.T., C.M.P., L.K., S.M.S., K.S. Management and supervision: Z.Y., C.T.J.v.V., D.M., J. Close, J. Gee, B. Long, T.M., B. Levi, C.F., R.Y., B.S., M.-Q.M.W., C.L.T., S. Mufti, N.D., S.M.S., L. Esposito, M.J.H., J.W., L.N., K.S., B.T., X.Z., H.Z. Manuscript writing and figure generation: Z.Y., C.T.J.v.V., M. Kunst, H.Z. Manuscript review and editing: Z.Y., C.T.J.v.V., M. Kunst, K.J., M.J.H., L.N., B.T., X.Z., H.Z.

**Competing interests** H.Z. is on the scientific advisory board of MapLight Therapeutics, Inc. X.Z. is a co-founder of and consultant for Vizgen.

**Additional information**
**Correspondence and requests for materials** should be addressed to Zizhen Yao or Hongkui Zeng.

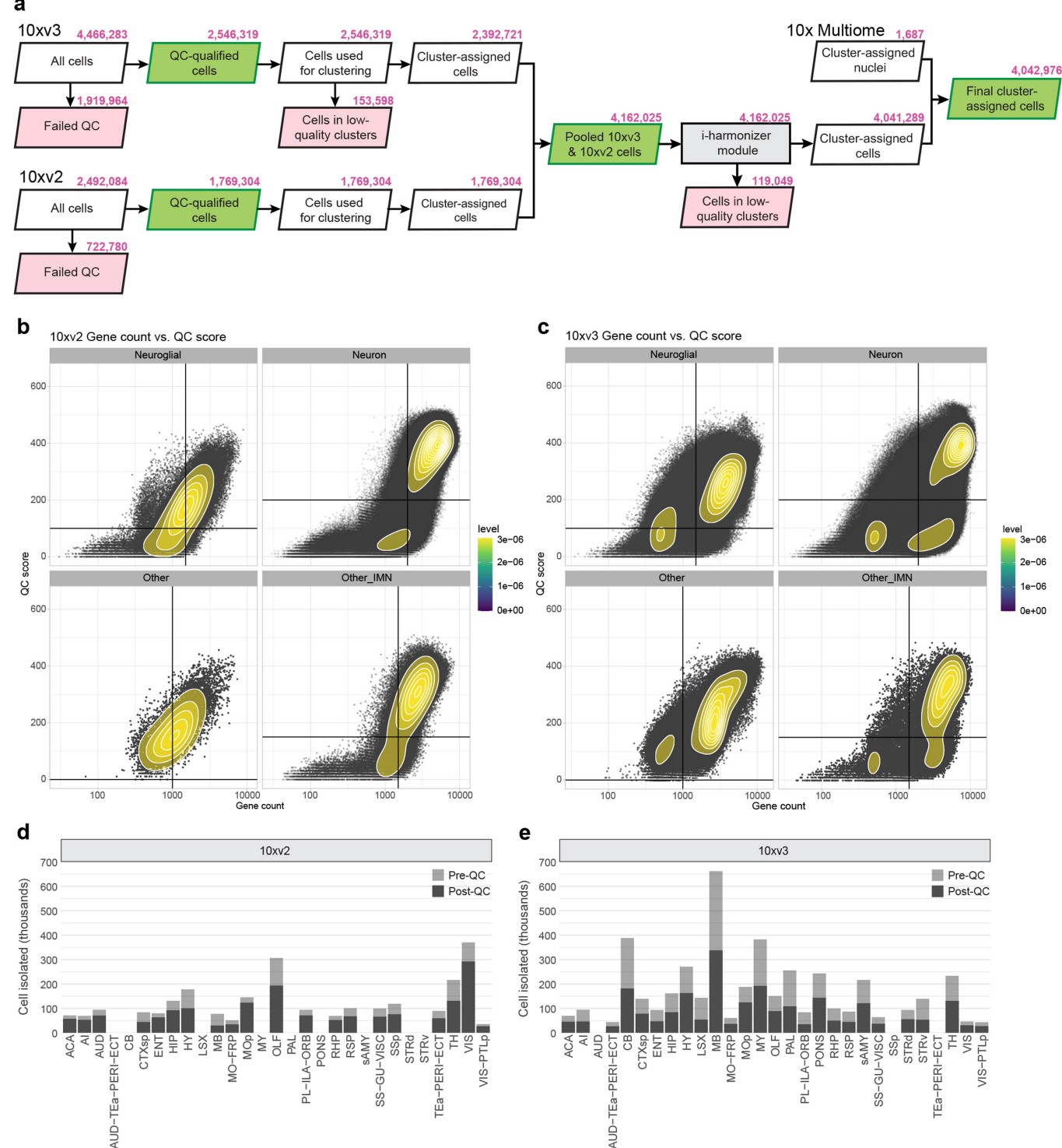

**Extended Data Fig. 1 | scRNA-seq data analysis workflow. (a)** Number of cells at each step in the scRNA-seq data analysis pipeline. The identification of doublets and low-quality clusters is described in more detail in Methods. The 10xv2 and 10xv3 data were first QC-ed and analyzed separately. After initial clustering the datasets were combined and QC-ed again before and after joint clustering. 10x Multiome snRNA-seq data was added to fill in gaps that were identified after joint clustering of 10xv2 and 10xv3 scRNA-seq data. **(b-c)** Gene count and qc score thresholds used for each of the four major cell populations (neuroglial cells, neurons, immature neurons and granule cells, and other) on the 10xv2 (b) and 10xv3 (c) datasets. **(d-e)** Number of cells isolated from dissection ROIs (pre-QC) and number of cells passing QC (post-QC) for 10xv2 (d) and 10xv3 (e) datasets. We didn't profile LSX, STR, sAMY, PAL, Pons, MY, and CB by 10xv2. Some regions were collected using different dissections between 10xv2 and 10xv3, but all regions were covered by 10xv3.

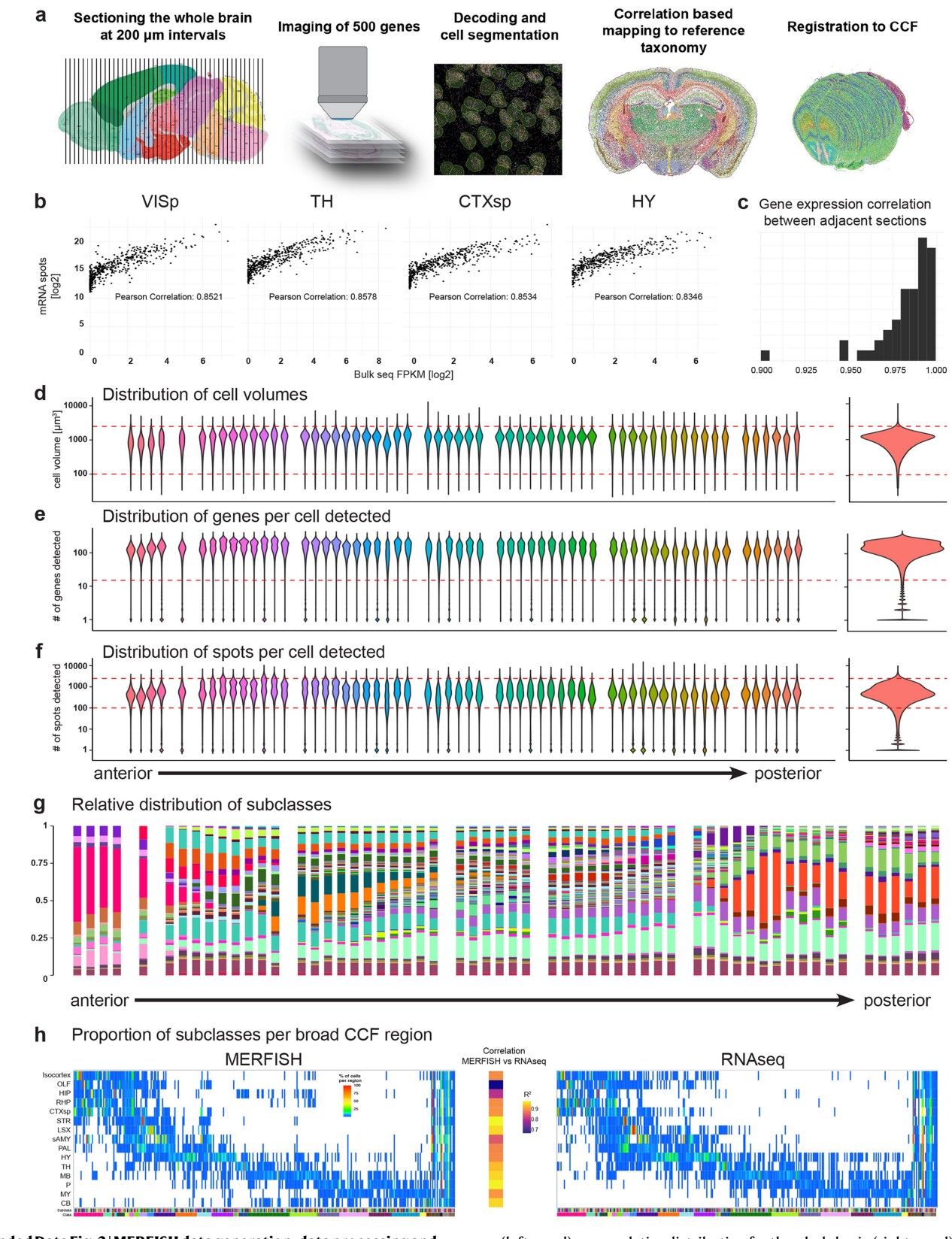

**Extended Data Fig. 2 | MERFISH data generation, data processing and summary of results.** (**a**) Workflow for generating and processing MERFISH data. (**b**) Correlation of gene detection between MERFISH and bulk RNA-sequencing for four different brain regions. (**c**) Histogram displaying the proportion of gene detection correlation between adjacent MERFISH sections. (**d-f**) Violin plots displaying distribution of cell volumes (d), gene detection (e), and mRNA molecule detection (f) for individual sections ordered from anterior to posterior (left panel) or cumulative distribution for the whole brain (right panel). Red dashed lines indicate cutoff for filtering. (**g**) Cumulative histogram showing the relative contribution of each subclass to each section ordered from anterior to posterior. (**h**) Heatmap showing the proportion of cells per region for each subclass in the MERFISH data (left) and scRNA-seq data (right). The heatmap in the middle shows the correlation between region distribution of MERFISH and scRNA-seq data.

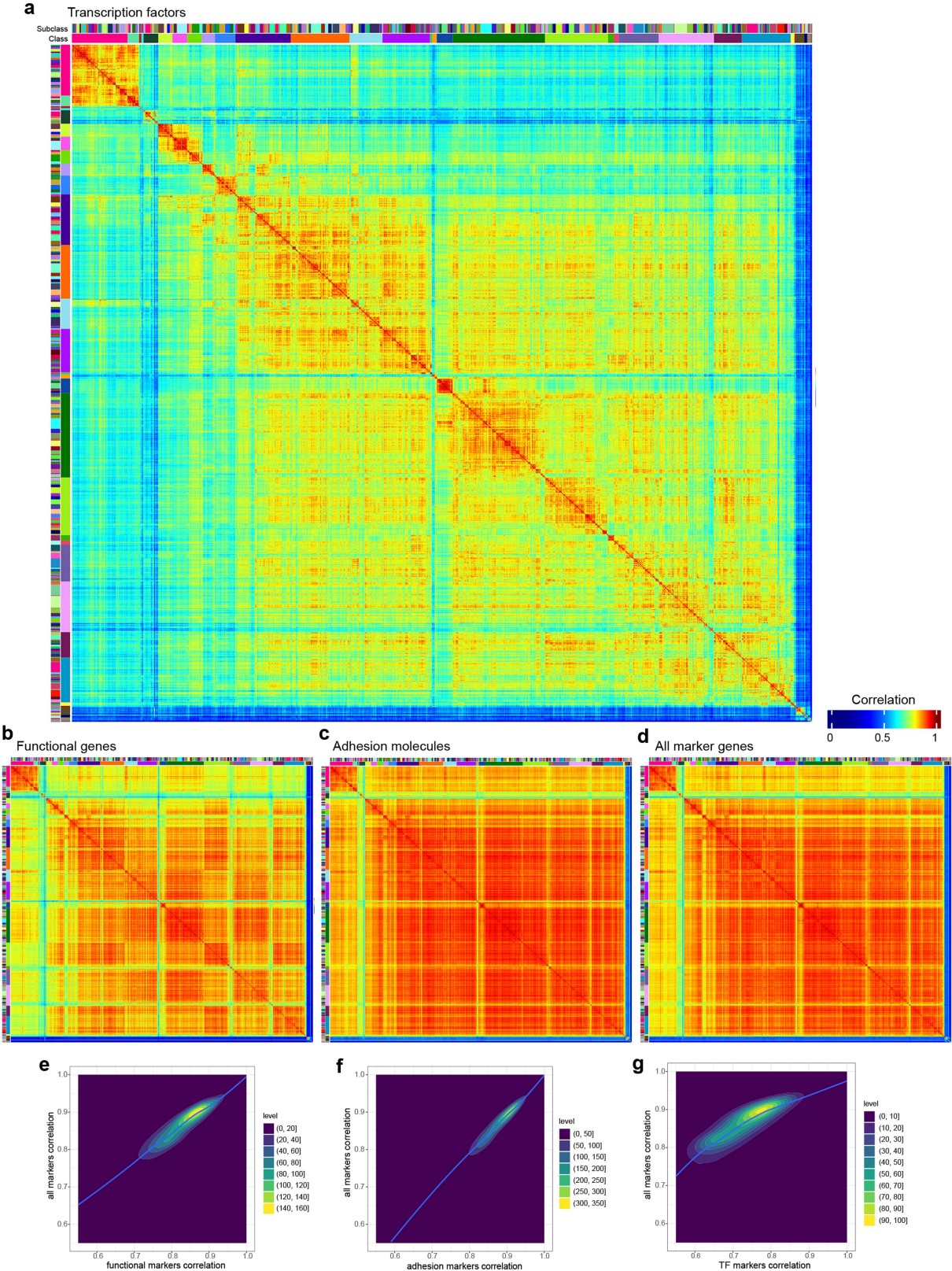

**Extended Data Fig. 3 | Marker gene expression correlation matrices showing relatedness among cell types. (a-d)** Heatmaps showing pairwise Pearson correlation of gene expression levels for each pair of clusters using marker gene sets of 534 transcription factors (a), 541 functional genes (including neuropeptides, GPCRs, ion channels, transporters, etc.) (b), 857 adhesion molecules (c), and all 8,460 marker genes (d). Correlations were computed using 10xv3 scRNA-seq data only except for the 31 nuclei-dominated clusters where 10xMulti snRNA-seq data were used. **(e-g)** All marker gene expression correlation between clusters compared to correlation between clusters of expression of functional marker genes (e), adhesion marker genes (f), and transcription factor (TF) marker genes (g). Correlation values are derived from a-d.

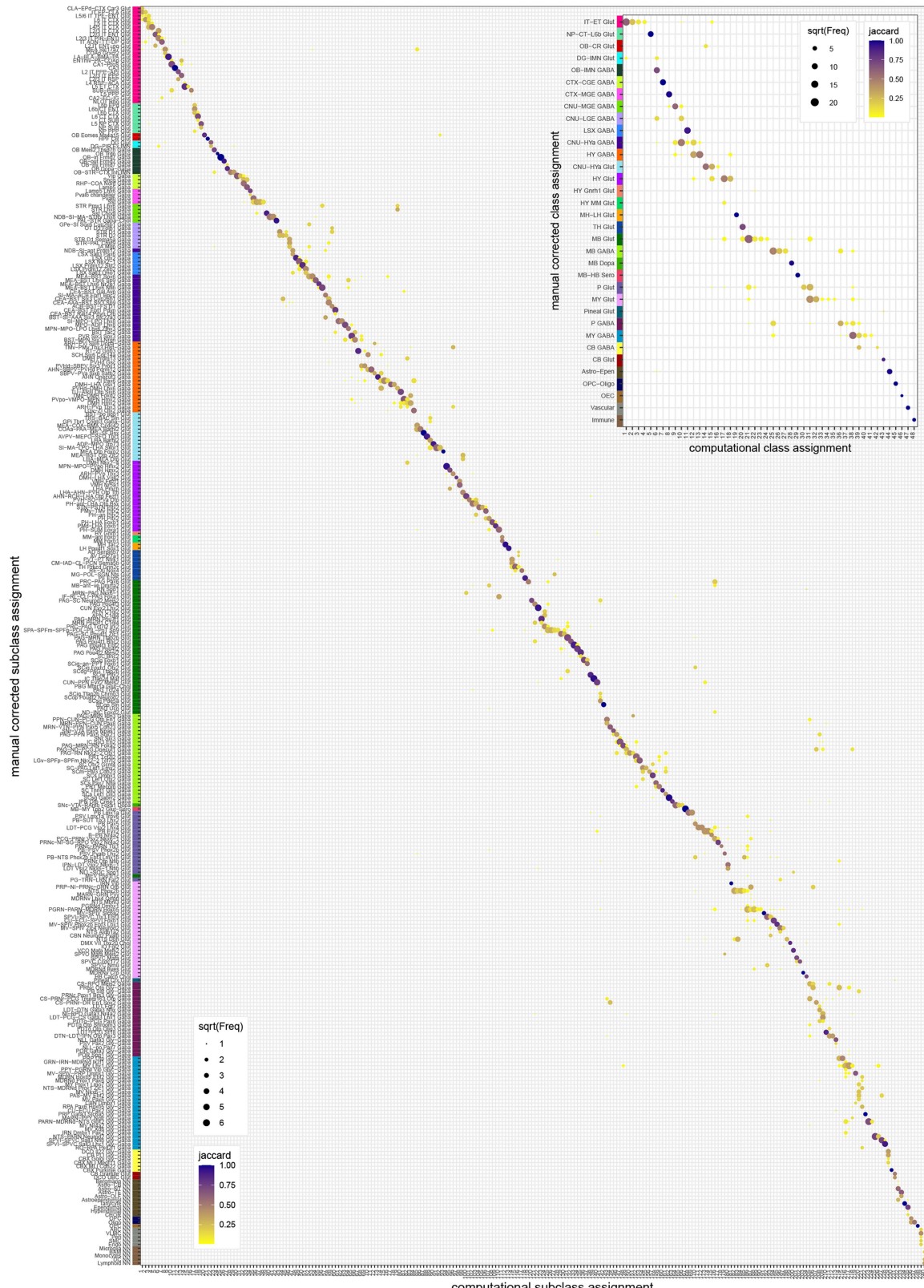

**Extended Data Fig. 4 | Comparison of the initial automated computational assignment of classes and subclasses with the manually revised, final assignment of classes and subclasses.** Size of the dot corresponds to the number of overlapping cells (frequency) in corresponding classes or subclasses, and color represents the Jaccard similarity between corresponding classes or subclasses.

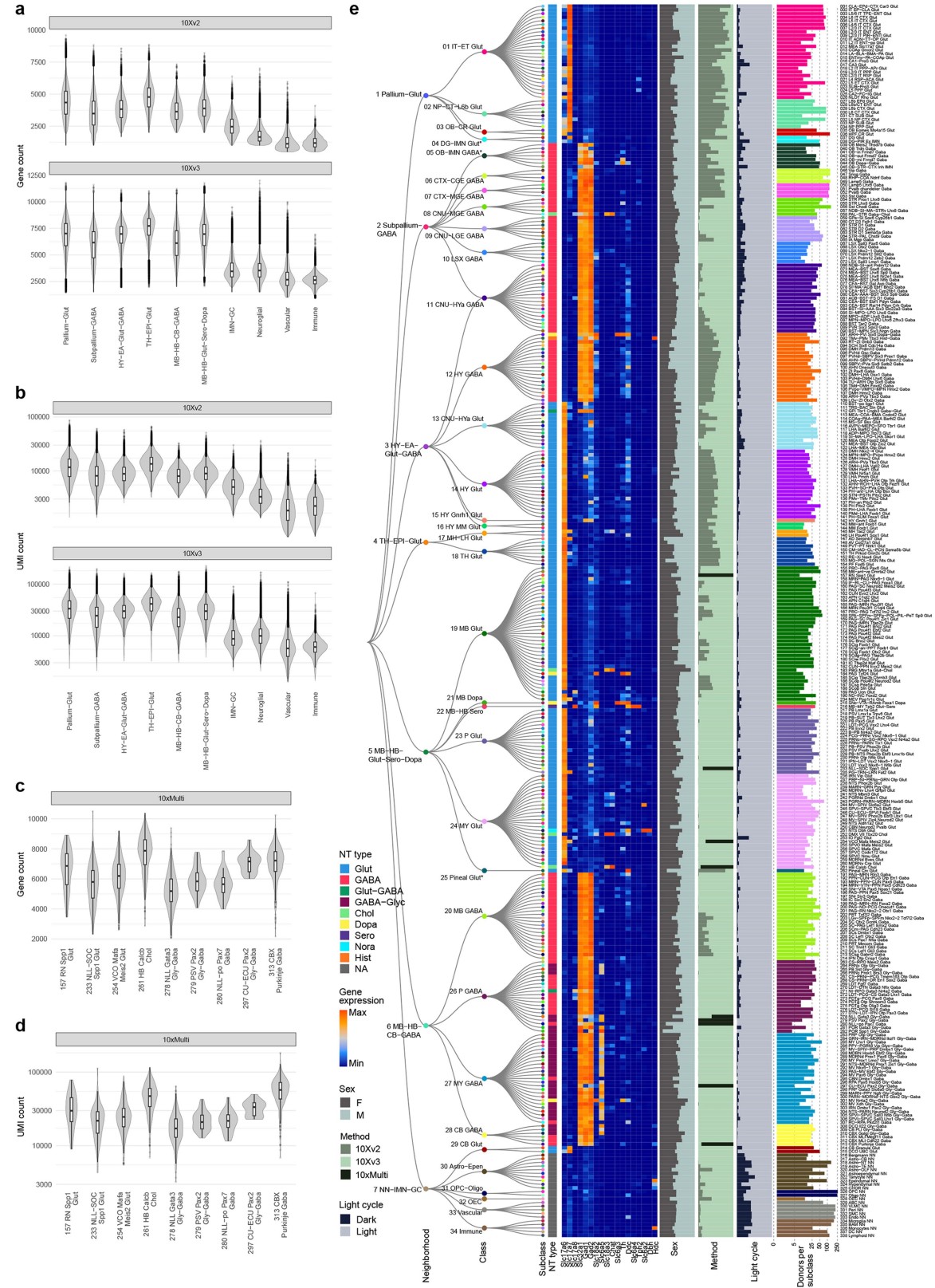

**Extended Data Fig. 5 |** See next page for caption.

**Extended Data Fig. 5 | Transcriptomic cell type taxonomy of the whole mouse brain with additional metadata information. (a-b)** Number of genes (a) or number of UMIs (b) detected per cell in 10xv2 (top) or 10xv3 (bottom) datasets for each cell type neighborhood. The data shown is post-QC. Numbers of cells for 10xv2: Pallium-Glut n = 1,128,664, Subpallium-GABA n = 269,307, HY-EA-Glut-GABA n = 107,706, TH-EPI-Glut n = 73,702, MB-HB-CB-GABA n = 18,590, MB-HB-Glut-Sero-Dopa n = 20,089, IMN-GC n = 123,650, Neuroglial n = 80,959, Vascular n = 6,894, Immune n = 4,941. Numbers of cells for 10xv3: Pallium-Glut n = 366,137, Subpallium-GABA n = 342,116, HY-EA-Glut-GABA n = 187,742, TH-EPI-Glut n = 52,469, MB-HB-CB-GABA n = 167,425, MB-HB-Glut-Sero-Dopa n = 159,653, IMN-GC n = 209,310, Neuroglial n = 774,537, Vascular n = 130,599, Immune n = 87,639. **(c-d)** Number of genes (c) or number of UMIs (d) detected per nucleus in the 10xMulti dataset (post-QC) for each subclass where 10xMulti nuclei were added. Numbers of nuclei: 157 RN Spp1 Glut n = 48, 233 NLL-SOC Spp1 Glut n = 115, 254 VCO Mafa Meis2 Glut n = 490, 261 HB Calcb Chol n = 274, 278 NLL Gata3 Gly-Gaba n = 339, 279 PSV Pax2 Gly-Gaba n = 43, 280 NLL-po Pax7 Gaba n = 17, 297 CU-ECU Pax2 Gly-Gaba n = 15, 313 CBX Purkinje Gaba n = 346. All box plots include the median line, the box denotes the interquartile range (IQR), whiskers denote the rest of the data distribution, and outliers are denoted by points greater than ±1.5× IQR. **(e)** The transcriptomic taxonomy tree of 338 subclasses organized in a dendrogram (same as Fig. 1a). From left to right, the bar plots represent neurotransmitter (NT) type assignment, heat map showing expression of major neurotransmitter marker genes, sex distribution, platform distribution, light-dark distribution of profiled cells, and number of donors that contributed to each subclass.

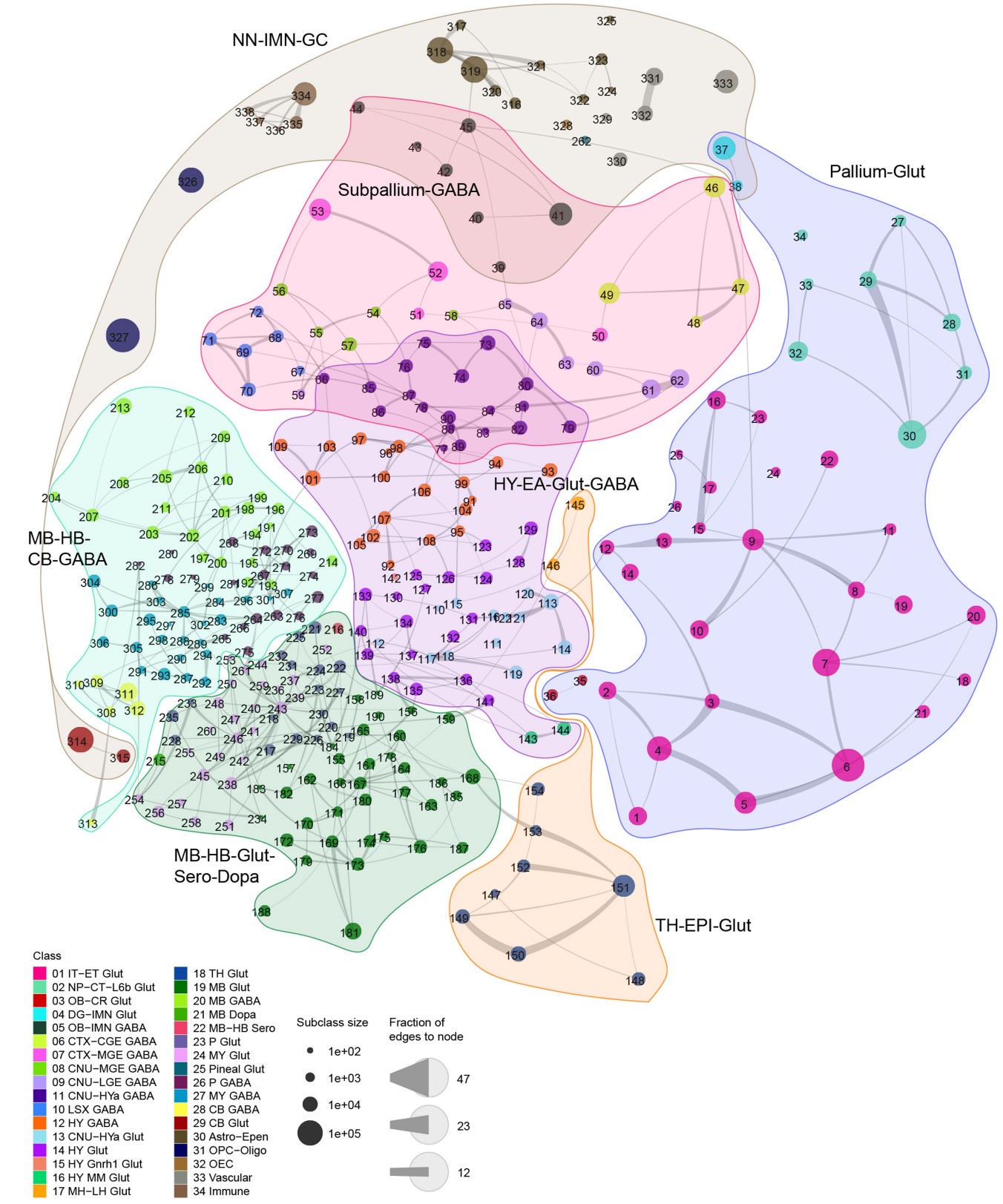

**Extended Data Fig. 6 | Constellation plot of the global relatedness between subclasses.** Each subclass is represented by a disk, labeled by the subclass ID and positioned at the subclass centroid in UMAP coordinates shown in Fig. 1c. The size of the disk corresponds to the number of cells within each subclass, and the edge weights correspond to the fraction of shared neighbors (see Methods) between subclasses. Each subclass is colored by the class it belongs to. Curved outlines drawn around subclasses show the major neighborhoods.

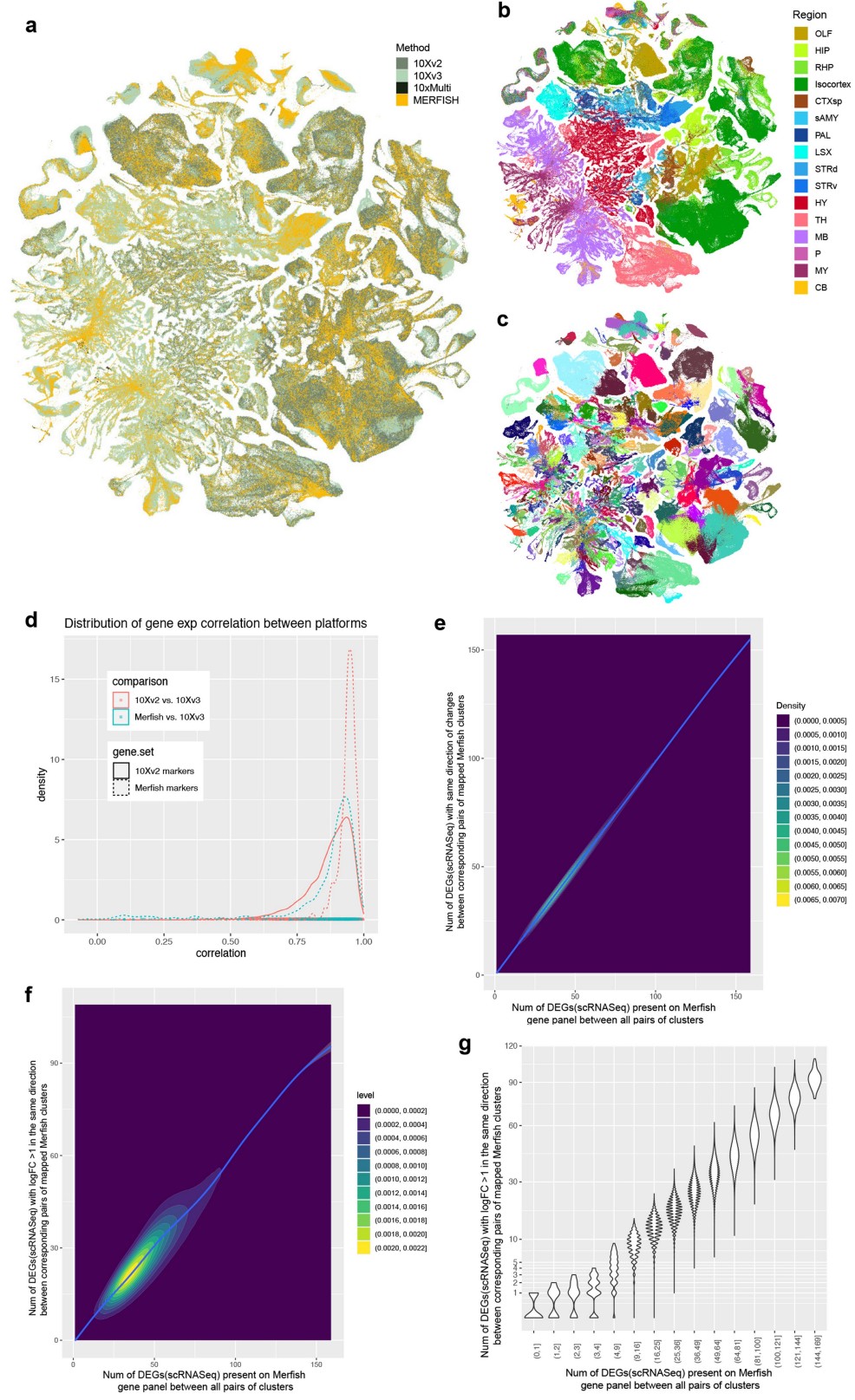

**Extended Data Fig. 7** | See next page for caption.

**Extended Data Fig. 7 | Validation of data integration across 10xv2, 10xv3, and MERFISH datasets.** (**a-c**) UMAP representation of all cell types colored by profiling platform (a), region (b), and subclass (c). Other than the regions only profiled by 10xv3 (LSX, STR, sAMY, PAL, Pons, MY), the cells from both 10xv2 and 10xv3 platforms integrate very well. Cell types in isocortex and HPF have a lot more 10xv2 cells, consistent with our sampling plan. For cell types/clusters containing many cells, we observed separation of 10xv2 and 10xv3 data in the UMAP space, but not at the cluster level. (**d**) Correlation of gene expression between 10xv2 and 10xv3 and between 10xv3 and MERFISH. For each gene, we computed the Pearson correlation of its average expression in each cluster across clusters between 10xv2 and 10xv3, and the correlation between 10xv3 and MERFISH. For 10xv3 and MERFISH comparison, distribution of the correlation values of all 500 genes in the MERFISH panel is shown. For 10xv3 and 10xv2 comparison, we show the correlation of 5383 marker genes based on 10xv2, and 466 10xv2 marker genes that are also present on the MERFISH gene panel (the other 34 MERFISH genes not shown have low expression in 10xv2 clusters). We manually inspected several genes with poor correlation and found them to have poor gene annotation or show relatively small variations across clusters. Most genes with low correlations between 10xv3 and MERFISH data are *Rik genes that are more likely to be poorly annotated, and the MERFISH probes selected for them might not work well. (**e**) 2D density plot showing on the X-axis the number of DEGs (based on 10xv3 dataset) present on the MERFISH gene panel between all pairs of clusters, and on the Y-axis the number of such DEGs showing the same direction of changes between corresponding pairs of mapped MERFISH clusters. Almost all the DEGs between all pairs of clusters show the same direction of changes between 10xv3 and MERFISH. (**f**) 2D density plot showing on the X-axis the number of DEGs (based on 10xv3 dataset) present on the MERFISH gene panel between all pairs of clusters, and on the Y-axis the number of such DEGs showing the same direction of changes, and $|\log_2(FC)| > 1$ between corresponding pairs of mapped MERFISH clusters. About 60% of DEGs between all pairs of clusters based on 10xv3 show significant fold change (FC) in MERFISH. (**g**) Similar analysis as in (f) but shown as violin plot by binning the number of 10xv3 DEGs present on the MERFISH gene panel on the X-axis, with better resolution on closely related pairs with four or fewer DEGs present on MERFISH gene panels. The MERFISH dataset can resolve the vast majority of clusters due to strong correlation of DEG expression between 10xv3 and MERFISH clusters. On the other hand, a few hundred pairs of clusters with fewer than two DEGs on the MERFISH gene panel remain unresolvable in the MERFISH data, and they are usually sibling clusters with indistinguishable spatial distribution.

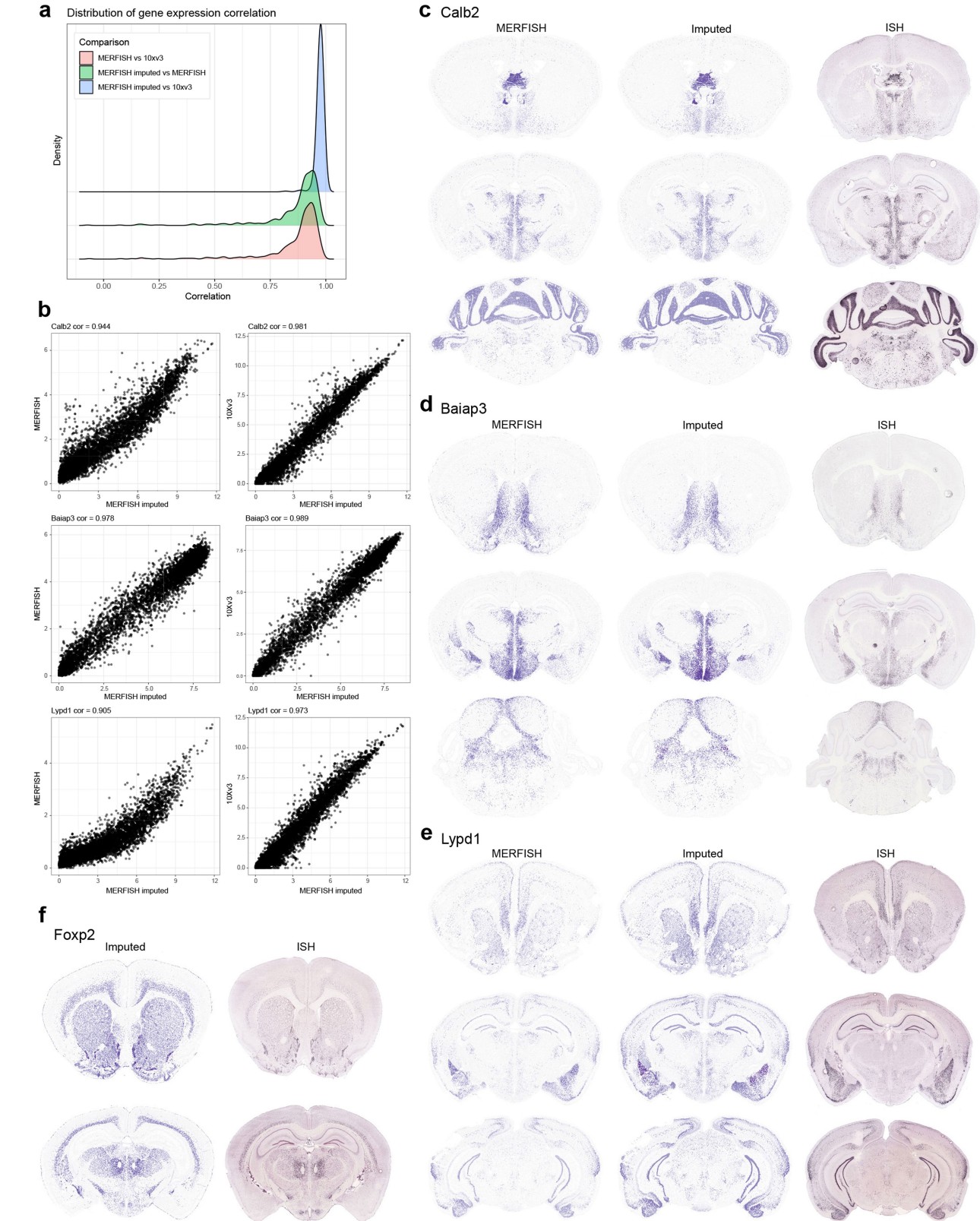

**Extended Data Fig. 8** | See next page for caption.

**Extended Data Fig. 8 | Validation of gene expression patterns of scRNA-seq transcriptomes imputed into MERFISH space.** (**a**) Correlation of expression for all 500 genes in the MERFISH panel between MERFISH and 10xv3 (red), imputed MERFISH and MERFISH (green), and imputed MERFISH and 10xv3 (blue). To test the accuracy of MERFISH imputation, one gene is excluded from the gene panel at a time from KNN computation at all levels and its imputed gene expression is compared with its original gene expression. The distribution of correlations between imputed expression and the original MERFISH expression or the reference 10xv3 expression is shown for each gene at the cluster level. (**b**) Scatterplots showing the correlation between imputed MERFISH gene expression vs. MERFISH gene expression (left panels) and imputed MERFISH gene expression vs. 10xv3 gene expression (right panels) for selected genes, *Calb2* (top row), *Baiap3* (middle row), and *Lypd1* (bottom row). (**c-e**) Examples of spatial gene expression patterns from MERFISH data (left panels), imputed MERFISH data (middle panels), and images from the Allen in situ hybridization (ISH) atlas (right panels) for select genes, *Calb2* (c), *Baiap3* (d), and *Lypd1* (e). (**f**) Representative MERFISH sections showing imputed expression of *Foxp2* (which was not directly profiled by MERFISH) and Allen ISH images. ISH image credit: Allen Institute for Brain Science, https://mouse. brain-map.org/.

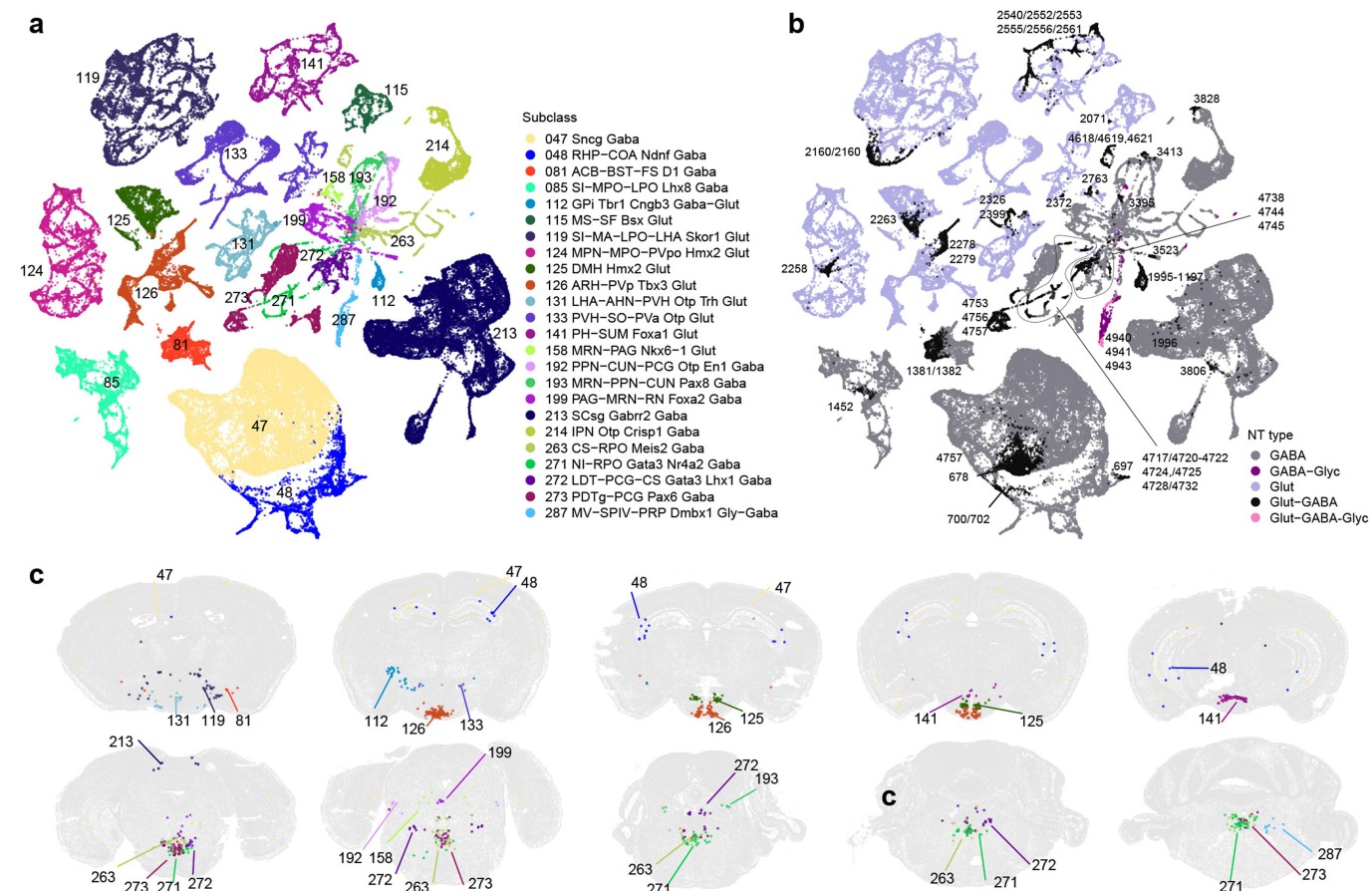

**Extended Data Fig. 9 | Distribution of glutamate-GABA dual transmitting cell types throughout the brain.** (**a-b**) Neuronal subclasses containing clusters releasing glutamate-GABA dual transmitters. UMAPs are colored by subclass (**a**) and neurotransmitter type (**b**). Glutamate-GABA co-releasing clusters are labeled by cluster ID in panel (**b**). (**c**) MERFISH sections showing glutamate-GABA co-releasing clusters colored by the subclass to which they belong. See Supplementary Table 7 for detailed neurotransmitter assignment for each cluster.

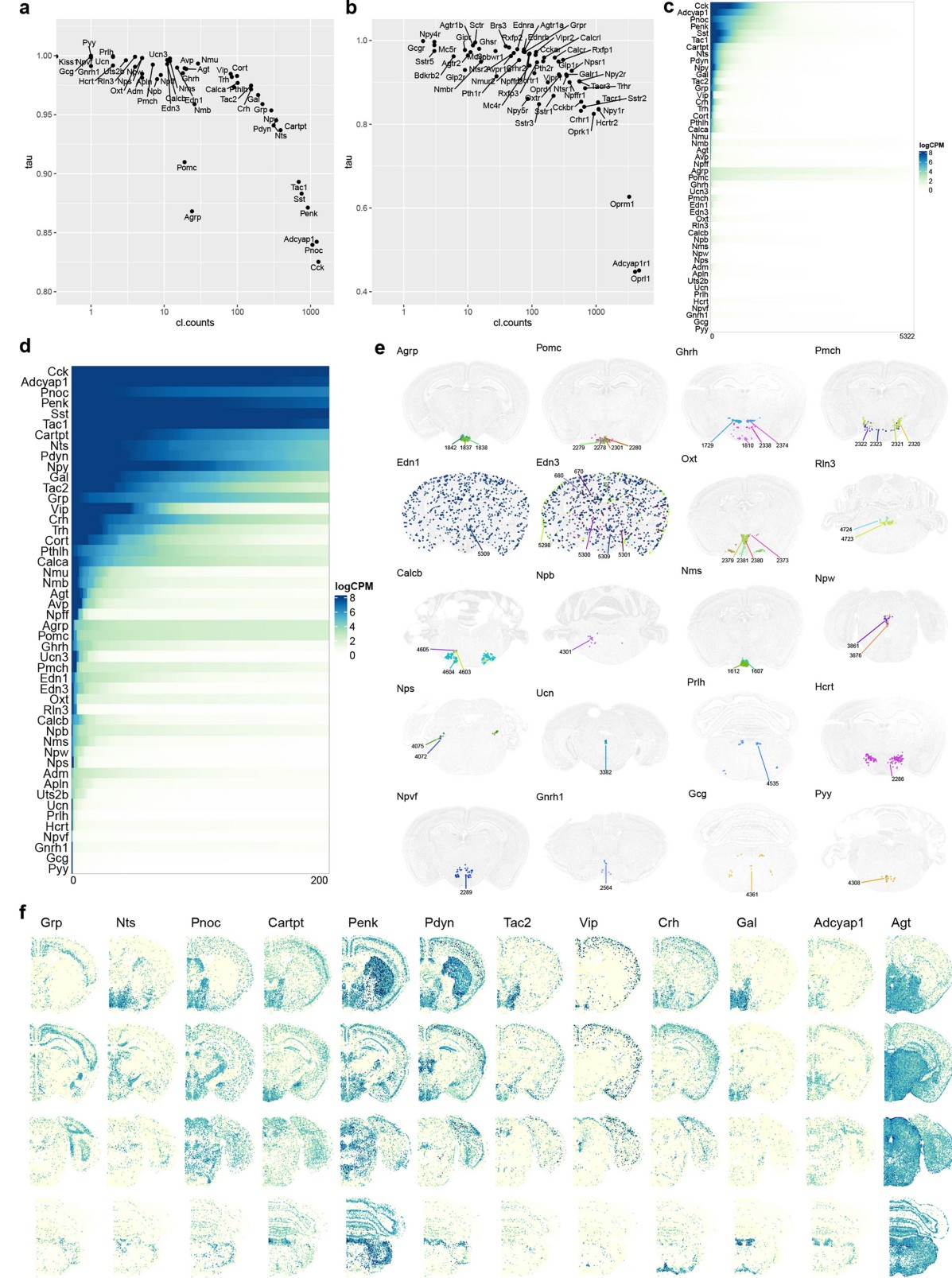

**Extended Data Fig. 10** | See next page for caption.

**Extended Data Fig. 10 | Neuropeptide distribution across the whole mouse brain.** (**a**) Scatter plot of Tau score over the number of clusters each neuropeptide is expressed in at the level of $\log_2(\text{CPM}) > 3$. The Tau score is a measurement of cell type specificity, which varies from 0 to 1 where 0 means uniformly expressed and 1 means highly specific to one type. (**b**) Scatter plot of Tau score over the number of clusters each peptide-liganded G-protein coupled receptor (GPCR) gene is expressed in at the level of $\log_2(\text{CPM}) > 3$. (**c**) Expression level of neuropeptide in $\log_2(\text{CPM})$ per cluster. For each neuropeptide along the Y axis, clusters are sorted from the highest to lowest mean gene expression level along the X axis. (**d**) Expression level of neuropeptide in $\log_2(\text{CPM})$ per cluster. For each neuropeptide along the Y axis, clusters are sorted from the highest to lowest mean gene expression level along the X axis. For each gene, only the top 200 highest-expressing clusters out of 5,322 clusters are shown. (**e**) Representative MERFISH sections highlighting the spatial location of clusters expressing each of the 20 highly cell-type-specific neuropeptide genes (expressed in 8 or fewer clusters). (**f**) Representative MERFISH sections showing the expression of the neuropeptides present on the MERFISH gene panel that are widely expressed. We also note that the relationships between mRNA levels, the post-translationally processed peptide levels, and the functional levels are unknown for most neuropeptides, thus, it is difficult to predict what mRNA levels would lead to sufficient functional expression of a given neuropeptide.

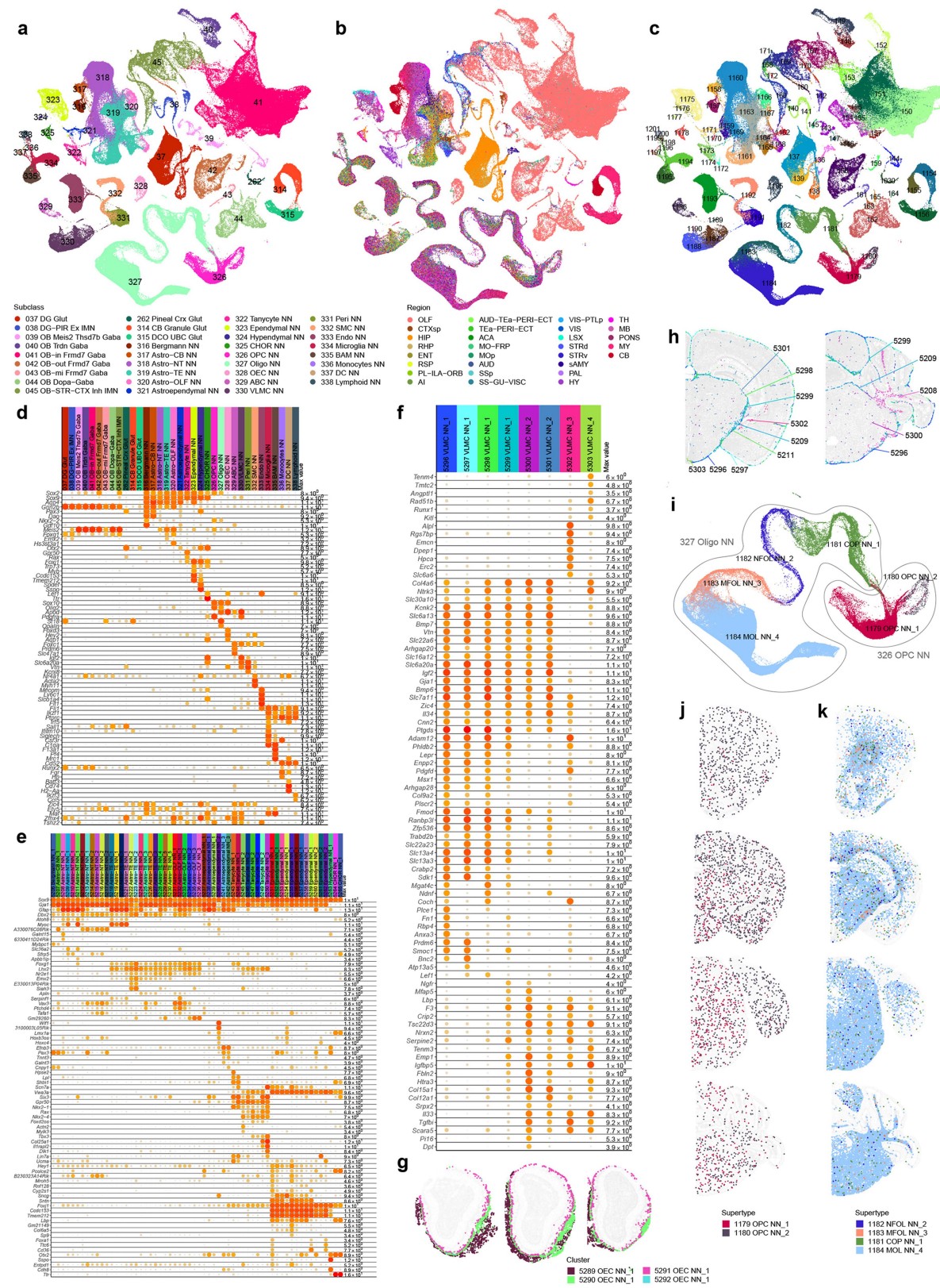

**Extended Data Fig. 11 | Additional non-neuronal UMAPs and marker genes.**
(**a-c**) UMAP representation of the NN-IMN-GC neighborhood colored by
subclass (a), region (b), and supertype (c). (**d**) Dot plot showing marker gene
expression in non-neuronal subclasses. Dot size and color indicate proportion
of expressing cells and average expression level in each subclass, respectively.
(**e**) Dot plot showing marker gene expression in all clusters in the Astro-Epen
class. Dot size and color indicate proportion of expressing cells and average
expression level in each cluster, respectively. (**f**) Dot plot showing marker gene
expression in VLMC clusters. Dot size and color indicate proportion of
expressing cells and average expression level in each cluster, respectively.
(**g**) Representative MERFISH sections showing the spatial gradient of OEC
clusters. (**h**) Co-localization of VLMCs with interlaminar astrocytes (ILA) as
shown in selected MERFISH sections. (**i**) UMAP representation of OPCs and
oligodendrocytes colored and labeled by supertype. (**j-k**) Representative
MERFISH sections showing the spatial distribution of OPC (j) and Oligo (k)
supertypes.

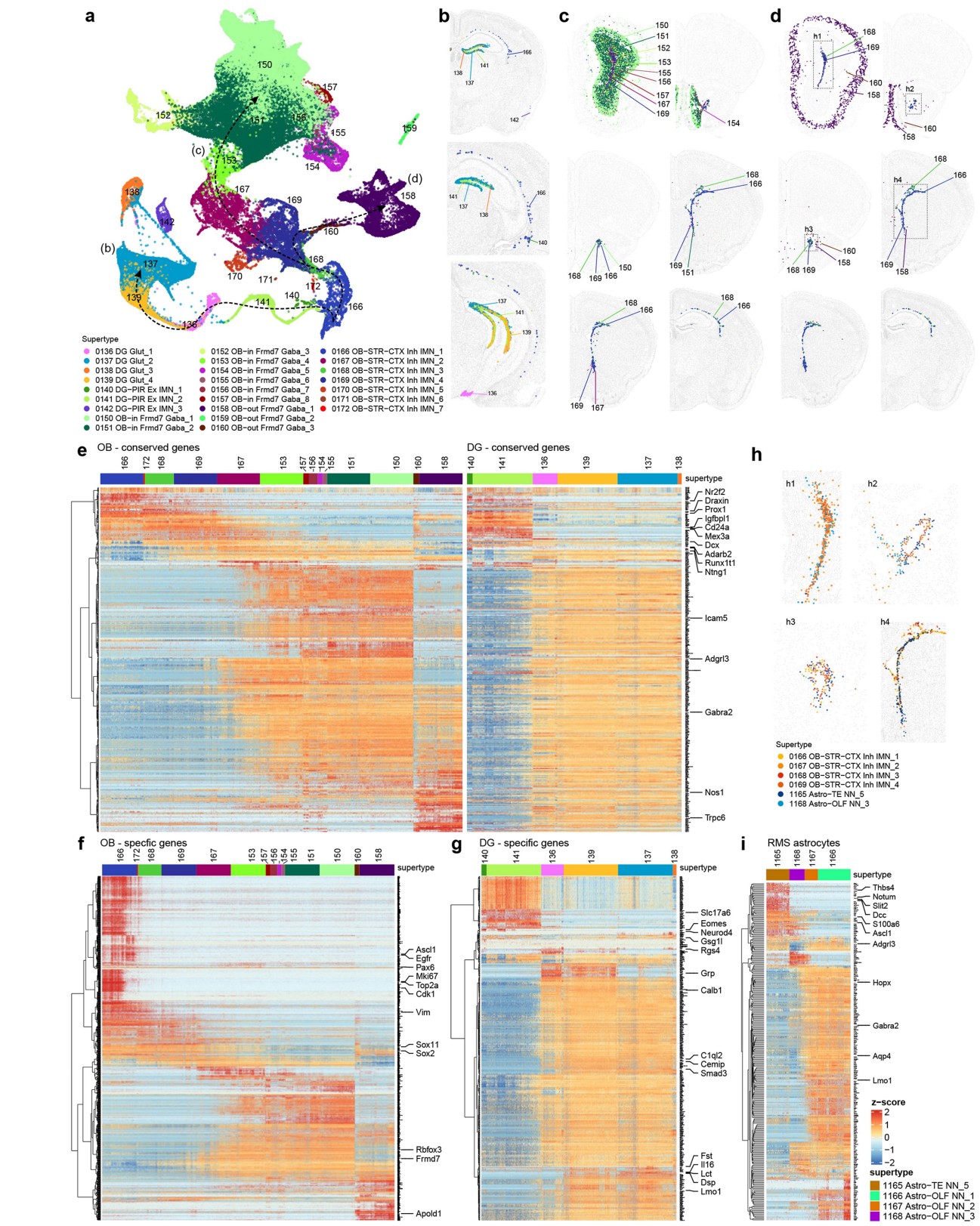

**Extended Data Fig. 12** | See next page for caption.

**Extended Data Fig. 12 | Gene expression patterns in immature neuronal populations and RMS astrocytes.** (**a**) UMAP representation of immature neuron populations colored by supertype. Maturation trajectories in dentate gyrus (DG) (b), inner olfactory bulb (c), and outer olfactory bulb (d) are highlighted. (**b**-**d**) Representative MERFISH sections showing location of immature neuronal supertypes from the three trajectories shown in (a). (**e**-**g**) Heatmap showing gene expression changes as immature neurons transition to mature cell types, conserved between OB (left) and DG (right) cell type development (e), specific to OB cell types (f), and specific to DG cell types (g). Key marker genes at each stage of development are highlighted. It seems, however, that the scRNA-seq data might not have captured all cell states along the DG maturation trajectory based on the gaps between clusters in the UMAP and absence of expression for genes like *Ascl1, Pax6*, *Top2a*, and *Mki67* along the DG trajectory. Various studies have tried to capture the transitional states between neural stem and neuronal progenitor cells in the DG with most making use of transgenic mice to isolate specific states[142,143]. (**h**) MERFISH sections showing the co-localization of immature neurons and astrocytes in the rostral migratory stream (RMS). The dashed boxes in (d) show the location of the highlighted regions in (h). (**i**) Heatmap showing gene expression changes in astrocytes associated with the RMS from SVZ to OB. Highlighted genes are conserved between the RMS-associated astrocytes and the OB trajectory.

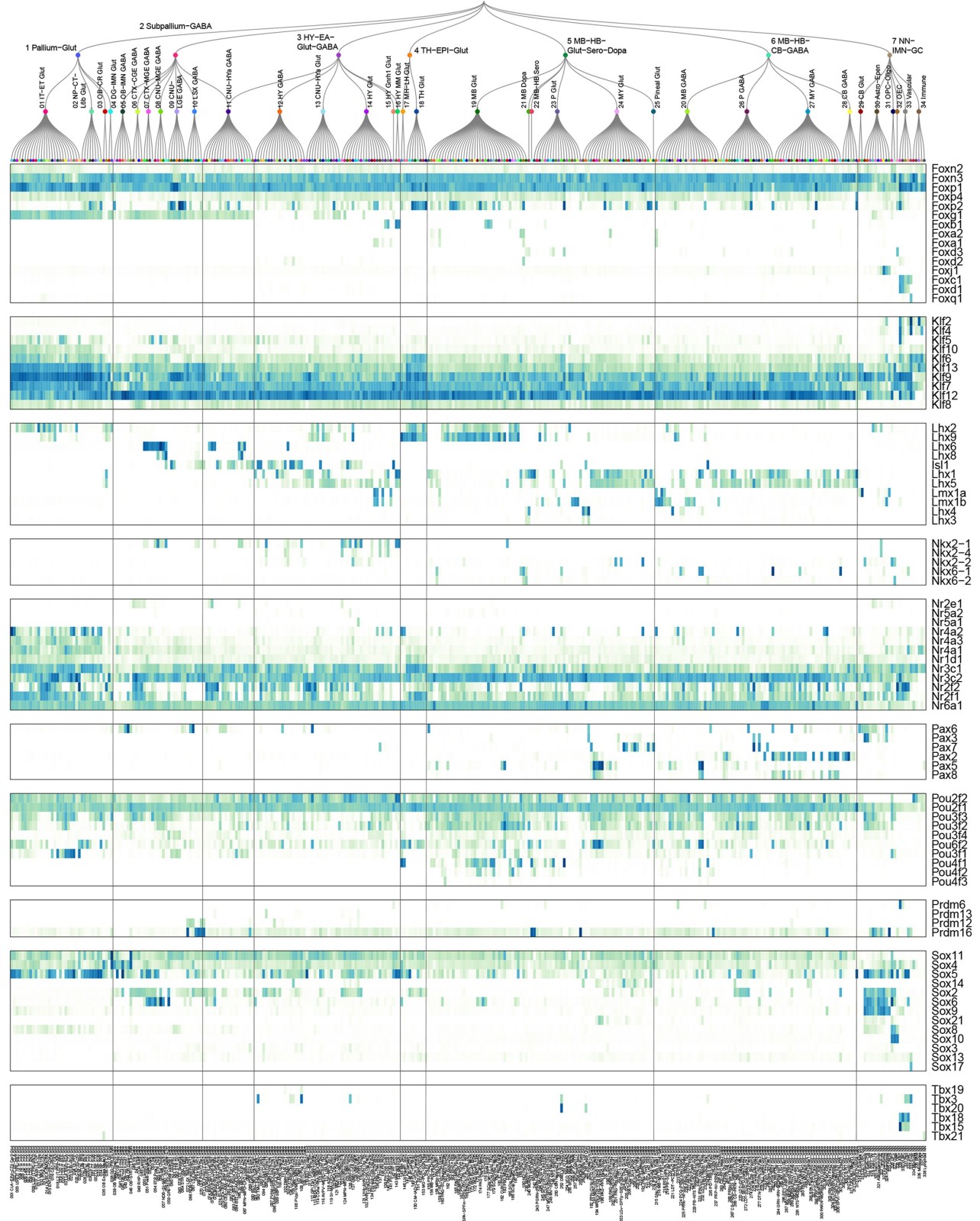

**Extended Data Fig. 13 | Transcription factor families.** Expression of key transcription factors for each subclass in the taxonomy tree, organized by transcription factor gene families. The lines divide the dendrogram into neighborhoods.

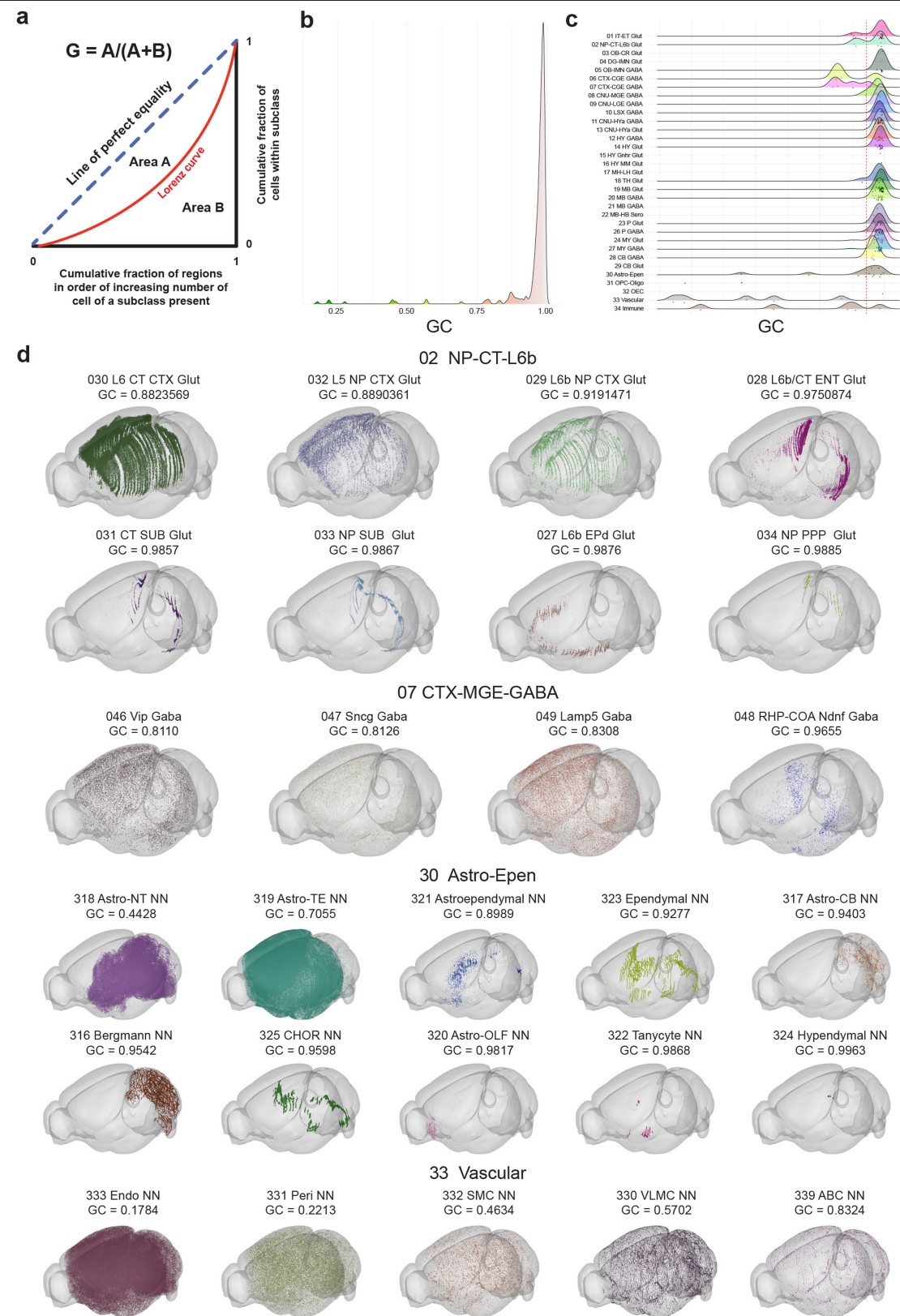

**Extended Data Fig. 14 | Distribution of Gini coefficient for subclasses.**
(**a**) Illustration explaining the concept of the Gini coefficient (GC). For each subclass, brain regions are ordered by number of cells present in x-axis. The y-axis is the cumulative fraction of cells for each subclass. The Gini coefficient is calculated by dividing the area (Area A) between the line of perfect equality and the observed distribution curve (the Lorenz curve) by the total area under the line of perfect equality (Areas A + B). The result is a value between 0 and 1 with 0 representing perfect equality and 1 maximum inequality. (**b**) Distribution of GCs for all subclasses. Color scheme is the same as used for Fig. 6a. (**c**) Ridge plot showing the distribution of GCs for subclasses grouped by class. (**d**) 3D example plots of subclasses for 4 classes (02 NP-CT-L6b, 07 CTX-MGE-GABA, 30 Astro-Epen, and 33 Vascular), illustrating the wide range of GCs present. Within each class, plots are ordered by GC from lowest to highest.

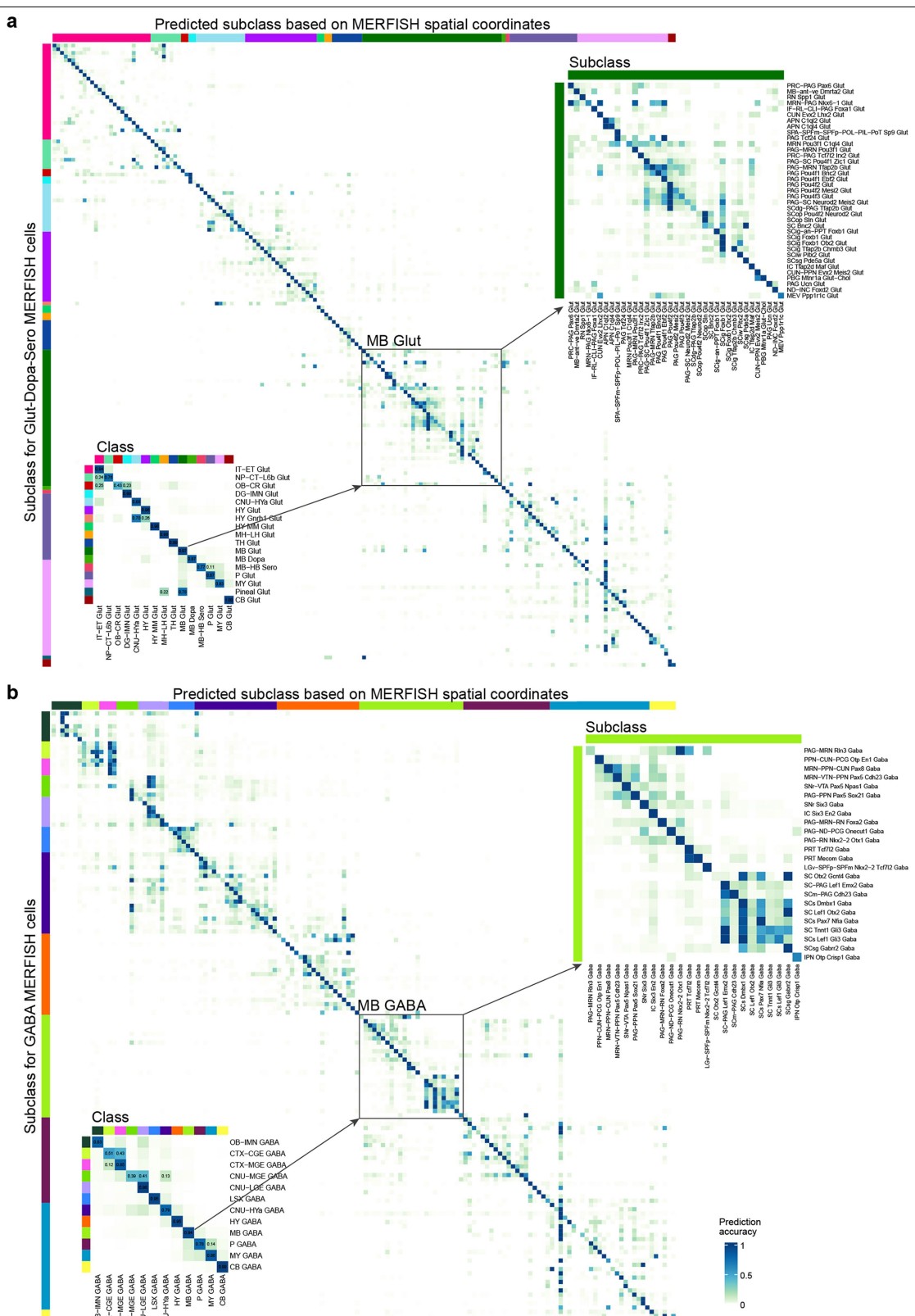

**Extended Data Fig. 15 | Predicting transcriptomic subclasses based on spatial location of MERFISH cells.** (**a**) Correspondence between assigned subclasses for MERFISH cells in glutamatergic, dopaminergic and serotonergic subclasses, and predicted subclasses based on the spatial coordinates of these MERFISH cells. Each row is normalized by dividing by the maximum number. Insert in the lower left corner shows the correspondence between assigned and predicted glutamatergic, dopaminergic, and serotonergic classes. Insert in the upper right corner highlights the correspondence between assigned and predicted subclasses in the MB Glut class. (**b**) Correspondence between assigned GABAergic subclasses and predicted subclasses based on the spatial coordinates of MERFISH cells. Insert in the lower left corner shows the correspondence between assigned and predicted GABAergic classes. Insert in the upper right corner highlights the correspondence between assigned and predicted subclasses in the MB GABA class.

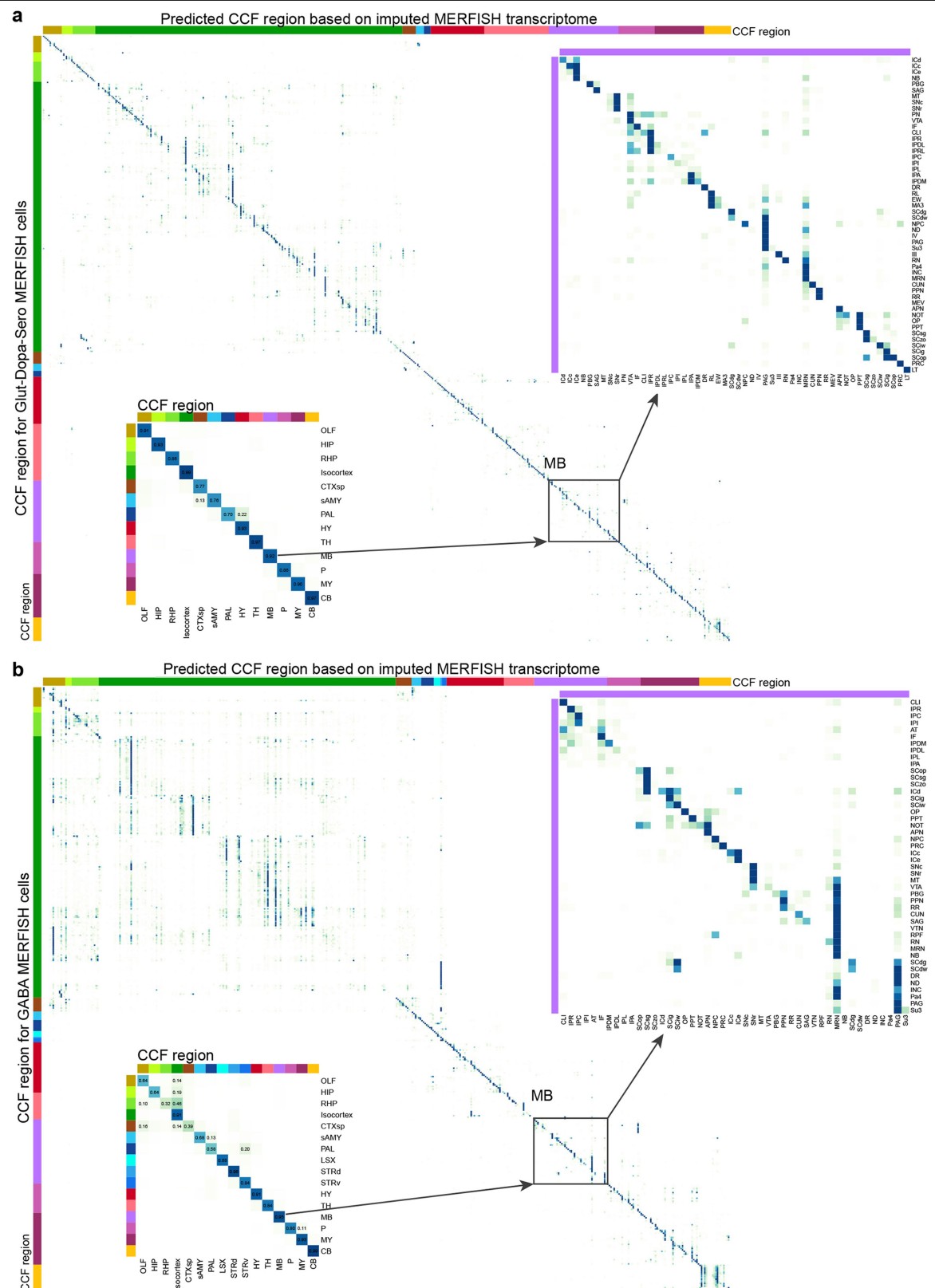

**Extended Data Fig. 16 | Predicting anatomical regions based on imputed MERFISH transcriptomes.** (**a**) Correspondence between assigned CCFv3 regions for MERFISH cells in glutamatergic, dopaminergic, and serotonergic cell types to predicted CCFv3 regions based on imputed transcriptomes of these MERFISH cells. Each row is normalized by dividing by the maximum number. Insert in the lower left corner shows the correspondence between assigned and predicted regions for glutamatergic, dopaminergic, and serotonergic cell types. Insert in the upper right corner highlights the correspondence between assigned and predicted subregions in midbrain. (**b**) Correspondence between assigned CCFv3 regions for MERFISH cells in GABAergic cell types and predicted CCFv3 regions based on imputed transcriptomes of these MERFISH cells. Insert in the lower left corner shows the correspondence between assigned and predicted regions for GABAergic cell types. Insert in the upper right corner highlights the correspondence between assigned and predicted subregions in midbrain.

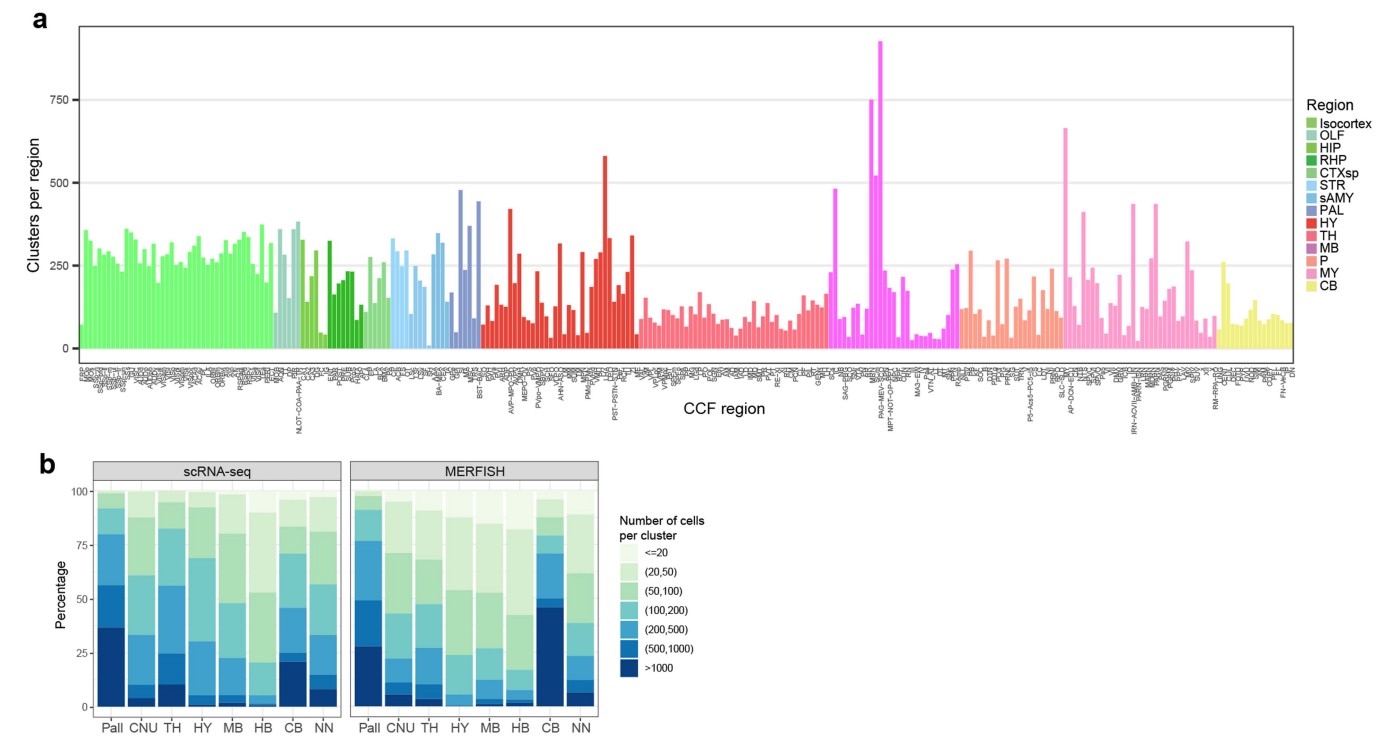

**Extended Data Fig. 17 | Distribution of cluster numbers and cluster sizes across different brain regions.** (**a**) Number of clusters per fine CCFv3 region (Supplementary Table 9) as analyzed using the MERFISH data. Bars are colored by broad CCFv3 regions. (**b**) Distribution of cluster size (number of cells per cluster) per major brain region in scRNA-seq data and MERFISH data.

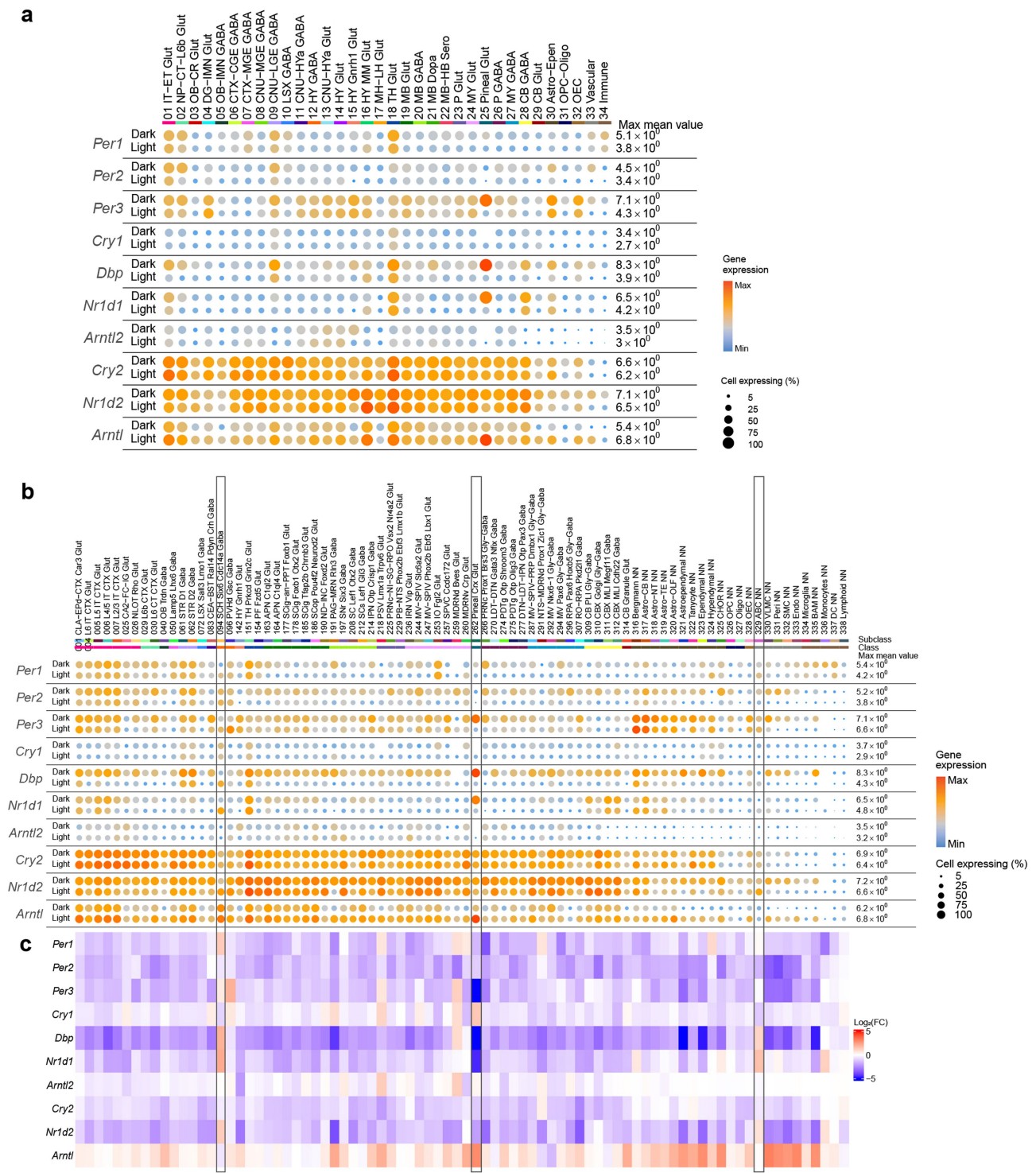

**Extended Data Fig. 18 | Circadian cycle associated expression changes in clock genes. (a-b)** Dot plots showing the expression of clock genes in light-phase and dark-phase cells within each cell class (a) or selected subclasses that have any clock genes with fold change |log₂(FC)| >1 between light and dark phases (b). Dot size and color indicate proportion of expressing cells and average expression level in each class or subclass, respectively. **(c)** Heatmap showing the log₂(FC) difference between light and dark phases for clock genes in the selected subclasses as in (b).

**Extended Data Table 1 | Summary of the whole mouse brain cell type atlas**

| Cell Class | No. of Subclasses | No. of Supertypes | No. of Clusters | Neighborhood |
|---|---|---|---|---|
| 01 IT-ET Glut | 26 | 102 | 402 | Pallium-Glut |
| 02 NP-CT-L6b Glut | 8 | 27 | 83 | Pallium-Glut |
| 03 OB-CR Glut | 2 | 6 | 16 | Pallium-Glut |
| 04 DG-IMN Glut | 2 | 7 | 16 | Pallium-Glut; NN-IMN-GC |
| 05 OB-IMN GABA | 7 | 30 | 105 | Subpallium-GABA; NN-IMN-GC |
| 06 CTX-CGE GABA | 4 | 30 | 101 | Subpallium-GABA |
| 07 CTX-MGE GABA | 4 | 30 | 105 | Subpallium-GABA |
| 08 CNU-MGE GABA | 5 | 27 | 105 | Subpallium-GABA |
| 09 CNU-LGE GABA | 7 | 35 | 94 | Subpallium-GABA |
| 10 LSX GABA | 6 | 38 | 146 | Subpallium-GABA |
| 11 CNU-HYa GABA | 19 | 94 | 395 | Subpallium-GABA; HY-EA-Glut-GABA |
| 12 HY GABA | 19 | 74 | 420 | HY-EA-Glut-GABA |
| 13 CNU-HYa Glut | 13 | 43 | 236 | HY-EA-Glut-GABA |
| 14 HY Glut | 19 | 84 | 339 | HY-EA-Glut-GABA |
| 15 HY Gnrh1 Glut | 1 | 1 | 1 | HY-EA-Glut-GABA |
| 16 HY MM Glut | 2 | 3 | 13 | HY-EA-Glut-GABA |
| 17 MH-LH Glut | 2 | 9 | 35 | TH-EPI-Glut |
| 18 TH Glut | 8 | 35 | 113 | TH-EPI-Glut |
| 19 MB Glut | 37 | 116 | 661 | MB-HB-Glut-Sero-Dopa |
| 20 MB GABA | 24 | 89 | 451 | MB-HB-CB-GABA |
| 21 MB Dopa | 1 | 8 | 43 | MB-HB-Glut-Sero-Dopa |
| 22 MB-HB Sero | 1 | 7 | 32 | MB-HB-Glut-Sero-Dopa |
| 23 P Glut | 18 | 58 | 288 | MB-HB-Glut-Sero-Dopa |
| 24 MY Glut | 26 | 76 | 406 | MB-HB-Glut-Sero-Dopa |
| 25 Pineal Glut | 1 | 1 | 1 | MB-HB-Glut-Sero-Dopa; NN-IMN-GC |
| 26 P GABA | 20 | 52 | 206 | MB-HB-CB-GABA |
| 27 MY GABA | 25 | 58 | 353 | MB-HB-CB-GABA |
| 28 CB GABA | 6 | 13 | 30 | MB-HB-CB-GABA |
| 29 CB Glut | 2 | 3 | 9 | NN-IMN-GC |
| 30 Astro-Epen | 10 | 22 | 60 | NN-IMN-GC |
| 31 OPC-Oligo | 2 | 6 | 23 | NN-IMN-GC |
| 32 OEC | 1 | 1 | 4 | NN-IMN-GC |
| 33 Vascular | 5 | 8 | 19 | NN-IMN-GC |
| 34 Immune | 5 | 8 | 11 | NN-IMN-GC |

The numbers of subclasses, supertypes, and clusters in each class, as well as the neighborhood(s) each class is assigned to, are listed. Classes are color coded consistently with the taxonomy.

# Reporting Summary

## Statistics

For all statistical analyses, confirm that the following items are present in the figure legend, table legend, main text, or Methods section.

| n/a | Confirmed | |
|---|---|---|
| ☐ | ☒ | The exact sample size (*n*) for each experimental group/condition, given as a discrete number and unit of measurement |
| ☐ | ☒ | A statement on whether measurements were taken from distinct samples or whether the same sample was measured repeatedly |
| ☒ | ☐ | The statistical test(s) used AND whether they are one- or two-sided<br>*Only common tests should be described solely by name; describe more complex techniques in the Methods section.* |
| ☒ | ☐ | A description of all covariates tested |
| ☒ | ☐ | A description of any assumptions or corrections, such as tests of normality and adjustment for multiple comparisons |
| ☐ | ☒ | A full description of the statistical parameters including central tendency (e.g. means) or other basic estimates (e.g. regression coefficient) AND variation (e.g. standard deviation) or associated estimates of uncertainty (e.g. confidence intervals) |
| ☒ | ☐ | For null hypothesis testing, the test statistic (e.g. *F*, *t*, *r*) with confidence intervals, effect sizes, degrees of freedom and *P* value noted<br>*Give P values as exact values whenever suitable.* |
| ☒ | ☐ | For Bayesian analysis, information on the choice of priors and Markov chain Monte Carlo settings |
| ☒ | ☐ | For hierarchical and complex designs, identification of the appropriate level for tests and full reporting of outcomes |
| ☒ | ☐ | Estimates of effect sizes (e.g. Cohen's *d*, Pearson's *r*), indicating how they were calculated |

*Our web collection on statistics for biologists contains articles on many of the points above.*

## Software and code

Policy information about availability of computer code

| Data collection | scRNA-seq data was collected using the 10x Genomics Chromium Single Cell 3' v2 and v3 kits and the Chromium Next GEM Single Cell Multiome ATAC + Gene Expression Reagent Bundle. Sequencing was performed on the Illumina HiSeq2500 and NovaSeq6000.<br>Flow cytometry data were acquired on a BD FACSAriaII running FACSdiva v8.<br>MERFISH data were acquired using the Vizgen MERSCOPE platform. |
|---|---|

| Data analysis | Raw scRNA-seq fastq files were processed with the CellRanger v6.0.1 pipeline. Raw 10xMultiome fastq files were processed using the 10x Genomics CellRanger Arc (v2.0) workflow with default parameters. |
|---|---|
| | Doublets were identified using a modified version of the DoubletFinder algorithm which is available in scrattch.hicat (https://github.com/AllenInstitute/scrattch.hicat, v1.0.9). |
| | scRNA-seq clustering and differential gene expression analysis was performed in R (v4.1.3) using the scrattch.bigcat package (https://github.com/AllenInstitute/scrattch.bigcat, v0.0.5), which also contains many functions to visualize the data together with the scrattch.vis package (https://github.com/AllenInstitute/scrattch.vis, v0.0210). Scrattch.bigcat adopted the parquet file format for storing sparse matrix, which allows for manipulation of matrices that are too large to fit in memory through memory mapping to files on disk. The whole gene count matrices were chunked to smaller parquet files with bin size of 50,000 for cells, and 500 for genes, which could be loaded efficiently and concurrently using the arrow package (v12.0.1, (https://github.com/apache/arrow/, https://arrow.apache.org/docs/r/). For joint clustering of 10xv2 and 10xv3 data, we used BiocNeighbor package (v1.16.0, https://github.com/LTLA/BiocNeighbors) for computing KNN using Euclidean distance within modality and Cosine distance across modality using the Annoy algorithm (v1.17.1, https://github.com/spotify/annoy). |
| | Raw MERSCOPE data were decoded using Vizgen software (v231). Mapping of MERFISH data to the scRNASeq taxonomy was performed using the scrattch-mapping package (v0.2, https://github.com/AllenInstitute/scrattch.mapping). |
| | To visualize the scRNA-seq data and MERFISH data we used the single-cell data visualization tool cirrocumulus (v1.1.56, https://cirrocumulus.readthedocs.io/en/latest/). |

For manuscripts utilizing custom algorithms or software that are central to the research but not yet described in published literature, software must be made available to editors and reviewers. We strongly encourage code deposition in a community repository (e.g. GitHub). See the Nature Portfolio guidelines for submitting code & software for further information.

# Data

Policy information about availability of data

All manuscripts must include a data availability statement. This statement should provide the following information, where applicable:

- Accession codes, unique identifiers, or web links for publicly available datasets
- A description of any restrictions on data availability
- For clinical datasets or third party data, please ensure that the statement adheres to our policy

The scRNA-seq FASTQ files were deposited in the NeMO archive and are available under accession https://assets.nemoarchive.org/dat-qg7n1b0. The MERFISH raw data are is available at Brain Image Library (BIL) under DOI https://doi.org/10.35077/g.610. Instruction for access of the processed 10X data is available at https://github.com/AllenInstitute/abc_atlas_access/blob/main/descriptions/WMB-10X.md, and instruction for access of the processed MERFISH data is available at https://github.com/AllenInstitute/abc_atlas_access/blob/main/descriptions/MERFISH-C57BL6J-638850.md

# Human research participants

Policy information about studies involving human research participants and Sex and Gender in Research.

| Reporting on sex and gender | N/A. This study does not involve human research participants. |
|---|---|
| Population characteristics | N/A |
| Recruitment | N/A |
| Ethics oversight | N/A |

Note that full information on the approval of the study protocol must also be provided in the manuscript.

# Field-specific reporting

Please select the one below that is the best fit for your research. If you are not sure, read the appropriate sections before making your selection.

☒ Life sciences          ☐ Behavioural & social sciences          ☐ Ecological, evolutionary & environmental sciences

For a reference copy of the document with all sections, see nature.com/documents/nr-reporting-summary-flat.pdf

# Life sciences study design

All studies must disclose on these points even when the disclosure is negative.

| Sample size | Sample size calculations were not performed. For scRNA-seq data, sample size was determined based on extensive prior experience using these technologies in brain. Post-hoc analysis of clustering shows that it is robust to downsampling, which indicates that the sample size is sufficient to distinguish cell types as described in the manuscript. |
|---|---|
| | For MERFISH data we evenly sampled a mouse brain to capture most brain regions and get a even distribution of cells types throughout the brain. We compared the results from this mouse with those from Zhang et al companion paper and found the results are consistent. |
| Data exclusions | Cells from scRNA-seq data underwent stringent QC and cell quality was assessed based on gene detection, qc score, and doublet score.The qc score was calculated by summing the log transformed expression of a set of genes whose expression level is decreased significantly in poor |

quality cells (full list of genes in Supplementary Table 4). We used this qc score to quantify the integrity of cytoplasmic mRNA content, which tended to show bimodal distribution. Cells at the low end were very similar to single nuclei, which we removed for downstream analysis. Doublets were identified using a modified version of the DoubletFinder algorithm172 (available in scrattch.hicat, https://github.com/AllenInstitute/scrattch.hicat, v1.0.9) and removed when doublet score > 0.3. Threshold parameters (qc score and gene counts) and number of cells filtered are summarized in Supplementary Table 4. For example, for neurons (excluding granule cells) we used gene counts cutoff of 2,000 and qc score cutoff of 200.

Post-clustering we excluded noise clusters which are clusters with significantly lower gene detection due to extensive drop out, and clusters due to doublets or contamination. We first identified doublet clusters based on the co-expression of any pair of broad class marker genes using find_doublet_by_marker function in scrattch.bigcat package. To identify other doublet clusters, we searched for triplets of clusters A, B and C, wherein A was the putative doublet cluster, such that up-regulated genes of A relative to B largely overlapped with up-regulated genes in C relative to B, and up-regulated genes in A relative to C largely overlapped with up-regulated genes of B relative to C. After removing all doublet clusters, we then identified clusters with lower gene detection. To do that, we identified pairs of clusters such that one cluster with at least 50% fewer UMIs or >100 lower QC score, smaller size, and no more than one up-regulated gene relative to another cluster was identified as the low-quality cluster.

For MERFISH data the cell-by-gene table containing segmented cells was filtered to keep cells with a volume > 100 μm3 and < 3,000 μm3, that have at least 15 genes detected and contain a minimum of 40 but no more than 3,000 mRNA molecules to remove low quality cells and doublets that are outside of these ranges.

| | |
|---|---|
| Replication | Most of the scRNA-seq experiments were carried out at least twice independently and at least 2 mice and multiple brain dissections were used. For MERFISH, our collaborator lab generated three additional datasets with additional genes tested (see Zhang et al companion paper). All replicates have been included in the study. |
| Randomization | Randomization of animals to different groups is not relevant to our study design. There were no experimental vs. control groups. |
| Blinding | Prior to clustering, single cell transcriptomes were analyzed for previously known marker genes and were segregated into large groups: non-neuronal, glutamatergic and GABAergic. Clustering was then performed blind to the cell source or any other metadata that could reveal sample identity. |

# Reporting for specific materials, systems and methods

We require information from authors about some types of materials, experimental systems and methods used in many studies. Here, indicate whether each material, system or method listed is relevant to your study. If you are not sure if a list item applies to your research, read the appropriate section before selecting a response.

## Materials & experimental systems

| n/a | Involved in the study |
|---|---|
| ☒ | Antibodies |
| ☒ | Eukaryotic cell lines |
| ☒ | Palaeontology and archaeology |
| ☐ | ☒ Animals and other organisms |
| ☒ | Clinical data |
| ☒ | Dual use research of concern |

## Methods

| n/a | Involved in the study |
|---|---|
| ☒ | ChIP-seq |
| ☐ | ☒ Flow cytometry |
| ☒ | MRI-based neuroimaging |

## Animals and other research organisms

Policy information about studies involving animals; ARRIVE guidelines recommended for reporting animal research, and Sex and Gender in Research

| | |
|---|---|
| Laboratory animals | All 317 animals used in this study were house mice (Mus musculus) maintained on the C57BL/6J background. Animals were euthanized at P53-59 (n = 141), P50-52 (n = 3), or P60-71 (n = 173). Each animal's unique ID, sex, age, and genotype are listed in Supplementary Table 2. Mice had ad libitum access to food and water and were group-housed within a temperature- (21-22°C), humidity- (40-51%), and light- (14/10 hr light/dark cycle, or 12/12 hr reversed light/dark cycle) controlled room within the vivariums of the Allen Institute for Brain Science. |
| Wild animals | This study did not involve wild animals. |
| Reporting on sex | For each brain region, both male and female mice were used to collect scRNA-seq data. Though sex-balancing was successful at the level of brain region, after clustering of the data we identified a small number clusters that were either sex-dominant or sex-specific. These are described in the manuscript. |
| Field-collected samples | This study did not involve field-collected samples. |
| Ethics oversight | All experimental procedures related to the use of mice were approved by the Institutional Animal Care and Use Committee of the Allen Institute for Brain Science, in accordance with NIH guidelines. |

Note that full information on the approval of the study protocol must also be provided in the manuscript.

# Flow Cytometry

## Plots

Confirm that:

☐ The axis labels state the marker and fluorochrome used (e.g. CD4-FITC).

☐ The axis scales are clearly visible. Include numbers along axes only for bottom left plot of group (a 'group' is an analysis of identical markers).

☐ All plots are contour plots with outliers or pseudocolor plots.

☐ A numerical value for number of cells or percentage (with statistics) is provided.

## Methodology

**Sample preparation**

Sample preparation was done according to protocols: Allen Institute for Brain Science 2021. Slice Preparation with Tissue Dissociation - Mouse Protocol. protocols.io https://dx.doi.org/10.17504/protocols.io.bq6wmzfe and Allen Institute for Brain Science 2020. FACS Single Cell Sorting. protocols.io https://dx.doi.org/10.17504/protocols.io.be4cjgsw.

We used the Allen Mouse Brain Common Coordinate Framework version 3 (CCFv3; RRID: SCR_002978) ontology65 (http://atlas.brain-map.org/, Supplementary Table 1) to define brain regions for profiling and boundaries for dissection. We covered all regions of the brain using sampling at top-ontology level with judicious joining of neighboring regions (Supplementary Table 3, Extended Data Figure 1d-e). These choices were guided by the fact that microdissections of small regions were difficult. Therefore, joint dissection of neighboring regions was sometimes necessary to obtain sufficient numbers of cells for profiling. Comparison with subsequently generated MERFISH data showed that our CCF-based microdissections were largely accurate at cell subclass and major brain region levels (Extended Data Figure 2h).

Single cells were isolated by adapting previously described procedures. The brain was dissected, submerged in ACSF, embedded in 2% agarose, and sliced into 350-μm coronal sections on a compresstome (Precisionary Instruments). Block-face images were captured during slicing. Regions of interest (ROIs) were then microdissected from the slices and dissociated into single cells as previously described. Fluorescent images of each slice before and after ROI dissection were taken at the dissection microscope. These images were used to document the precise location of the ROIs using annotated coronal plates of CCFv3 as reference.

Dissected tissue pieces were digested with 30 U/ml papain (Worthington PAP2) in ACSF for 30 minutes at 30°C. Due to the short incubation period in a dry oven, we set the oven temperature to 35°C to compensate for the indirect heat exchange, with a target solution temperature of 30°C. Enzymatic digestion was quenched by exchanging the papain solution three times with quenching buffer (ACSF with 1% FBS and 0.2% BSA). Samples were incubated on ice for 5 minutes before trituration. The tissue pieces in the quenching buffer were triturated through a fire-polished pipette with 600-μm diameter opening approximately 20 times. The tissue pieces were allowed to settle and the supernatant, which now contained suspended single cells, was transferred to a new tube. Fresh quenching buffer was added to the settled tissue pieces, and trituration and supernatant transfer were repeated using 300-μm and 150-μm fire polished pipettes. The single cell suspension was passed through a 70-μm filter into a 15-ml conical tube with 500 μl of high BSA buffer (ACSF with 1% FBS and 1% BSA) at the bottom to help cushion the cells during centrifugation at 100 x g in a swinging bucket centrifuge for 10 minutes. The supernatant was discarded, and the cell pellet was resuspended in the quenching buffer. We collected 1,508,284 cells without performing FACS. The concentration of the resuspended cells was quantified, and cells were immediately loaded onto the 10x Genomics Chromium controller.

To enrich for neurons or live cells, cells were collected by fluorescence-activated cell sorting (FACS, BD Aria II running FACSdiva v8) using a 130-μm nozzle. Cells were prepared for sorting by passing the suspension through a 70-μm filter and adding Hoechst or DAPI (to a final concentration of 2 ng/ml). Sorting strategy was as previously described (Tasic et al 2018), with most cells collected using the tdTomato-positive label. 30,000 cells were sorted within 10 minutes into a tube containing 500 μl of quenching buffer. We found that sorting more cells into one tube diluted the ACSF in the collection buffer, causing cell death. We also observed decreased cell viability for longer sorts.

**Instrument**

FACSAria II or FACSAria Fusion

**Software**

FACSDiva v8

**Cell population abundance**

Abundance of RFP+ cell populations for 10x genomics scRNA-seq were determined by hemocytometer post-FACS.

**Gating strategy**

The morphology gate (SSC-A vs FSC-A) here includes all events that pass FSC threshold to allow profiling of all possible RFP+ cells. SC-FSC and SC-SSC are used to exclude doublets, and RFP+ cells are sorted from the rest of the cell based on the RFP+ DAPI- phenotype. Gating strategy for RFP+ mouse neurons with an example figure is described in more detail here: Allen Institute for Brain Science 2020. FACS Single Cell Sorting. protocols.io https://dx.doi.org/10.17504/protocols.io.be4cjgsw. Because of this, a figure exemplifying the gating strategy is not provided in the SI of the manuscript.

☐ Tick this box to confirm that a figure exemplifying the gating strategy is provided in the Supplementary Information.

