## [Peer Review File · Nature]

Manuscript Title: A high-resolution transcriptomic and spatial atlas of cell types in the whole mouse brain

Editorial Notes:

Redactions – Third Party Material

Reviewer Comments & Author Rebuttals

Reviewer Reports on the Initial Version:

Referees' comments:

Referee #1:

Remarks to the Author:

The manuscript "A high-resolution transcriptomic and spatial atlas of cell types in the whole mouse brain" by Yao and colleagues reports on the generation and initial analysis of cell type and spatial transcriptomic atlases from the entire adult mouse brain. The authors identify 5,200 high-quality clusters (scRNAseq), map their spatial distribution in the brain (MERFISH), and describe more in depth the expression of neurotransmitters, neuropeptides and transcription factors (TFs) to infer principles of cell types diversity and spatial distribution in the brain.

The work described here is truly a herculean effort. It entailed sequencing about 7 million single cells, MERFISH experiments on ~50 brain sections, and development of new computational tools to handle these large datasets. For comparison, two highly-cited studies describing mouse brain cell type atlases, Zeisel et al Cell 2018 and Saunders et al Cell 2018, sequenced ~500,000 and ~690,000 cells, respectively.

Notably, the experimental design included sampling from males and females, to discover sex-specific clusters, and from animals in different phases of the circadian cycle. The analysis pipeline included stringent quality controls (QC) to filter out low-quality cells; these QC criteria are more stringent than those commonly used in the field.

Furthermore, the manuscript describes biological insights emerging from the analysis of these data. Figure 5 includes a nice quantification of the classification power of transcription factors for cell identity. In Figure 6, cell type diversity by brain region is nicely analyzed.

The Discussion is a pleasure to read, and it strikes an honest balance between highlighting the strengths of the paper and disclosing the limitations of the study.

Taken together, this manuscript has the potential to become a landmark study in the field, and describes a resource that dozens of laboratories around the world will be using as soon as the data become available.

This manuscript also exemplifies the challenges related to analyzing, describing and understanding these massive transcriptomic datasets.

The manuscript and its biological findings would be strengthened significantly by further quantitative analysis of the spatial heterogeneity of cell types. I think that these new datasets offer the exciting opportunity of validating or revising the anatomical ontologies that have informed neuroscience research over decades.

Below I provide general comments and criticism that apply to the entire study, and then specific and often minor comments on individual parts of the manuscript.

GENERAL COMMENTS

Here I raise three critiques to the study

1. Levels of classification and hierarchical organization of cell types

As the authors recognize in the Discussion, "the relationships between the large number of cell types across the entire brain are [...] impossible to be fully captured by a one-dimensional hierarchical tree or two-dimensional UMAPs". Yet, the entire paper is built around the idea that cell types should fall into a taxonomical classification. The idea motivates the introduction of different levels of classification, from clusters up to subclasses, classes, and divisions. This hierarchical classification is practically useful because it facilitates the description of the data. Cell type taxonomies have been used successfully by the same lead authors and by others to describe cellular diversity in smaller and more homogeneous parts of the brain.

However, does the assumption that cell types are hierarchical still hold true in whole-brain datasets?

From the way the main text is written, it appears as if the taxonomy is entirely data-driven. But that is not necessarily the case.

Subclasses are necessary to describe the data, and classes are acceptable (although some of them are counterintuitive). These levels of the taxonomy were obtained by graph clustering with different resolution parameters, which is a good approach, but using TFs instead of the entire gene set used for the initial clustering. Furthermore, the Methods section reveals that building these levels of the taxonomy involved multiple choices, extensive manual curation, and decisions on what to split and what to merge. The extent of manual curation and the criteria used need further clarification (how many classes and subclasses were edited by hand? Were quantitative criteria used to split and merge clusters?).

The seven divisions are more problematic. According to the Methods section (line 1495), divisions were obtained from the hierarchical clustering tree and "prior knowledge". This clustering tree (Figure 1a) is built by hierarchical clustering (build_dend function), an approach that is different from the graph-clustering approach used for the lower levels of the taxonomy. Notably, the fact that TFs were used to build the taxonomy is mentioned only in the Discussion of the paper.

Divisions seem rather arbitrary and frankly unnecessary - they do not add anything to the description of the dataset. On the contrary, they may generate confusion by grouping together heterogeneous groups of cells.

What is the authors' goal here?

1.1. If the goal is to provide a tool or simplified language to describe cellular diversity, then the process that went into defining these groups, and the limitations, should be described upfront in the main text. As it is now, the text reads as if the divisions were entirely data-driven (example: line 257-258).

1.2. If the goal is to describe molecular similarities and differences across cell types, then the taxonomy should be built from all variable/DE genes, instead of TFs.

1.3. If the goal is to build an atlas of cell types that adds to the existing anatomical ontology described in the Introduction (lines 71-82), then cell types should be described and classified in light of their assignment to those brain regions (Figure 1e) (inferred not from dissection, but after mapping on the MERFISH data). These previously-defined brain regions are not mentioned again until Figure 6. It appears indeed that Classes largely correspond to regions further split by neurotransmitter type and in some cases spatial position (e.g. anterior vs posterior). The relationships between the data-driven Classes and the ontology-based regions could be shown with an alluvial plot or similar; this would be an interesting way to show whether and how transcriptomic data are congruent with neuroanatomy or not (example: similar clusters in adjacent brain areas).

Smaller points:

-- MOB-DG-IMN: this subclass groups together neurons that presumably share the expression of genes defining an "immature state" (arguably, a "cell state"); however, these neurons are highly diverse. Does it make sense to group them in the same class? Indeed glutamatergic (DG, PIR IMN) and GABAergic (OB GABAergic INs) neurons are being grouped here!

Line 270: neurons in PAL-sAMY-TH-HY-MB-HB division were grouped together because of their "high degree of similarity and continuity". How is similarity quantified here? Similar question arises for line 304 "closely related neuronal subclasses" and line 309-310 "highly similar", and several other places in paragraph "Organization of neuronal cell types across the mouse brain". Proximity in the UMAP space is a poor criterion to infer similarity, because UMAP is a non-linear dimensionality reduction method.

2. The manuscript as a resource paper

The aim of this study was to generate a resource for the neuroscience community. As a resource, this body of work does not match yet the high standards established by the Allen Institute in the past. There is no web interface to explore the data, and it appears that cluster annotation needs further work, as admitted in the Discussion: "we have not extensively searched and utilized the vast amount of existing data and knowledge about cell types in many parts of the brain to help better annotate our cell type atlas." (Line 853 ss). It appears that an online platform for cell type exploration and annotation will not be available in the near future (line 857). I acknowledge that there is still value in presenting this dataset to the community, and that it will take several years to curate the cell type annotation. It would be useful, however, to provide a web interface even for a basic exploration of the scRNAseq and MERFISH data - it is extremely hard to navigate Figure 2 and similar figures, because the color schemes are complex and the MERFISH dots are small and often intermingled. Being able to visualize these clusters and their spatial location one by one would improve significantly the quality and accessibility of the paper.

3. Biological insights

The biological insights described in the manuscript include: an analysis of the spatial diversity of non-neuronal cells (the spatial heterogeneity of VLNC cells is extremely interesting), an analysis of neurotransmitter and neuropeptide expression (with a focus on cotransmission), and an analysis of TF modules. These analyses and their results are interesting, however they appear confirmatory. For example, Zeisel et al (Cell 2018) already described the molecular and spatial heterogeneity of non-neuronal cells, the spatial restriction of neuronal clusters, and transcription factor combinatorial codes. This manuscript confirms that the same principles hold true with a deeper cell type sampling, which is valuable but not novel.

Here, further quantitative analysis of the spatial heterogeneity of cell types would add novel insights. I believe that these analyses will be relatively easy, because they build directly on the existing datasets. There is an opportunity here to validate or revise the anatomical ontologies that have informed neuroscience research over decades.

3.1. Would be useful to get a better sense of the complexity and cell type diversity captured in the AIBS MERFISH dataset. For example: how many clusters can be inferred from MERFISH data alone? How does the UMAP of MERFISH data look like? This would be a useful benchmark for people interested in applying these techniques to different systems or species.

3.2. The Discussion states that "One of the most striking findings from our study is the high degree of correspondence between transcriptomic identity and spatial specificity." (Lines 748-749; see also Lines 279-281). However, spatial specificity is demonstrated qualitatively, by plotting some of the 5,200 clusters in the MERFISH space (Figure 2 and the like). Can this statement be strengthened by a quantitative analysis? For example, using the CCFv3 ontology as a reference: how many cell types fall entirely in a single anatomical region, and how many span multiple anatomical regions? One could devise a region-matching score and plot the distribution of these scores to demonstrate spatial specificity. This would also be a nice way to link the preexisting ontology to the new cell type data (and maybe find places where the anatomical ontology does not match cell type diversity?).

3.3. Figure 6a shows the number of clusters by brain region (very interesting!). How does this plot look at finer levels of the CCFv3 anatomical ontology? I suspect that there might be interesting surprises - if hypothalamic cell types, for example, segregate by nuclei, whereas cortical cell types remain intermingled over larger areas. What is the distribution of the number of cell types in finer anatomical regions?

3.4. What proportion of transcriptional heterogeneity is explained by the spatial distribution of cell types in the brain? Are there any gene families that explain spatial distribution more than others (example: transcription factors)?

3.5. Expanding on the point above: it appears that the neuronal classes shown in Figure 2c could be derived by the intersection of spatial coordinates (brain region) and neurotransmitter identity (largely glutamatergic and GABAergic). Could neuronal diversity in the mouse brain be described by such a simple rule? If so, are there any specific gene families whose expression explains these major axes of neuronal diversity?

SPECIFIC COMMENTS

ABSTRACT

Line 37: The number of single cells profiled is ~ 7 million, but the cells that survived QC are ~ 4.1 million. Please include this number in the abstract. Would be useful to indicate in the main text what fraction of the total number of cells and neurons in the mouse brain this represents.

Line 44: dichotomy between the dorsal and ventral parts of the brain. The terms dorsal and ventral are vaguely defined - do dorsal and ventral here correspond to alar and basal subdivisions?

Line 46: how is "closely related" measured here? Number of DE genes? Proximity in the UMAP space? See also comments above

INTRODUCTION

General comment: the Introduction as a whole does not outline well the biological questions addressed in the results section of the paper. Lines 84-100, in particular, do not seem useful. The second part of the Introduction outlines the importance of a complete resource, but besides the future applications of the atlas, it would be extremely interesting to outline here the biological questions that can be addressed with the atlas directly - and that are addressed here.

Line 60: brain is "the most complex system in life" -> aren't animal societies or ecosystems arguably complex too?

Line 72: This is a minor point, but still worth mentioning. I would agree with the statement that the mesoscale architecture of the mammalian brain is well understood. However, there is still some debate on what is the most meaningful description of this architecture, particularly between proponents of developmental bauplan-based models, and proponents of circuit-based models ("columnar models" in some literature). Citations 7 and 8 refer to the latter; however, the results of this paper are more in line with developmental models! (For example: transcription factor analysis) Cfr the work of Puelles, Nieuwenhuys and colleagues

Lines 74-82: These lines summarize the CCFv3 ontology of the mouse brain established by the Allen Institute. This ontology is based on adult neuroanatomy and for some parts of the brain does not match our understanding of the developmental and molecular diversity of neuron types - take the amygdala as an example. It would be useful to remind the reader that the anatomical ontology doesn't necessarily reflect the developmental history, nor the molecular and cellular diversity of these brain areas.

Line 102: cell types as functional units. Wouldn't it be more accurate and complete to say that cell types are "structural and functional" units? There are many examples where cell types alone are not sufficient for function - in the brain, neural circuits are arguably the functional units.

Lines 102-115: would be worth mentioning here that the cell type classification obtained by transcriptomics is congruent with morphological and functional classifications - as demonstrated by outstanding work by the Allen institute.

RESULTS

FIGURE 1

- I think it is important to specify here in the text or in the figure legend that the taxonomy (Figure 1a) is built using transcription factors
- the multiple color codes in Figure 1a are very hard to follow and disentangle. Any better way to represent the same information?
- what do the shades of gray in the dendrogram (Figure 1a) indicate? It is not explained in the legend.

FIGURE 2

- Color schemes and spatial distribution of clusters are very hard to track and follow! The numbers help, but do not solve all problems. Potential solutions: (i) a web interface where clusters and subclasses can be plotted one by one in the UMAP and MERFISH spaces; (ii) close-up on a specific anatomical region (amygdala? Hypothalamus?) to exemplify the insights gained from these new datasets.

Lines 284-286: it is not clear from the data reported here what would support the statement of "homology" of subclasses of glutamatergic neurons across the pallium (a supplementary figure missing?). Of note, the term homology refers to evolutionary relationships (descend from common ancestor). Without an evolutionary analysis, this term is not appropriate to describe similarities of cell types across anatomical areas.

Lines 292-299: do the classes reflect the distinction between projection neurons and interneurons? Why using a nomenclature that is not internally consistent? (some classes named after their ganglionic eminence of origin, other after the anatomical destination of neurons...)

Lines 476-477: do all telencephalic astrocytes express *emx2*, a pallial marker? Or is *emx2* expressed only in a subset of the Astro-TE cells?

Line 489: what is the exact percentage of clusters expressing at least one neuropeptide gene?

FIGURE 4

Lines 530-532: a subset of CR cells is born in the prethalamic eminence, the site of origin of mitral and tufted cells (M/T) of the posterior accessory olfactory bulb (see for example Huilgol et al Nature Neuroscience 2013 PMID: 23292680). M/T cells of the main and accessory olfactory bulb are transcriptomically distinct (Zeppilli et al eLife 2021 PMID: 34292150). The manuscript does not distinguish between these two parts of the olfactory bulb (only MOB is mentioned). It is possible that the relatedness of CR cells is limited to M/T cells of the AOB, and in that case, shared developmental origin would be a potential explanation.

Lines 535ss: I wonder whether a potential alternative explanation for the lack of a trajectory between DG Imn and DG differentiated neurons is the fact that DG and piriform immature neurons are lumped together here (unclear on what basis).

Lines 543ss: RMS astrocytes were already introduced earlier, without explaining the existence of a "tunnel" for neuronal migration. This section of the paper would be easier to read after bringing these paragraphs closer together.

Section: "Transcription factor modules across the whole mouse brain"

General comment:

Many of the observations listed here (for example: the existence of a dichotomy between forebrain and midbrain-hindbrain GABAergic TFs, the fact that LIM homeodomain TFs have complementary

expression patterns in the brain, line 628) are not entirely new; classical in situ hybridization studies reached similar conclusions. The relevant references should be cited.

Specific comments:

Lines 573-574: because the TF modules were computed at the subclass level, some of the results are not consistent with the biology. For example, Tbr1 and Satb2 come up as a pallium glutamatergic-specific module, although these two TFs are well known for being expressed selectively in distinct types of cortical neurons. This limitation of the TF analysis should be made explicit in the main text.

Lines 582-583: Gbx2 labels a subset of thalamic glutamatergic neurons; midbrain neurons express other TFs (example: pax3 and pax7) besides Shox2 and Tcf7l2

FIGURE 5

- the axis labels in 5c and 5d are hard/impossible to read

FIGURE 6

- Fig 6d: if I understand this correctly, spatial span is not normalized by area size. Then, the fact that pallium neurons are more broadly distributed is not a surprise. Any way to take brain region sizes into account?

DISCUSSION

Line 748ss: the correspondence between transcriptomic identity and spatial specificity is not strictly quantified (see general comments above).

METHODS

- how well do the microdissected regions match with the post-hoc MERFISH-based spatial allocation of cells?

- how many cells were loaded on the 10x Chromium Controller?

Lines 1483-1485: identification of classes and subclasses. The taxonomy was built using TF expression, because the results "recapitulate existing knowledge" better. It would be useful to see a comparison between these two clustering approaches (499 TFs vs ~8k marker genes)

- is the new classification (clusters and subclasses) consistent with Yao et al Cell 2021?

Lines 1489ss: how many classes and subclasses underwent manual correction?

Referee #2:
Remarks to the Author:

This landmark study represents the first effort to integrate both scRNA-Seq and spatial transcriptomics to generate a full catalog of cell types in the adult mouse brain. This work is a true tour de force, and the cumulation of steadily improved and refined multidecade effort from the Allen Brain Atlas. The data quality is, as we have come to expect, excellent, the methodology transparent, and the data itself freely accessible to the broader research community. The authors should be warmly congratulated on their effort, skill, and public spiritedness. While I have very few concerns about the data or its presentation, I have several general questions about the completeness of this resource, its broader relevance to schema for CNS cell taxonomy, its direct relevance to existing criteria used for brain cell classification, and its comparison to similar datasets from other CNS regions. I also have some suggestions for how the data can be made even more useful to the broader research community. These are listed below.

General questions:

1. Despite the very impressive analysis done here, this is likely not close to a complete inventory of brain cell types. On the one hand, is highly likely that many rare cell types have been missed, particularly since scRNA-Seq rather than snRNA-Seq was used here and the still somewhat limited sensitivity of MERFISH analysis. Ongoing analysis of cell diversity in the *Drosophila* brain, for instance, is identifying multiple examples of molecularly, morphological, and functionally distinct neuronal subtypes that are found only once in each hemisphere (PMID:33438579 and unpublished). Likewise, analysis in the mouse retina suggests that fewer than ten cells may be present per retina for some subtypes of wide-field amacrine cells (PMID:17048228). This implies that potentially thousands (or even tens of thousands) of additional cell types may remain to be identified.

However, it is equally clear that some of the cell subtypes identified represent transient cell states. This is obviously the case for immature adult-generated neurons and differentiating oligodendrocytes but is also likely to account for some of the other identified subtypes. For instance, many of the astrocyte subclusters are distinguished by differential expression of *Gfap*, which is a classic marker of astrocyte activation, while at least five neuronal subtypes are distinguished by expression of classic IEGs.

This raises the critical question of when the task of generating a brain cell atlas is actually going to be *done*, and how complex the final taxonomy of brain cell types is likely to be. In other words, how long do we have to wait for the next iteration of this study, and how many more are there likely to be? The authors could help address this by adding the following analysis and discussion:

A. Downsample their scRNA-Seq dataset to determine the rate at which new taxonomic categories (Division, Class, Subclass, Supertype, Subtype) emerge as the number of cells increases. Are there any brain regions which have clearly reached saturation? Which are still relatively undersampled?

B. Extrapolate these results to estimate the number of cells that would need to be analyzed to reach saturation for each major anatomical subdivision.

C. Estimate (or at least discuss) the effect of biases that arise from the use of scRNA-Seq instead of snRNA-Seq for cell profiling. It is fairly well-established at this point that scRNA-

Seq enriches at least to some extent for cell types that tolerate dissociation and high-pressure microfluidic handling well, while large, fragile neurons tend to drop out. Have the authors conducted scRNA-Seq and snRNA-Seq in parallel on identical samples, and what do they observe?

D. While the study (correctly) focuses using scRNA-Seq to identify cell cluster, and validating key cluster-specific markers using MERFISH, and focusing on taxonomic categories identified using both methods, this raises the question of what was detected using only one of these methods. What fraction of scRNA-seq based clusters did not validate with MERFISH, and was it possible to identify novel (likely rare) cell clusters using MERFISH that were not detected in the scRNA-Seq dataset?

E. While the dataset generally doesn't miss distinct well-characterized cell types (and I spend quite a few hours looking up my favorite known and as yet unpublished cell types to make sure they were there), there are a few examples. Taking glial cells, it is clear that some cell subtypes may be missed. This study distinguishes only six distinct subtypes of astrocytes, while other studies have identified substantial regional differences in astrocyte gene expression profile (e.g. PMID:33827819). Likewise, while this analysis cleanly resolves some distinctions among hypothalamic tanycytes (e.g. alpha from beta1 and beta2), it does not distinguish alpha1 and alpha2 subtypes, and it is unclear whether it identifies radial glial cell types found in other circumventricular organs such as the SFO, OVLT, and area postrema. Can the authors identify some other examples of missing cell types, particularly for reasonably well-characterized neurons?

F. Since the MERFISH-based validation profiles coronal sections separated by 200 microns, can the authors comment on which ABA-defined anatomical structures might be missed entirely

2. Approaches to cell taxonomy:

The author's observation that persistent adult expression of developmentally to important transcription factors is an accurate predictor of cell identity is both appealing and unsurprising to developmental neuroscientists, since controlling cell-type specific gene expression is what transcription factors do, and in any case these results tally closely with findings from adult *C. elegans*. However, this perspective is completely different from how neurophysiologists are used to thinking of cell identity, which they typically see as determined by expression of neurotransmitters, neuropeptides, and calcium binding proteins (one only needs to look at which Cre lines are in Jackson Lab's live repository to see which perspective dominates here). Since this finding will be resisted by some, and ignored by many, it's worth expanding and discussing further.

A. The comparison of the predictive value of TFs vs. random markers shown in Fig. 5b and 5c isn't really the proper comparison here. A more useful comparison would be TFs vs. (neurotransmitters+neuropeptides+calcium binding proteins), which directly compares the predictive value of these "developmental" vs. "physiological" molecular markers.

B. In contrast to *C. elegans*, large-scale scRNA-Seq analysis in *Drosophila* brain seems to suggest that cell adhesion molecules and GPCRs may be better markers of cell identity than TFs.

Are there any other functional categories of genes that have better predictive power than TFs in this dataset?

C. The particular prominence of transcription factors that are expressed during neurogenesis, rather than in mature neurons, in determining identity strongly implies that analysis of earlier stages of development could be a royal road to a final taxonomy of CNS cell types. If cell types are specified at (or shortly after) exit from mitosis, as developmental data suggests, it may be possible to rigorously distinguish fixed cell types from dynamic cell states -- between which cells can shift depending on internal state, extrinsic signals, or neuroplastic changes -- by analyzing the developing brain. The lead author has previously discussed these issues at length in multiple review articles, but it would be worth revisiting this issue here.

D. The hierarchical taxonomy used here is almost explicitly Linnean, with Division corresponding to Linnean Class, Class to Linnean Order, Subclass to Family, Supertype to Genus, and Cluster/Type to Species. This is intuitive and appealing, and makes use of rigorous molecular criteria in much the same way traditional taxonomy made use of comparative anatomy. This has the potential to be a useful general framework for classifying CNS cell identity, and I would like to see it discussed further and made more explicit.

3. Relevance to existing neuroanatomic models:

There are a great many interesting details buried in the study which have important implications for our existing models of brain organization. While there's obviously far too much to cover in this article, it would be good if some of these points could be at least touched on.

A. Which histoarchitecturally distinct brain regions and cytoarchitecturally defined cell groups (as defined in the ABA Reference Atlas) lack clear region-specific markers? Considering DA neurons, for instance, it seems there is high diversity among midbrain (A8-10), arcuate (A12), and olfactory (A16) clusters, but where are the others (A11/13/14/15)?

B. The observation that dorsally located cell types (pallium+thalamus+cerebellum) are less diverse but more transcriptionally distinct than ventrally located cell types is very interesting. Does this also hold for cells of subpallial origin (e.g. GABAergic pallial neurons)? Is it a general alar/basal (sensory vs. motor) distinction, and does it hold for dorsal midbrain (SC/IC), or dorsal HB?

C. Both the distribution of individual neuronal cell types, and the transcriptional relatedness of these subtypes, generally fits very well with current understanding of the organization of both the developing and adult brain. However, there are several major exceptions, which the authors mention but do not discuss further. First, they observe a close transcriptional relationship of the thalamic reticular nucleus (TRN) and hypothalamic GABAergic cells. Standard prosomeric models of forebrain organization (which are used as the framework for the ABA's own Developmental Brain Atlas) state that the TRN and the hypothalamus proper derive from prosomere 3 and secondary prosencephalon, respectively. What is the significance of this and does this call this aspect of the model into question? Second, the observation that certain closely related glutamatergic cell types are found in thalamus and midbrain is also surprising, given that cell mixing between these compartments has not been previously reported during neurogenesis.

Might these correspond to neuronal subtypes that migrate postmitotically (like thalamic GABAergic neurons) or cells derived from the isthmus, which forms the border zone between these regions?

4. Comparison to other CNS structures:

It's something of a missed opportunity to not compare this brain dataset to the extensive datasets generated from spinal cord and retina, thereby obtaining a truly integrated view of CNS cell diversity. This is particularly the case for the retina, where an essentially scRNA-Seq based catalog of cell subtypes has already been generated, and a great deal of high-quality data is publicly available (c.f. PMID:34651173). Can this data be integrated into the analysis?

Practical questions:

Most researchers using this resource will be either physiologists who simply want to selectively monitor or manipulate specific cell types in their region or circuit of interest, using site-specific recombinases, viral minipromoters, etc or molecular neuroscientists wanting to analyze mutants, disease states, etc. As a result, it's essential that users can readily find the exact minimal number of genes needed to identify each taxonomic category described here. This is pretty non-obvious at this point, and this information needs to be provided in a separate file. The number of genes used to uniquely define each category should be listed, and this should be stated separately for both the brain as a whole and for each major anatomic subdivision. This will allow rigorous interpretation of the effects of both systemic (mutant analysis, BBB-permeable AAV delivery) and anatomically targeted (e.g. AAV-based) manipulations of cell function. This really has the potential to dramatically alter the design of these experiments, shifting effort towards analysis of cell types that can be defined by relatively small numbers of genes.

Other questions:

A. In Fig. S3a, it appears that most neurons seem to have 2-2.5x UMI/cell relative to CBX/MOB cells and astrocytes. This doesn't seem to reflect a difference in cell volume, at least according to Fig. S2d, so what is the significance of this?

Clarifications:

A. Please define hypendymal cells and spell out choroid plexus (CHOR).

Author Rebuttals to Initial Comments:

Response to referees

Overall responses

We thank the referees for their extensive and informative comments on this manuscript. The comments have been extremely helpful in our effort to improve the manuscript both scientifically and for readability. All referees recognized the value of this “landmark” effort as a deep-dive investigation of cell types in the whole mouse brain, providing to the community not only an unprecedentedly rich set of data and resources but also a conceptual framework to guide future efforts in understanding brain cell types and their functions. At the same time, the referees also provided many constructive critiques and suggestions, emphasizing some of the critical issues around defining cell types and understanding the relationships between cell types from different brain regions.

In the revised manuscript, we have addressed each of the points raised by the referees. Importantly, we have carefully revised our whole mouse brain (WMB) transcriptomic cell type taxonomy, based on referees’ comments as well as our more detailed anatomical and literature-based annotation. We have also publicly released our scRNA-seq and MERFISH datasets in a new online platform which our Technology team built through years-long efforts. This new online platform allows not only data download but also interactive exploration of the cell type taxonomy and atlas, for every class, subclass, supertype and cluster, and their spatial location and gene expression signature. Detailed point-by-point responses are shown in the next section, with changes to the manuscript underlined. Here we provide a summary of the major revisions we have made to the manuscript:

- Present our revised (and refined) WMB cell type taxonomy and atlas, and update all the figures based on the revised taxonomy. We have coordinated with other groups contributing other companion papers of this publication package, and this revised taxonomy will be used in all other relevant companion papers as well.
- Introduce our new online platform, Allen Brain Cell Atlas (ABC Atlas) for the whole mouse brain, which displays both scRNA-seq and MERFISH datasets in an interactive manner, visualizing gene expression, cell type definition at all hierarchical levels, and each cell type’s regional and spatial localization.
- Incorporate new figures and revision of the text thoroughly to address referees’ questions and suggestions. Major changes to figures include:
 - To address the issue of how hierarchical cell type taxonomy was built and why using transcription factors (TFs), we have added Extended Data Figures 3 and 4.
 - To illustrate the correspondence between transcriptomic identity and relatedness and spatial specificity and relatedness, we have added several panels to Figure 6, Extended Data Figures 8, 13, 14 and 15.
- Remove a few extended figures and figure panels that are not essential to the core findings of the paper to make space for the new figures. These include old Extended

Data Figure 6 (highly distinct cell types), old Extended Data Figure 10 (transcription factors along the hierarchical tree), and old Figure 6e-g (transitioning cell types).

With these changes and clarifications, we hope the editor and referees agree that we have revised our manuscript satisfactorily for publication in *Nature*. We would be happy to address any further comments you may have.

Referees' comments:

Referee #1 (Remarks to the Author):

The manuscript "A high-resolution transcriptomic and spatial atlas of cell types in the whole mouse brain" by Yao and colleagues reports on the generation and initial analysis of cell type and spatial transcriptomic atlases from the entire adult mouse brain. The authors identify 5,200 high-quality clusters (scRNAseq), map their spatial distribution in the brain (MERFISH), and describe more in depth the expression of neurotransmitters, neuropeptides and transcription factors (TFs) to infer principles of cell types diversity and spatial distribution in the brain.

The work described here is truly a herculean effort. It entailed sequencing about 7 million single cells, MERFISH experiments on ~50 brain sections, and development of new computational tools to handle these large datasets. For comparison, two highly-cited studies describing mouse brain cell type atlases, Zeisel et al Cell 2018 and Saunders et al Cell 2018, sequenced ~500,000 and ~690,000 cells, respectively.

Notably, the experimental design included sampling from males and females, to discover sex-specific clusters, and from animals in different phases of the circadian cycle. The analysis pipeline included stringent quality controls (QC) to filter out low-quality cells; these QC criteria are more stringent than those commonly used in the field.

Furthermore, the manuscript describes biological insights emerging from the analysis of these data. Figure 5 includes a nice quantification of the classification power of transcription factors for cell identity. In Figure 6, cell type diversity by brain region is nicely analyzed.

The Discussion is a pleasure to read, and it strikes an honest balance between highlighting the strengths of the paper and disclosing the limitations of the study.

Taken together, this manuscript has the potential to become a landmark study in the field, and describes a resource that dozens of laboratories around the world will be using as soon as the data become available.

This manuscript also exemplifies the challenges related to analyzing, describing and understanding these massive transcriptomic datasets.

The manuscript and its biological findings would be strengthened significantly by further quantitative analysis of the spatial heterogeneity of cell types. I think that these new datasets offer the exciting opportunity of validating or revising the anatomical ontologies that have informed neuroscience research over decades.

Below I provide general comments and criticism that apply to the entire study, and then specific and often minor comments on individual parts of the manuscript.

We appreciate the referee's very positive comments on our manuscript. We agree that the manuscript could be strengthened by further quantitative analysis of the spatial heterogeneity of cell types which has the potential to update the anatomic ontologies of the mouse brain. We have dedicated the entire Figure 6 and four new Extended Data Figures for the comparative spatial analysis as detailed below.

GENERAL COMMENTS

Here I raise three critiques to the study

1. Levels of classification and hierarchical organization of cell types

As the authors recognize in the Discussion, "the relationships between the large number of cell types across the entire brain are [...] impossible to be fully captured by a one-dimensional hierarchical tree or two-dimensional UMAPs". Yet, the entire paper is built around the idea that cell types should fall into a taxonomical classification. The idea motivates the introduction of different levels of classification, from clusters up to subclasses, classes, and divisions.

This hierarchical classification is practically useful because it facilitates the description of the data. Cell type taxonomies have been used successfully by the same lead authors and by others to describe cellular diversity in smaller and more homogeneous parts of the brain.

However, does the assumption that cell types are hierarchical still hold true in whole-brain datasets?

From the way the main text is written, it appears as if the taxonomy is entirely data-driven. But that is not necessarily the case.

Subclasses are necessary to describe the data, and classes are acceptable (although some of them are counterintuitive). These levels of the taxonomy were obtained by graph clustering with different resolution parameters, which is a good approach, but using TFs instead of the entire gene set used for the initial clustering. Furthermore, the Methods section reveals that building these levels of the taxonomy involved multiple choices, extensive manual curation, and decisions on what to split and what to merge. The extent of manual curation and the criteria used need further clarification (how many classes and subclasses were edited by hand? Were quantitative criteria used to split and merge clusters?).

We have provided more detailed clarification on how classes and subclasses were computationally defined and then manually curated, in the first Results section (text lines

188-233). Clusters were defined solely based on fully automated computational pipeline using all the genes and pre-determined quantitative criteria based on DE genes for splitting and merging as described in Methods. TF expression was only used to define the cell type hierarchy, which, based on our observation, better represents lineage and spatial distribution than categorization based on using all differentially expression (DE) genes. In addition, using TF expression for calculating hierarchy provides cleaner borders between classes/subclasses while still preserving the global transcriptomic similarity. Clusters that are similar based on TFs are also similar based on all DE genes, but not necessarily so vice versa. Using all DE genes to compute similarity may dilute the TF signals that separate cell types in distinct anatomical locations. These observations have now been succinctly captured in the new Extended Data Figure 3 (text lines 188-197), which shows the gene expression correlation matrices between each pair of the ~5,300 clusters computed using all DE genes, subsets of TFs, functional genes, or adhesion molecules, as well as summarizing plots in Figure 5a-f (text lines 677-685). These plots together show that TFs have the best distinguishing power of cell types at different levels and creating the hierarchical taxonomy compared to all DE genes and other example gene families.

A critical examination of any classification approach is essential, and all computational algorithms will imperfectly characterize complex biology. In this study, our manual examination for robustness of identified classes and subclasses through application of the Jaccard-Louvain algorithm suggested that the choice of parameter based on k-nearest neighbors may not work equally well in all cases (e.g., highly distinct cell types from all other types), and this algorithm does not account for the distance to specific neighbors. Furthermore, manual curation allowed us to identify additional low-quality clusters that had escaped our initial generic QC process (as described in Methods and Extended Data Figure 1a) and might have affected the computational process as well. To account for these effects, manual adjustments were made, assisted with MERFISH data, in re-assignment of transitional types to increase the spatial proximity, as well as misplaced cell types (for unknown reasons) whose re-assignment to a new but related supertype and subclass was validated by spatial correspondence. Likewise, subclasses with distinct sub-populations that were also spatially separated were further split manually to reflect this spatial separation. We believe these modifications improve the robustness of the taxonomy. We have provided confusion matrices for classes and subclasses before and after manual correction in the new Extended Data Figure 4 (text lines 223-226), to show that our manual adjustments only finetuned the taxonomy, without substantial alteration of the initial computationally generated framework.

The seven divisions are more problematic. According to the Methods section (line 1495), divisions were obtained from the hierarchical clustering tree and "prior knowledge". This clustering tree (Figure 1a) is built by hierarchical clustering (build_dend function), an approach that is different from the graph-clustering approach used for the lower levels of the taxonomy. Notably, the fact that TFs were used to build the taxonomy is mentioned only in the Discussion of the paper.

Divisions seem rather arbitrary and frankly unnecessary - they do not add anything to the description of the dataset. On the contrary, they may generate confusion by grouping together heterogeneous groups of cells.

We agree that the divisions were less informative and more arbitrary, and we have removed them from our revised cell type taxonomy. Also, because the one-dimensional hierarchical tree (dendrogram shown in the original Figure 1a) and the taxonomy class groupings do not always agree, resulting in confusion, we have replaced it with the simple order of classes and subclasses as the major representation of the taxonomy (new Figure 1a) (text lines 226-242), which will correspond well with the anatomical/regional organization of the brain.

What is the authors' goal here?

1.1. If the goal is to provide a tool or simplified language to describe cellular diversity, then the process that went into defining these groups, and the limitations, should be described upfront in the main text. As it is now, the text reads as if the divisions were entirely data-driven (example: line 257-258).

1.2. If the goal is to describe molecular similarities and differences across cell types, then the taxonomy should be built from all variable/DE genes, instead of TFs.

1.3. If the goal is to build an atlas of cell types that adds to the existing anatomical ontology described in the Introduction (lines 71-82), then cell types should be described and classified in light of their assignment to those brain regions (Figure 1e) (inferred not from dissection, but after mapping on the MERFISH data). These previously-defined brain regions are not mentioned again until Figure 6. It appears indeed that Classes largely correspond to regions further split by neurotransmitter type and in some cases spatial position (e.g. anterior vs posterior). The relationships between the data-driven Classes and the ontology-based regions could be shown with an alluvial plot or similar; this would be an interesting way to show whether and how transcriptomic data are congruent with neuroanatomy or not (example: similar clusters in adjacent brain areas).

We thank the referee for the great suggestions. Our goal is to define and describe whole brain cell types in an accessible way that captures the different levels and dimensions of inter-relatedness among cell types (e.g., developmental, evolutionary, anatomical, physiological, etc.). Due to the complex transcriptional landscape with many levels and dimensions, this is a formidable goal, and any representation is unlikely to completely incorporate all dimensions of variation into one simplified representation. For example, some molecularly highly distinct cell types from the same region may share common molecular signatures that correspond to a spatial specificity or gradient. An example of this is that the highly distinct striatal D1 and D2 neurons share a similar gene expression axis that corresponds to the same spatial gradient. At the same time, many of these molecularly highly distinct cell types also have their own molecularly close relatives in other regions. In our hierarchical taxonomy, the latter aspect is the main driver for cell type organization; however, a cell type may be misplaced to a very different branch due to regional signatures.

As mentioned above, at the finest-level cluster splitting, we used all the DE genes to define clusters. Thus, cluster segregation reflects all molecular variables and represents various cellular properties. To build the hierarchical taxonomy, we found that using transcription factors most faithfully recapitulates the global relationships among cell types, likely due to the fact that TFs better recapitulate the developmental origins of cell types. Many other marker genes, on the other hand, could be related to the cell types' physiological or functional properties, which can be highly variable but do not systematically delineate the overall distance relationships across all cell types. As mentioned above, we have now shown

this through the new Extended Data Figure 3 and Figure 5a-f to compare the performance of TFs, other gene families, and all DE genes in creating the hierarchical taxonomy.

We have generated the new Figure 6a to show the relationship between transcriptomic data-driven classes/subclasses and brain ontology-based regions, which reveals a high degree of concordance between transcriptomics and neuroanatomy (text lines 747-761). We have also described some cross-region transitions in sections introducing cell types in hypothalamus, thalamus and midbrain, which may reflect cell type migration patterns during development as well as potential limitations and knowledge gaps in the current neuroanatomical delineations.

Smaller points:

-- MOB-DG-IMN: this subclass groups together neurons that presumably share the expression of genes defining an "immature state" (arguably, a "cell state"); however, these neurons are highly diverse. Does it make sense to group them in the same class? Indeed glutamatergic (DG, PIR IMN) and GABAergic (OB GABAergic INs) neurons are being grouped here!

Indeed, they were grouped together because of their common immature states, i.e., the progenitors share common signatures. We admit that this may become problematic, especially considering a future developmental cell type atlas where many cell types may be derived from common progenitors. Thus, in our revised taxonomy we have now separated the MOB-DG-IMN class into two classes, DG-IMN Glut and OB-IMN GABA.

Line 270: neurons in PAL-sAMY-TH-HY-MB-HB division were grouped together because of their "high degree of similarity and continuity". How is similarity quantified here? Similar question arises for line 304 "closely related neuronal subclasses" and line 309-310 "highly similar", and several other places in paragraph "Organization of neuronal cell types across the mouse brain". Proximity in the UMAP space is a poor criterion to infer similarity, because UMAP is a non-linear dimensionality reduction method.

We agree that we used the words "similar" and "similarity" too casually. We understand that UMAP is a non-linear dimensionality reduction method, though its relative proximity is not completely meaningless (especially 3D UMAP which was what we used most often in our assessment). To define inter-relatedness among cell types more clearly, we have now provided Extended Data Figure 3, a quantitative plot to show the transcriptomic similarity between each pair of clusters, using correlation of gene expression of all marker genes or TF markers as measurement of similarity (text lines 188-197).

2. The manuscript as a resource paper

The aim of this study was to generate a resource for the neuroscience community. As a resource, this body of work does not match yet the high standards established by the Allen Institute in the past. There is no web interface to explore the data, and it appears that cluster annotation needs further work, as admitted in the Discussion: "we have not extensively searched and utilized the vast amount of existing data and knowledge about cell types in many parts of the brain to help better annotate our cell type atlas." (Line 853 ss). It appears that an online platform for cell type exploration and annotation will not be available in the near future (line 857). I acknowledge that there is still value in presenting this dataset to the

community, and that it will take several years to curate the cell type annotation. It would be useful, however, to provide a web interface even for a basic exploration of the scRNAseq and MERFISH data - it is extremely hard to navigate Figure 2 and similar figures, because the color schemes are complex and the MERFISH dots are small and often intermingled. Being able to visualize these clusters and their spatial location one by one would improve significantly the quality and accessibility of the paper.

The Allen Institute for Brain Science team has been working hard over the past two years to build a new online platform to visualize both our scRNA-seq and MERFISH datasets. We are very pleased to announce here that we have now released this online platform, named Allen Brain Cell Atlas (ABC Atlas), through which the entire datasets and detailed cell type annotations (including both scRNA-seq and MERFISH datasets, as well as additional MERFISH datasets from Xiaowei Zhuang's lab as described in the Zhang et al companion paper) are made fully and freely available to the scientific community. Furthermore, ABC Atlas enables interactive visualization and navigation of the datasets to allow exploration of specific cell types, their spatial locations and their gene expression profiles. In the revised manuscript we have now added a Results section to describe the new ABC Atlas online platform as a major resource for the community (text lines 283-311).

Indeed, the cell type annotation is likely to be further refined and updated in upcoming years with additional data and analysis as well as community engagement. We will continue to update our ABC Atlas periodically to reflect those efforts.

3. Biological insights

The biological insights described in the manuscript include: an analysis of the spatial diversity of non-neuronal cells (the spatial heterogeneity of VLMC cells is extremely interesting), an analysis of neurotransmitter and neuropeptide expression (with a focus on cotransmission), and an analysis of TF modules. These analyses and their results are interesting, however they appear confirmatory. For example, Zeisel et al (Cell 2018) already described the molecular and spatial heterogeneity of non-neuronal cells, the spatial restriction of neuronal clusters, and transcription factor combinatorial codes. This manuscript confirms that the same principles hold true with a deeper cell type sampling, which is valuable but not novel.

Here, further quantitative analysis of the spatial heterogeneity of cell types would add novel insights. I believe that these analyses will be relatively easy, because they build directly on the existing datasets. There is an opportunity here to validate or revise the anatomical ontologies that have informed neuroscience research over decades.

The concept of transcription factor code is indeed very old, especially in the context of developmental studies, but identification of such a code in the adult stage at fine cell type resolution and at the whole brain level has been lacking. We were inspired by the work of Zeisel et al and tried to go beyond that by making the TF code more comprehensive. Here we attempted to provide such a code covering all cell types at both subclass and individual cluster levels. We identified many differentially expressed TFs even between cell type clusters within the same subclass (Figure 5a) and showed that TFs have great predictive power even at the cluster level (Figure 5b-f). Therefore, we believe that TFs are involved in

cell type specification at all levels for majority of cell types, even in the adult stage examined here. Figure 5g provides a global view of TF gene modules at a high level. While we didn't discuss this in the manuscript, we also provided a unique combination of TFs for every cluster. This is a starting point that will facilitate future developmental studies to examine how faithful such a TF code recapitulates the process of cell lineage specification.

Please see our responses below regarding further quantitative analyses of the spatial heterogeneity of cell types.

3.1. Would be useful to get a better sense of the complexity and cell type diversity captured in the AIBS MERFISH dataset. For example: how many clusters can be inferred from MERFISH data alone? How does the UMAP of MERFISH data look like? This would be a useful benchmark for people interested in applying these techniques to different systems or species.

We previously tried to use MERFISH data alone for clustering but found two major issues with this approach. First, the 500-gene panel is not able to generate the high-resolution taxonomy with as many clusters as our scRNA-seq data. This is because to separate a pair of clusters, we require the presence of a considerable number of significant differentially expressed genes between the two clusters. With the 500-gene panel, we can only afford to include a couple of DE genes between closely related types, which is usually enough to map the MERFISH cells to corresponding scRNA-Seq clusters, but not enough for de novo clustering. On the other hand, lowering the statistical stringency for clustering makes the results susceptible to serious technical artifacts. One key artifact is due to cell segmentation, which is still imperfect at this stage (though it is improving as we have sequentially applied several methods including Watershed, Cellpose and Baysor). Even minute amounts of contaminant transcripts from neighboring cells lead to multitudes of artificial clusters, because each cell's neighboring cells are different. Therefore, our experience is that mapping segmented MERFISH cells to scRNA-seq based clusters provides more reliable results.

3.2. The Discussion states that "One of the most striking findings from our study is the high degree of correspondence between transcriptomic identity and spatial specificity." (Lines 748-749; see also Lines 279-281). However, spatial specificity is demonstrated qualitatively, by plotting some of the 5,200 clusters in the MERFISH space (Figure 2 and the like). Can this statement be strengthened by a quantitative analysis? For example, using the CCFv3 ontology as a reference: how many cell types fall entirely in a single anatomical region, and how many span multiple anatomical regions? One could devise a region-matching score and plot the distribution of these scores to demonstrate spatial specificity. This would also be a nice way to link the preexisting ontology to the new cell type data (and maybe find places where the anatomical ontology does not match cell type diversity?).

These are great suggestions. We admit that although we performed detailed anatomical annotation of each cell type (shown in Supplementary Table 7) and named each subclass by the major region(s) it resides in, we did not provide an overview graphic to show the spatial specificity of cell types at different brain structure levels as well as cell-type taxonomy levels. In the revised manuscript, we have now provided a quantitative graph in the new Figure 6a using CCF-registered MERFISH data, as well as quantitative scores such as the Gini

coefficient (new Extended Data Figure 13) and Shannon diversity index, to show the highly specific distribution patterns of cell types across anatomical regions and their correspondence with transcriptomic class and subclass identities at the global level (text lines 747-761). More detailed investigation of molecular-anatomical relationship at cluster level will continue in future years, likely in conjunction with developmental studies, which could even lead to refinement and revision of brain regional ontology and anatomical boundaries.

3.3. Figure 6a shows the number of clusters by brain region (very interesting!). How does this plot look at finer levels of the CCFv3 anatomical ontology? I suspect that there might be interesting surprises - if hypothalamic cell types, for example, segregate by nuclei, whereas cortical cell types remain intermingled over larger areas. What is the distribution of the number of cell types in finer anatomical regions?

We have added a new plot, Figure 6d, at CCFv3 mid-ontology level, which contains ~270 brain subregions, to show the number of clusters in each subregion (text lines 792-801). Indeed as the referee expected, the difference among major brain regions shown in Figure 6b,c is evened out and sometimes even reversed in finer subregions, for example, in general there are more clusters in each cortical area than hypothalamic nucleus. This is consistent with our other observations that while there are more clusters in HY, MB and HB overall, these clusters are more confined to specific subregions or nuclei and thus spatially more isolated from each other, resulting in fewer clusters each subregion; clusters in pallium and subpallium (CNU), on the other hand, tend to be more widely distributed across multiple subregions, leading to more clusters intermixed in each subregion.

Of note, as shown in companion paper Zhang et al, Figure 2C, the local neuronal-composition complexity is still higher in the HY, MB and HB regions. The local complexity of neuronal cell-type composition in the neighborhood of any given cell is defined as the number of different neuronal cell types (at the subclass level) present in the 50 nearest-neighbor neurons surrounding that cell. This is not inconsistent with our results shown in Figure 6b, as sub-regions in HY, MB and HB have relatively smaller volumes than those in other major brain regions.

3.4. What proportion of transcriptional heterogeneity is explained by the spatial distribution of cell types in the brain? Are there any gene families that explain spatial distribution more than others (example: transcription factors)?

These are two interesting and different questions. For the first question, the referee seems to ask how predictive spatial location is of cell type identity. The second question is about prediction of spatial distribution by transcriptomic profiles and gene families. To address these questions, we first computationally imputed scRNA-seq whole transcriptome data into the MERFISH space (new Extended Data Figure 8) (text lines 269-281) and then used the imputed MERFISH expression patterns of all 8,460 DEGs to compute the mutual predictability between transcriptomic identity and spatial localization by training k-nearest neighbor (KNN) classifiers. The results show high degrees of predictability in both directions (new Extended Data Figures 14, 15) (text lines 763-777). We also tried to predict spatial specific by different gene families but did not observe obvious differences.

3.5. Expanding on the point above: it appears that the neuronal classes shown in Figure 2c could be derived by the intersection of spatial coordinates (brain region) and neurotransmitter identity (largely glutamatergic and GABAergic). Could neuronal diversity in the mouse brain be described by such a simple rule? If so, are there any specific gene families whose expression explains these major axes of neuronal diversity?

This observation is largely correct, especially at the class level. Every class has a dominant broad brain region and a dominant neurotransmitter type. Transcription factors are the most informative genes for separating classes as shown in Figure 5 and Extended Data Figure 3. For most classes, one can find specific small combos of TFs that have been known in developmental studies to be master regulators of these classes. As described in Results section for transcription factors, we have also identified TFs for glutamatergic or GABA neuronal types in different brain regions. For example, in forebrain: Dlx family for GABA, Neurod/Tbr1/Satb2 for Glut; in MB: Gbx2/Shox2/Tcf7l2 for Glut, Gata3/Tal1 for GABA (also extended to pons); in MY: Hox family for both GABA and Glut, and Pax2/8 for GABA.

SPECIFIC COMMENTS

ABSTRACT

Line 37: The number of single cells profiled is ~7 million, but the cells that survived QC are ~4.1 million. Please include this number in the abstract. Would be useful to indicate in the main text what fraction of the total number of cells and neurons in the mouse brain this represents.

We have added these to the Abstract and main text (text lines 138-139).

Line 44: dichotomy between the dorsal and ventral parts of the brain. The terms dorsal and ventral are vaguely defined - do dorsal and ventral here correspond to alar and basal subdivisions?

We have tried to correspond dorsal vs ventral regional comparison analysis with the alar and basal subdivisions. However, we should note that to our knowledge, there is no agreement in the field yet on where the line dividing alar and basal plates is. Moreover, it is not entirely clear which brain regions are derived from the alar or basal plate, as the field's knowledge about brain region development is still very much incomplete. For our analysis, we have adopted Puelles and Rubenstein's prosomeric model ¹ (also see Allen Developing Mouse Brain Atlas annotated by Dr. Puelles) as we believe it is the most updated model of alar/basal subdivisions. But this model still presents ambiguities as to the assignment of each adult brain region (in Allen CCF reference atlas) to the alar or basal part. Since referee 2 also raised the question about alar/basal comparison with some more specific points, please see our more detailed response to referee 2's point 3B below (page 27).

Line 46: how is "closely related" measured here? Number of DE genes? Proximity in the UMAP space? See also comments above

Please see our response above for 'Smaller points' in the general point 1 (page 6). We computed DE genes between all pairs of clusters, but this metric heavily depends on the

number of cells and penalizes rare clusters, so correlation matrix (new Extended Data Figure 3) seems to be a more reliable and fairer metric overall.

INTRODUCTION

General comment: the Introduction as a whole does not outline well the biological questions addressed in the results section of the paper. Lines 84-100, in particular, do not seem useful. The second part of the Introduction outlines the importance of a complete resource, but besides the future applications of the atlas, it would be extremely interesting to outline here the biological questions that can be addressed with the atlas directly - and that are addressed here.

We thank the referee for the insightful comments. We have removed the paragraph about the functional organization of the brain in the original text lines 84-100. We have added a sentence about questions that can be addressed in the current cell atlasing study (text lines 126-129).

Line 60: brain is "the most complex system in life" -> aren't animal societies or ecosystems arguably complex too?

We have removed this statement.

Line 72: This is a minor point, but still worth mentioning. I would agree with the statement that the mesoscale architecture of the mammalian brain is well understood. However, there is still some debate on what is the most meaningful description of this architecture, particularly between proponents of developmental bauplan-based models, and proponents of circuit-based models ("columnar models" in some literature). Citations 7 and 8 refer to the latter; however, the results of this paper are more in line with developmental models! (For example: transcription factor analysis) Cfr the work of Puelles, Nieuwenhuys and colleagues

This is an interesting point. For opening introduction though, we prefer to keep it simple as is because we study cell types (and development) for the purpose of understanding brain function. We refer to development-based brain organization in a new paragraph in Discussion (text lines 955-967).

Lines 74-82: These lines summarize the CCFv3 ontology of the mouse brain established by the Allen Institute. This ontology is based on adult neuroanatomy and for some parts of the brain does not match our understanding of the developmental and molecular diversity of neuron types - take the amygdala as an example. It would be useful to remind the reader that the anatomical ontology doesn't necessarily reflect the developmental history, nor the molecular and cellular diversity of these brain areas.

Again, we defer to Discussion to address this point (text lines 955-967). We agree with the referee and further believe that the anatomical ontology of the adult brain can be and should be refined by the new knowledge we are gaining about cell types and their development.

Line 102: cell types as functional units. Wouldn't it be more accurate and complete to say that cell types are "structural and functional" units? There are many examples where cell

types alone are not sufficient for function - in the brain, neural circuits are arguably the functional units.

We have added “structural” to the sentence (text line 91).

Lines 102-115: would be worth mentioning here that the cell type classification obtained by transcriptomics is congruent with morphological and functional classifications - as demonstrated by outstanding work by the Allen institute.

We have added a sentence in Introduction for this good point (text lines 99-100).

RESULTS

FIGURE 1

- I think it is important to specify here in the text or in the figure legend that the taxonomy (Figure 1a) is built using transcription factors

- the multiple color codes in Figure 1a are very hard to follow and disentangle. Any better way to represent the same information?

- what do the shades of gray in the dendrogram (Figure 1a) indicate? It is not explained in the legend.

We have replaced the original hierarchical dendrogram with a simpler class and subclass-based taxonomy that is ordered according to anatomical structures (Figure 1a). Color schemes for classes, subclasses, regions and neurotransmitter types are consistent with other figure panels (Figure 1b-e). We have clarified in the first Results section that classes and subclasses were initially computationally defined using transcription factors (text lines 188-197).

FIGURE 2

- Color schemes and spatial distribution of clusters are very hard to track and follow! The numbers help, but do not solve all problems. Potential solutions: (i) a web interface where clusters and subclasses can be plotted one by one in the UMAP and MERFISH spaces; (ii) close-up on a specific anatomical region (amygdala? Hypothalamus?) to exemplify the insights gained from these new datasets.

A web platform is available now (see above point 2 on page 7) to enable users to explore clusters and subclasses one-by-one in UMAP and MERFISH spaces. Given that the current manuscript is already well beyond the normal length of a Nature article, especially after adding all the new analyses requested by the referees, we don't think we will be able to accommodate additional close-up descriptions on a specific anatomical region in this manuscript that will do justice to that region. To exemplify the insights on specific cell types (down to cluster level) gained from the tremendous WMB datasets, we chose to provide in-depth analyses on all the non-neuronal cell types across the brain (Figure 4) as well as all the neurotransmitter types and their diverse co-transmitting patterns (Figure 3). We are

working on additional papers focused on more detailed cell type characterization of specific brain regions (e.g., amygdala, hypothalamus, etc.) in the context of prior knowledge about the cell types, anatomy, and function of these regions to reveal new insights gained, knowing that each of these regions is still highly complex.

Lines 284-286: it is not clear from the data reported here what would support the statement of "homology" of subclasses of glutamatergic neurons across the pallium (a supplementary figure missing?). Of note, the term homology refers to evolutionary relationships (descend from common ancestor). Without an evolutionary analysis, this term is not appropriate to describe similarities of cell types across anatomical areas.

We have changed the phrase "homologous relationships" to "parallel relationships" (text line 330).

Lines 292-299: do the classes reflect the distinction between projection neurons and interneurons? Why using a nomenclature that is not internally consistent? (some classes named after their ganglionic eminence of origin, other after the anatomical destination of neurons...)

The classes were defined based on intrinsic transcriptomic distance relationships. They were not designed to distinguish projection neurons and interneurons, although they may coincide with these connectivity properties. We thank the referee for pointing out the inconsistencies in our original naming of the classes. In the revised cell type taxonomy, we have changed CGE GABA and MGE GABA class names to CTX-CGE GABA and CTX-MGE GABA, and also split the CNU GABA class to two classes, CNU-MGE GABA and CNU-LGE GABA, to reflect both developmental origins and anatomical destinations of these neurons (Supplementary Table 7).

Lines 476-477: do all telencephalic astrocytes express *emx2*, a pallial marker? Or is *emx2* expressed only in a subset of the Astro-TE cells?

All clusters within the Astro-TE NN subclass show at least low-level *Emx2* expression.

Line 489: what is the exact percentage of clusters expressing at least one neuropeptide gene?

We have included 62 neuropeptides in our assessment of neuropeptide expression and used a cutoff of average $\log_2\text{CPM} \geq 3$ (i.e., ≥ 8 counts per million transcripts) to determine if a neuropeptide is expressed in a cluster or not. Based on these criteria, 79% of clusters express at least one neuropeptide and 21% of clusters do not. We have changed "More than 80%" to "About 79%" (text line 509).

FIGURE 4

Lines 530-532: a subset of CR cells is born in the prethalamic eminence, the site of origin of mitral and tufted cells (M/T) of the posterior accessory olfactory bulb (see for example Huilgol et al Nature Neuroscience 2013 PMID: 23292680). M/T cells of the main and accessory olfactory bulb are transcriptomically distinct (Zeppilli et al eLife 2021 PMID:

34292150). The manuscript does not distinguish between these two parts of the olfactory bulb (only MOB is mentioned). It is possible that the relatedness of CR cells is limited to M/T cells of the AOB, and in that case, shared developmental origin would be a potential explanation.

There are two clusters in the MOB Eomes Ms4a15 Glut subclass that are located in AOB. These two clusters are not the ones most transcriptomically related to the CR cells; some other MOB clusters are more related to the CR cells. There are also two clusters in the HPF CR Glut subclass that are located within MOB. Therefore, we think that CR cells are somehow related to both AOB and MOB cells. The referee's comment reminded us that our naming of the class and subclass here is not representative of both MOB and AOB. Therefore, in the revised taxonomy we have now renamed the MOB-CR Glut class to OB-CR Glut, and the MOB Eomes Ms4a15 Glut subclass to OB Eomes Ms4a15 Glut, where OB refers to 'Olfactory Bulb' (Supplementary Table 7).

Lines 535ss: I wonder whether a potential alternative explanation for the lack of a trajectory between DG Imn and DG differentiated neurons is the fact that DG and piriform immature neurons are lumped together here (unclear on what basis).

In the DG-PIR Ex IMN subclass, PIR IMN clusters and DG IMN clusters are segregated into distinct supertypes. The PIR clusters do not affect the relationship between the immature DG clusters and the more mature DG clusters. To illustrate this point, below we show a UMAP of cell types (using nomenclature in our revised taxonomy) in the DG-IMN Glut (including the PIR-specific supertype 142) and compare that to a UMAP of class DG-IMN Glut without supertype 142. In the DE gene analysis, we also observe discrete binary differences of marker gene expression between late-stage IMN development and mature DG cell types.

Lines 543ss: RMS astrocytes were already introduced earlier, without explaining the existence of a "tunnel" for neuronal migration. This section of the paper would be easier to read after bringing these paragraphs closer together.

We have rearranged the text to describe RMS astrocytes together with the immature neurons in the OB-IMN class (text lines 644-663). We have also added a figure panel, Figure 4I, to show the colocalization of these OB-IMN and astrocyte clusters in RMS in MERFISH sections.

Section: "Transcription factor modules across the whole mouse brain"

General comment:

Many of the observations listed here (for example: the existence of a dichotomy between forebrain and midbrain-hindbrain GABAergic TFs, the fact that LIM homeodomain TFs have complementary expression patterns in the brain, line 628) are not entirely new; classical in situ hybridization studies reached similar conclusions. The relevant references should be cited.

We have added the following references in the Results section on transcription factors:

Text lines 731-740:

Lhx2 and Lhx9 are co-expressed in TH and MB glutamatergic types, but Lhx2 is also specifically expressed in the pallium IT-ET types ²⁻⁵.

Lhx6 and Lhx8 orchestrate development of GABAergic neurons in CNU/HY ^{6,7}.

Lmx1a and Lmx1b are co-expressed in HB glutamatergic and MB dopaminergic cell types ^{8,9}. Lmx1b is also specifically expressed in MB/HB serotonergic types ^{10,11}.

Isl1 is widely expressed in HY/CNU, and much more highly in GABAergic than glutamatergic types ¹².

Text lines 699-700:

Gbx2, Shox2 and Tcf7l2 are highly expressed in thalamus glutamatergic neurons ¹³⁻¹⁶, while Shox2 and Tcf7l2 are also expressed in MB.

Specific comments:

Lines 573-574: because the TF modules were computed at the subclass level, some of the results are not consistent with the biology. For example, Tbr1 and Satb2 come up as a pallium glutamatergic-specific module, although these two TFs are well known for being expressed selectively in distinct types of cortical neurons. This limitation of the TF analysis should be made explicit in the main text.

Even though we included Tbr1 and Satb2 into a pallium glutamatergic (pallium glut) module, it doesn't mean that we did not detect their differential expression patterns. As shown in Figure 5g, Tbr1 and Satb2 indeed show differential expression in different subclasses of 'pallium glut'. While Tbr1 is developmentally associated with early-born deep-layer neurons and Satb2 with later-born upper-layer neurons, we can observe their expression in all cortical glutamatergic neurons, although at different expression levels in different

subclasses. We have now stated that they have differential expression in different subclasses (text lines 689-690).

Lines 582-583: Gbx2 labels a subset of thalamic glutamatergic neurons; midbrain neurons express other TFs (example: pax3 and pax7) besides Shox2 and Tcf7l2

As shown in Figure 5g, Gbx2 indeed labels a subset of thalamic glutamatergic neurons. Pretectum-derived MB neurons express Pax3 and Pax7. In the adult stage, we find sparse expression for Pax3 and very low expression for Pax7 in MB glutamatergic neurons labeled by Tcf7l2 and Shox2, but Pax3 and Pax7 are expressed in some pretectal GABAergic neurons. The data on Pax3 and Pax7 can be found in Figure 5g (Pax7 in MB GABA gene module, and Pax3 in astrocyte gene module).

FIGURE 5

- the axis labels in 5c and 5d are hard/impossible to read

The original Figure 5c and 5d panels have been removed and replaced by panels better able to show the role of TFs in distinguishing cell types.

FIGURE 6

- Fig 6d: if I understand this correctly, spatial span is not normalized by area size. Then, the fact that pallium neurons are more broadly distributed is not a surprise. Any way to take brain region sizes into account?

We want to preserve the old Figure 6d (now Figure 6g and updated) as a faithful representation of the 3D span of each cluster in the MERFISH space. Because brain regions are usually defined by observable specific populations of cells (i.e., cytoarchitecture), we are not sure what we would learn by normalizing the spatial span of cell types by region volumes. On the other hand, we did generate a new plot, Figure 6c, to normalize number of clusters per major brain region by the region volume (text lines 779-790). This plot further supports the observation that HY, MB and HB regions contain more clusters than the other major brain regions.

DISCUSSION

Line 748ss: the correspondence between transcriptomic identity and spatial specificity is not strictly quantified (see general comments above).

As mentioned in above points 3.2-3.4, we have now provided a quantitative graph in the new Figure 6a using CCF-registered MERFISH data, as well as quantitative scores such as the Gini coefficient (new Extended Data Figure 13) and Shannon diversity index, to show the highly specific distribution patterns of cell types across anatomical regions and their correspondence with transcriptomic class and subclass identities at the global level (text lines 747-761). We have also quantified both how predictive spatial location is of transcriptomic identity and how predictive transcriptomic identity is of spatial location, by building KNN classifiers in both directions and at different cell type and anatomical resolutions (new Extended Data Figures 14 and 15) (text lines 763-777).

METHODS

- how well do the microdissected regions match with the post-hoc MERFISH-based spatial allocation of cells?

Our microdissections for scRNA-seq correspond well with MERFISH spatial assignments. We have added a new figure panel, Extended Data Figure 2h, to show the regional composition of subclasses by scRNA-Seq and MERFISH side-by-side (text lines 1181-1183).

- how many cells were loaded on the 10x Chromium Controller?

We have added the following to the Methods section (text lines 1260-1278):

10x v2 11,870 ± 4,146 (mean cells loaded ± sd)

10x v3 13,404 ± 2,798

10x Multiome 16,007 ± 692

Lines 1483-1485: identification of classes and subclasses. The taxonomy was built using TF expression, because the results "recapitulate existing knowledge" better. It would be useful to see a comparison between these two clustering approaches (499 TFs vs ~8k marker genes)

Please see our response in general point 1 above.

- is the new classification (clusters and subclasses) consistent with Yao et al Cell 2021?

Yes, the classification of clusters and subclasses in the current WMB dataset is consistent with CTX-HPF cell type classification described in Yao et al ¹⁷. However, the resolution of the current dataset is higher than the previously described CTX-HPF dataset because we have added more cells from the same brain regions as well as other closely related regions. To show the consistency between the WMB classification and the Yao et al 2021 CTX-HPF classification on the same cells (the Yao et al 2021 dataset is a subset of the WMB dataset), below is the confusion matrix (x-axis: WMB taxonomy, y-axis: CTX-HPF taxonomy), and we have added the CTX-HPF cluster calls to the WMB cluster annotation table (Supplementary Table 7).

Lines 1489ss: how many classes and subclasses underwent manual correction?

Please see our response in the general point 1 above (pages 4).

Referee #2 (Remarks to the Author):

This landmark study represents the first effort to integrate both scRNA-Seq and spatial transcriptomics to generate a full catalog of cell types in the adult mouse brain. This work is a true tour de force, and the culmination of steadily improved and refined multidecade effort from the Allen Brain Atlas. The data quality is, as we have come to expect, excellent, the methodology transparent, and the data itself freely accessible to the broader research community. The authors should be warmly congratulated on their effort, skill, and public spiritedness. While I have very few concerns about the data or its presentation, I have several general questions about the completeness of this resource, its broader relevance to schema for CNS cell taxonomy, its direct relevance to existing criteria used for brain cell classification, and its comparison to similar datasets from other CNS regions. I also have some suggestions for how the data can be made even more useful to the broader research community. These are listed below.

General questions:

1. Despite the very impressive analysis done here, this is likely not close to a complete inventory of brain cell types. On the one hand, it is highly likely that many rare cell types have been missed, particularly since scRNA-Seq rather than snRNA-Seq was used here and the still somewhat limited sensitivity of MERFISH analysis. Ongoing analysis of cell diversity in the *Drosophila* brain, for instance, is identifying multiple examples of molecularly, morphological, and functionally distinct neuronal subtypes that are found only once in each hemisphere (PMID:33438579 and unpublished). Likewise, analysis in the mouse retina suggests that fewer than ten cells may be present per retina for some subtypes of wide-field amacrine cells (PMID:17048228). This implies that potentially thousands (or even tens of thousands) of additional cell types may remain to be identified.

However, it is equally clear that some of the cell subtypes identified represent transient cell states. This is obviously the case for immature adult-generated neurons and differentiating oligodendrocytes but is also likely to account for some of the other identified subtypes. For instance, many of the astrocyte subclusters are distinguished by differential expression of *Gfap*, which is a classic marker of astrocyte activation, while at least five neuronal subtypes are distinguished by expression of classic IEGs.

We agree with the referee on both points.

This raises the critical question of when the task of generating a brain cell atlas is actually going to be *done*, and how complex the final taxonomy of brain cell types is likely to be. In other words, how long do we have to wait for the next iteration of this study, and how many more are there likely to be? The authors could help address this by adding the following analysis and discussion:

While we agree that the more cells profiled, the more comprehensive a catalog of cell types will be obtained, we are not sure if this would lead to a deeper understanding of cell types. In this process, it is likely that more cell states will also be captured, and it may be increasingly difficult to distinguish them as the differences among close clusters are diminishing. At the same time, our taxonomy with over 5,300 clusters already presents a formidable challenge to the field in understanding how these cell types contribute to brain function, and how they relate to other cellular properties such as morphology, connectivity or physiology. Furthermore, our study already demonstrated that we were able to capture many rare cell types, such as the hypothalamic *Gnrh1* neurons, neurons in the subfornical organ (SFO), the Edinger-Westphal nucleus (EW), the Barrington's nucleus (B), and the pineal cells (also see more examples in point E below), barring technical confounds due to scRNA-seq (see more detailed discussion in point C below). We would like to suggest that it is more imperative now to study the structure, function, as well as development of these cell types, without having to wait for the next iteration. These studies will also be very informative in telling us how much more cell type granularity is needed for the understanding of cell type function.

A. Downsample their scRNA-Seq dataset to determine the rate at which new taxonomic categories (Division, Class, Subclass, Supertype, Subtype) emerge as the number of cells increases. Are there any brain regions which have clearly reached saturation? Which are still relatively undersampled?

We have previously performed down-sampling analysis^{18,19}, and it is not always easy to interpret the results. Designing meaningful down-sampling tests given the overall scale/complexity, different FACS sampling strategies and different RNA-seq methods is difficult. The number of resulting clusters depends heavily on choice of the clustering algorithms and cut-off parameters, and even in that case the number of clusters does not necessarily reflect the true cell type complexity, which can still be partially attributed to technical artifacts (e.g., 10x cells/nuclei are always distributed in a continuum of detected gene counts and expression levels, with variable degrees of dropouts).

Given these complications, we decided to provide a plot of number of cells per cluster by major brain regions in both scRNA-seq and MERFISH datasets (new Figure 6e) (text lines 801-803). The MERFISH plot likely reflects the true relative abundance of each cell type in the brain, and it shows that HY, MB and HB regions have more smaller clusters compared to other brain regions. The scRNA-seq plot could reflect both the true relative abundance and the sampling saturation in our dataset. The two plots correspond with each other very well. The distribution also suggests that HB contains a large proportion of small clusters in the scRNA-seq dataset and is thus likely still under-sampled.

B. Extrapolate these results to estimate the number of cells that would need to be analyzed to reach saturation for each major anatomical subdivision.

At this stage, we are still not comfortable with extrapolation. We keep finding very rare cell type populations. For the rare cell types that we might miss, it is difficult to tell how rare they are, and how much more sampling is needed. In addition, it is not only about the number of cells sampled, but also about sampling strategy, so it is difficult to provide a reliable estimate. On the other hand, based on MERFISH analysis, we know that we have already identified cell types that are extremely rare (see examples above and in part E below). Therefore, we believe that we already have very comprehensive sampling of the entire brain. There are only a couple of subclasses that we are still not confident about in the MB and HB areas (see point C below), which we will try to supplement in future studies using different modalities.

C. Estimate (or at least discuss) the effect of biases that arise from the use of scRNA-Seq instead of snRNA-Seq for cell profiling. It is fairly well-established at this point that scRNA-Seq enriches at least to some extent for cell types that tolerate dissociation and high-pressure microfluidic handling well, while large, fragile neurons tend to drop out. Have the authors conducted scRNA-Seq and snRNA-Seq in parallel on identical samples, and what do they observe?

While scRNA-seq generates high-quality transcriptomic data that can define cell types more clearly, we are aware of the drawback of scRNA-seq in missing certain cell types that are particularly vulnerable to tissue dissociation. In fact, we did generate an additional snRNA-seq dataset as part of the 10x Multiome data collection for the whole mouse brain. We performed a preliminary independent clustering of the snRNA-seq data and compared it with our scRNA-seq clusters. We found the snRNA-seq clusters are largely captured in scRNA-seq clusters except for a small number of neuronal clusters from MB and HB. Since we are not able to fully integrate the two datasets without compromising the high quality of the scRNA-seq clusters with snRNA-seq transcriptomes (which generally have lower gene

counts), we have decided to only add the 33 snRNA-seq clusters from 5 distinct subclasses that are depleted in the scRNA-seq taxonomy (text lines 217-221). This addition does fill essential gaps in the scRNA-seq taxonomy to make our WMB cell type atlas more complete.

For example, we only had one cerebellar Purkinje cell cluster with 33 cells from scRNA-seq while the snRNA-seq taxonomy includes 4 Purkinje clusters with 346 nuclei. The scRNA-seq taxonomy only had one HB Calcb Chol cluster with 19 cells, while the snRNA-seq taxonomy includes 5 clusters in this subclass with 274 nuclei, and these 5 clusters are distributed in highly distinct anatomical locations (motor nuclei III, V, VI, VII, XII, respectively) based on MERFISH. We also missed the large spinal-projecting neurons in the red nucleus (RN) nearly completely in the scRNA-seq taxonomy, and this type is found in snRNA-seq taxonomy. We still have a couple of heterogenous scRNA-seq clusters in the MB/HB that lack distinct markers and are not well separated from other clusters, and they mapped to thousands of MERFISH cells (with relatively low confidence scores). Unfortunately, our snRNA-seq dataset does not offer better cell type resolution for these clusters either. It is possible that these clusters present damaged transcriptomes of large/fragile cell types, and cell type signatures between these types are quite subtle, unlike Purkinje and HB Calcb Chol types which can be identified despite severe undersampling in the scRNA-seq transcriptomes. To fully categorize these remaining MB/HB cell types, we need to improve both the quality (more transcripts per cell) and quantity (cell numbers) of sampling for these cells. This is an active direction of continued investigation as we test different technologies and sampling strategies, but it cannot be completed in time for this manuscript.

We have now described the inclusion of a small set of 10x Multiome snRNA-seq data (1,687 nuclei contributing 33 clusters to the taxonomy) in the first Results section as we present the revised taxonomy (text lines 217-221).

D. While the study (correctly) focuses using scRNA-Seq to identify cell cluster, and validating key cluster-specific markers using MERFISH, and focusing on taxonomic categories identified using both methods, this raises the question of what was detected using only one of these methods. What fraction of scRNA-seq based clusters did not validate with MERFISH, and was it possible to identify novel (likely rare) cell clusters using MERFISH that were not detected in the scRNA-Seq dataset?

Nearly all scRNA-seq clusters reported here have been successfully mapped to our MERFISH dataset. In fact, MERFISH validation was one of the QC steps in our workflow. Those clusters that could not be validated (i.e., with either no MERFISH cells mapped to them or, more often, with too many MERFISH cells mapped to them at low confidence scores and poor spatial specificity) were almost always found to be low-quality transcriptomes either with contamination from other cell types or having low number of transcripts compared to related cell types and thus failed in our QC process.

As mentioned above in our response to Referee 1's comment 3.1 (page 8), we tried to use MERFISH data alone for clustering but found two major issues with this approach. Therefore, our experience is that mapping segmented MERFISH cells to scRNA-seq based clusters provides more reliable results.

E. While the dataset generally doesn't miss distinct well-characterized cell types (and I spend quite a few hours looking up my favorite known and as yet unpublished cell types to make sure they were there), there are a few examples. Taking glial cells, it is clear that some cell subtypes may be missed. This study distinguishes only six distinct subtypes of astrocytes, while other studies have identified substantial regional differences in astrocyte gene expression profile (e.g. PMID:33827819). Likewise, while this analysis cleanly resolves some distinctions among hypothalamic tanycytes (e.g. alpha from beta1 and beta2), it does not distinguish alpha1 and alpha2 subtypes, and it is unclear whether it identifies radial glial cell types found in other circumventricular organs such as the SFO, OVLT, and area postrema. Can the authors identify some other examples of missing cell types, particularly for reasonably well-characterized neurons?

We thank the referee for carefully checking our taxonomy and pointing out these interesting examples. We are glad to say that all the cell types the referee mentioned here have been found in our taxonomy.

Astrocytes: We had many more than 6 astrocyte clusters in our initial taxonomy and don't think we missed any astrocyte subtypes compared to ref PMID:33827819 (Herrero-Navarro et al, Sci Adv 2021, which we already cited), and the number of cells in each of our clusters is also generally quite large (Supplementary Table 7). It is still likely that we used too stringent criteria for splitting clusters (the same criteria were used for defining clusters across all neuronal and non-neuronal cells for consistency). In our revised taxonomy, in an effort to better characterize astrocyte and ependymal cell diversity since these cells often have continuous spatial gradients of gene expression, we have now lowered the DE genes criteria (de.score.th = 80 instead of 150). This modification resulted in more clusters in the Astro-Epen class overall (from 41 to 60 clusters) and specifically for astrocytes (from 18 to 30 clusters). The updated astrocyte clusters contain many regionally specific or enriched ones, such as those enriched in isocortex/OLF, HIP/CTXsp, DG, STR, TH, and cerebellar nuclei. On top of these, we still observe further spatial gradients which tend to be very continuous.

Tanycytes: We can indeed identify the alpha1 and alpha2 tanycytes. Based on the expression of marker genes described before and our MERFISH data, we have identified clusters 5245/5246, 5247, 5249, and 5250 as alpha1, alpha2, beta1, and beta2 tanycytes lining the 3rd ventricle of the hypothalamus, respectively. Furthermore, within the same Tanycyte subclass, we have identified two populations of tanycyte-like ependymal cells, clusters 5243 and 5244, that are located in SFO and OVLT (named OV in Allen CCF), respectively. These two types are molecularly similar to the self-renewing tanycyte-like ependymal cell types described previously. Finally, within the Astroependymal subclass, we have identified cluster 5240 as an astrocyte-like ependymal type with self-renewing properties located in area postrema (AP). And we found the hypendymal cluster 5263 is located in subcommissural organ (SCO). Thus, we have identified ependymal/tanycyte related clusters specifically localized in each of the circumventricular organs (CVOs). All cluster numbers mentioned here are from our revised cell type taxonomy (Supplementary Table 7).

As mentioned above in point C, by comparison with our snRNA-seq data, we have identified a set of MB/HB neuronal types that were indeed missing in our scRNA-seq dataset. We

have now added the cell types uniquely found in our snRNA-seq dataset into the WMB taxonomy to make it more complete.

We have added all the specific astrocyte, tanycyte and CVO clusters mentioned here into the Results section about non-neuronal cell types, along with MERFISH images showing the spatial specificity of these clusters (revised Figure 4b-e) (text lines 568-603).

F. Since the MERFISH-based validation profiles coronal sections separated by 200 microns, can the authors comment on which ABA-defined anatomical structures might be missed entirely

After CCF registration, we find that out of 554 terminal regions (grey matter only, Supplementary Table 1), there are only 7 small subregions completely missed in the MERFISH dataset: FRP1, FRP2/3, FRP5, AOBgl, AOBgr, AOBmi, and ASO. We have added this info to the Methods section (text lines 1836-1838).

2. Approaches to cell taxonomy:

The author's observation that persistent adult expression of developmentally important transcription factors is an accurate predictor of cell identity is both appealing and unsurprising to developmental neuroscientists, since controlling cell-type specific gene expression is what transcription factors do, and in any case these results tally closely with findings from adult *C. elegans*. However, this perspective is completely different from how neurophysiologists are used to thinking of cell identity, which they typically see as determined by expression of neurotransmitters, neuropeptides, and calcium binding proteins (one only needs to look at which Cre lines are in Jackson Lab's live repository to see which perspective dominates here). Since this finding will be resisted by some, and ignored by many, it's worth expanding and discussing further.

A. The comparison of the predictive value of TFs vs. random markers shown in Fig. 5b and 5c isn't really the proper comparison here. A more useful comparison would be TFs vs. (neurotransmitters+neuropeptides+calcium binding proteins), which directly compares the predictive value of these "developmental" vs. "physiological" molecular markers.

We agree and have thus compiled a 541 DEG list that includes neurotransmitter genes, neuropeptides, GPCRs, ion channels, etc. and another 857 DEG list with adhesion molecules, as the referee suggested in A and B here. We have tested and compared the predictive power of these different gene categories with the 534 TF gene list at different levels of the cell type hierarchy (new Extended Data Figure 3, Figure 5a-f) (text lines 188-197, 666-685). The results show that TFs still have the greatest power in distinguishing different cell types and their relatedness compared to other gene families.

B. In contrast to *C. elegans*, large-scale scRNA-Seq analysis in *Drosophila* brain seems to suggest that cell adhesion molecules and GPCRs may be better markers of cell identity than TFs. Are there any other functional categories of genes that have better predictive power than TFs in this dataset?

See our response to the above point A.

C. The particular prominence of transcription factors that are expressed during neurogenesis, rather than in mature neurons, in determining identity strongly implies that analysis of earlier stages of development could be a royal road to a final taxonomy of CNS cell types. If cell types are specified at (or shortly after) exit from mitosis, as developmental data suggests, it may be possible to rigorously distinguish fixed cell types from dynamic cell states -- between which cells can shift depending on internal state, extrinsic signals, or neuroplastic changes -- by analyzing the developing brain. The lead author has previously discussed these issues at length in multiple review articles, but it would be worth revisiting this issue here.

Note that what we showed in this study is the adult mature stage expression of transcription factors, rather than expression during neurogenesis, that defines cell type identities. Nonetheless, indeed, the continued expression of the same sets of TFs from development to adult defines and maintains cell type identities through time. This will allow us to study how cell types transition through different states, migrate to the final destinations, and form functionally specific circuits that are shaped by both intrinsic and extrinsic factors. We have added two new paragraphs in Discussion to address this point in the context of other points raised by this referee (see D below) and referee 1 above (text lines 955-982).

D. The hierarchical taxonomy used here is almost explicitly Linnean, with Division corresponding to Linnean Class, Class to Linnean Order, Subclass to Family, Supertype to Genus, and Cluster/Type to Species. This is intuitive and appealing, and makes use of rigorous molecular criteria in much the same way traditional taxonomy made use of comparative anatomy. This has the potential to be a useful general framework for classifying CNS cell identity, and I would like to see it discussed further and made more explicit.

This is a great point. We have added a new paragraph in Discussion to address this point in conjunction with the discussion (in response to referee 1 above) about the multidimensional relationships between cell types (text lines 969-982).

3. Relevance to existing neuroanatomic models:

There are a great many interesting details buried in the study which have important implications for our existing models of brain organization. While there's obviously far too much to cover in this article, it would be good if some of these points could be at least touched on.

We absolutely agree that our data and atlas contain so many interesting details, many of which we will not be able to cover in this manuscript. Therefore, we have chosen to address some of the below points raised by this referee that are most relevant to the main theme of this manuscript.

A. Which histoarchitecturally distinct brain regions and cytoarchitecturally defined cell groups (as defined in the ABA Reference Atlas) lack clear region-specific markers? Considering DA neurons, for instance, it seems there is high diversity among midbrain (A8-10), arcuate (A12), and olfactory (A16) clusters, but where are the others (A11/13/14/15)?

The referee asked an interesting question here. We hope to address this question comprehensively in future studies. Even though it is likely that for certain regions there are

no uniquely region-specific markers, one can almost always find a combinatorial set of marker genes that collectively specify that region. To do this, a computational approach will need to be developed.

Here we focus on addressing the issue of identifying transcriptomic clusters corresponding to all of the histoarchitecturally defined dopamine (DA) neuron groups. Indeed, as the referee pointed out, in our current manuscript, we only identified some DA groups but not others. We have realized that this is due to the stringent yet incomplete criteria we used to define neurotransmitter (NT) types, not considering alternative NT release mechanisms. The specific criteria used in our original manuscript are the following:

Glutamate: Slc17a6 (aka VGLUT2) or Slc17a7 (VGLUT1) or Slc17a8 (VGLUT3)

GABA: Slc32a1 (VGAT)

Glycine: Slc6a5

ACh (Chol): Slc18a3 (VACHT) (cross-checked with Chat)

Dopamine: Slc6a3 (DAT)

Serotonin: Slc6a4 (SERT)

NA/NE (Nora): Slc6a2 (NET) (cross-checked with Dbh)

Histamine: Hdc

In our revised taxonomy, we have now revised the criteria for identifying NT release and co-release cell types to the following (text lines 1656-1689):

Glutamatergic: Slc17a6 or Slc17a7 or Slc17a8

GABAergic: [Slc32a1 or Slc18a2] and [Gad1 or Gad2 or Aldh1a1]

Glycinergic: Slc6a5

Cholinergic: Slc18a3 and Chat

Dopaminergic: [Slc6a3 or Slc18a2] and [Th and Ddc]

Serotonergic: [Slc6a4 or Slc18a2] and [Tph2 and Ddc]

Noradrenergic: [Slc6a2 or Slc18a2] and Dbh

Histaminergic: Slc18a2 and Hdc

The new criteria have two major improvements: 1) they are more stringent as they require co-expression of both a NT transporter and the corresponding key NT synthesizing enzyme(s); 2) they are more inclusive as alternative NT synthesizing and releasing genes are added. In particular, we have added the vesicular monoamine transporter Slc18a2 (VMAT2) to all monoamine NTs as well as GABA. It is known that in many midbrain DA neurons (in VTA and SNc), Aldh1a1 is used for synthesizing GABA in the absence of Gad1 or Gad2, and Slc18a2 is used for co-release of DA and GABA in the absence of Slc32a1²⁰. Please see the Methods section for more details. It is also important to note that the new criteria may still miss other alternative NT release mechanisms, and they can continue to be refined in the future.

With the new criteria, we have now identified more DA neuron clusters in various parts of the brain and have identified DA clusters corresponding to all of the A8-A16 DA groups (text lines 460-468).

B. The observation that dorsally located cell types (pallium+thalamus+cerebellum) are less diverse but more transcriptionally distinct than ventrally located cell types is very interesting. Does this also hold for cells of subpallial origin (e.g. GABAergic pallial neurons)? Is it a general alar/basal (sensory vs. motor) distinction, and does it hold for dorsal midbrain (SC/IC), or dorsal HB?

As mentioned in our response to referee 1's specific comments on Abstract line 44 above (page 11), we have tried to correspond dorsal vs ventral regional comparison analysis with the alar vs basal, and sensory vs motor, subdivisions. However, we should note that to our knowledge, the dorsal/ventral, alar/basal, and sensory/motor divisions are not all consistent with each other. This makes it challenging for us to decide how to do the exact comparisons. Below are some example reasons why this is the case.

Regarding the dorsal/ventral division, we can consider a) pallium (cerebral cortex, CTX) as dorsal and subpallium (cerebral nuclei, CNU) as ventral for telencephalon, b) thalamus (TH) as dorsal and hypothalamus (HY) as ventral for diencephalon, and c) cerebellum (CB) as dorsal. We can divide up midbrain (MB) and hindbrain (HB, containing pons and medulla) each into dorsal and ventral parts. These divisions are consistent with the traditional view of alar/basal division (the columnar model developed by His, Herrick, et al., summarized in Swanson, Brain Architecture 2011) which is unfortunately considered outdated now ¹ (see the alar/basal part below).

A practical problem here is that it is unclear to us exactly where the dividing line between dorsal and ventral MB and HB should be (see sensory/motor and alar/basal divisions below). Also, an alternative division of telencephalon could be to divide both subpallium (as the referee suggested) and pallium into dorsal and ventral parts. For pallium perhaps we can separate ventral pallium from dorsal, medial and lateral pallium. For subpallium, what would be the rationale to sort the regions within it into dorsal and ventral groups – purely anatomical or considering developmental origins (e.g., various ganglionic eminences)?

Regarding the sensory/motor division, this would be the most applicable to MB and HB regions only. While we might consider TH as sensory and HY as motor, it doesn't make much sense and is practically impossible to partition pallium and subpallium into sensory and motor parts as both are mainly involved in integrative and cognitive functions. And CB is usually considered motor even though it is dorsal (and alar). For MB and HB, Allen CCF reference atlas (as well as other atlases) has designated sensory and motor regions specifically, which will be straightforward for us to adopt.

However, the sensory and motor regions do not fall neatly to the dorsal/ventral partition nor the alar/basal subdivision (see below). For example, superior colliculus is the most dorsal region of MB, but it is already divided into a sensory (SCs) and a motor (SCm) subregion. And other than SCs, inferior colliculus (IC) and a few other very small nuclei, the rest part of MB is all motor areas and behavioral state-related areas (which are likely derived from floor plate ¹). For HB (both pons and medulla), the sensory and motor/state regions are actually mostly segregated along the lateral (sensory) to medial (motor/state) axis, rather than dorsal-to-ventral.

Regarding the alar/basal division, the comparison of the older columnar model and the newer prosomeric model is shown in the diagram below (from ref ¹):

[REDACTED]

We will adopt the prosomeric model (ref ¹ and Allen Developing Mouse Brain Atlas) because it's based on newer and stronger experimental evidence related to brain development. However, in this model, the alar/basal dividing line at the rostral end is running through the middle of hypothalamus (see the red line in the diagram below from ref ¹), making the anterior-dorsal HY (and all other regions above it) alar and the ventral-posterior HY basal, which is not consistent with the sensory/motor functional segregation.

[REDACTED]

Furthermore, the model does not have a clear delineation yet on the alar or basal origin of all the numerous MB/HB regions (as annotated in Allen CCF reference Atlas based on Swanson/Dong ontology). Inconsistency with the sensory/motor division includes the following examples. Major dorsal structures in MB, including SCm, pretectal area, and periaqueductal gray (PAG) are alar even though they are motor regions (but part of PAG is basal plate derived). Two regions at the bottom of HB, pontine gray (PG) and inferior olivary complex (IO), both also motor regions, could be considered alar because these cells were generated in the rhombic lip (like CB) and then migrated to the ventral-most locations during development.

Considering all these complications, we have generated plots showing cluster numbers by alar vs basal division (using the Puelles definition) for all major brain regions (A-B below), and cluster numbers by sensory vs motor/state division for MB and HB only (C-D below). Comparing them with the plots shown in Figure 6b,c suggests that the cluster number difference between dorsal (Pall/CNU/TH/CB, low) vs ventral (HY/MB/HB, high) parts of the brain is not recapitulated by either alar vs basal or sensory vs motor/state divisions. There is no consistent difference between the alar and basal parts of HY, MB and HB (A-B), nor between the sensory and motor/state parts of MB and HB (C-D). In the Discussion section we have proposed an evolution-based hypothesis to explain the dichotomy, with the ventral part of the brain being more ancient, carrying out survival function of the organism and thus more subject to evolutionary constraints, versus the dorsal part of the brain being more recently emerged and mainly carrying out adaptive function of the organism (e.g., cognition) and thus having expanded and diverged more rapidly (text lines 900-908). This hypothesis does not demand any cell type difference between sensory and motor components of the brain. And the alar vs basal division appears to be a developmental phenomenon with unclear functional significance to us at this time. Therefore, for our manuscript we have decided not to relate our finding on the dorsal-ventral difference to sensory-motor or alar-basal divisions.

C. Both the distribution of individual neuronal cell types, and the transcriptional relatedness of these subtypes, generally fits very well with current understanding of the organization of both the developing and adult brain. However, there are several major exceptions, which the authors mention but do not discuss further. First, they observe a close transcriptional relationship of the thalamic reticular nucleus (TRN) and hypothalamic GABAergic cells. Standard prosomeric models of forebrain organization (which are used as the framework for the ABA's own Developmental Brain Atlas) state that the TRN and the hypothalamus proper derive from prosomere 3 and secondary prosencephalon, respectively. What is the significance of this and does this call this aspect of the model into question? Second, the observation that certain closely related glutamatergic cell types are found in thalamus and midbrain is also surprising, given that cell mixing between these compartments has not been previously reported during neurogenesis. Might these correspond to neuronal subtypes that migrate postmitotically (like thalamic GABAergic neurons) or cells derived from the isthmus, which forms the border zone between these regions?

First, we would like to point out that the exceptions to core organizational principles described here are real biological phenomena. We can confidently rule out the possibility

that these exceptions are driven by tissue-dissection errors because our cell type annotation and region assignment are based on MERFISH data. Therefore, these exceptions most likely reflect cell migration during development.

The reticular nucleus (RT) and hypothalamic zona inserta (ZI) cells both originate from the zona limitans intrathalamica (ZLI)²¹⁻²⁴. During early development, this structure is wedged in between prosomeres 2 and 3, produces Shh and Wnt signals, and has been designated as a secondary structural organizer of the diencephalon²⁵⁻²⁷. At the midbrain-hindbrain boundary, there is the isthmus organizer, which secretes Fgf8 and orchestrates patterning of mid- and hindbrain structures^{28,29}. The ZLI and isthmus represent two major anteroposterior organizers and between the two, a mitogen gradient is created that guides newly formed cells to proliferate, migrate, and differentiate^{30,31}.

The thalamic subclass 168 SPA-SPFm-SPFp-POL-PIL-PoT Sp9 Glut (found in various thalamic nuclei as shown in the subclass name) is transcriptionally more similar to midbrain glutamatergic subclasses, such as 167 PRC-PAG Tcf7l2 Irx2 Glut, than other thalamic glutamatergic subclasses. Subclasses 168 and 167, as well as other pretectal subclasses, are spatially located in the same anteroposterior plane. The midbrain neurons in these subclasses originate from prosomere 1, the diencephalic pretectum, whereas the thalamic nuclei originate from prosomere 2¹. Even though the adult types are similar, they still express key regional transcription factors Otx2 and Gbx2 for P1 and P2 respectively. It is possible that some of these glutamatergic types migrate from pretectum to thalamic nuclei as has been described for discrete GABAergic populations that migrate from pretectum to thalamus.

We have added explanations to these cross-regional cell types we observed (text lines 372-379, 406-414).

4. Comparison to other CNS structures:

It's something of a missed opportunity to not compare this brain dataset to the extensive datasets generated from spinal cord and retina, thereby obtaining a truly integrated view of CNS cell diversity. This is particularly the case for the retina, where an essentially scRNA-Seq based catalog of cell subtypes has already been generated, and a great deal of high-quality data is publicly available (c.f. PMID:34651173). Can this data be integrated into the analysis?

Expanding cell type comparison from the whole brain to the whole CNS is a very interesting question, but we think adding retina and/or spinal cord datasets to our WMB cell type analysis is beyond the scope of the current paper. A major technical challenge in doing this is that the previously published sc/snRNA-seq datasets are of variable qualities, and to make a meaningful comparison with our dataset we would need to subject them to the same QC criteria we have used, which could lead to the failing of a large portion of cells/nuclei and reduced cell type complexity. Furthermore, there are no comparable single-cell resolution spatial transcriptomics datasets for retina and spinal cord yet for the analysis of molecular-spatial correspondence like what we have done extensively in our manuscript. We look forward to future opportunities of integrative cell type analysis across the entire CNS.

Practical questions:

Most researchers using this resource will be either physiologists who simply want to selectively monitor or manipulate specific cell types in their region or circuit of interest, using site-specific recombinases, viral minipromoters, etc or molecular neuroscientists wanting to analyze mutants, disease states, etc. As a result, it's essential that users can readily find the exact minimal number of genes needed to identify each taxonomic category described here. This is pretty non-obvious at this point, and this information needs to be provided in a separate file. The number of genes used to uniquely define each category should be listed, and this should be stated separately for both the brain as a whole and for each major anatomic subdivision. This will allow rigorous interpretation of the effects of both systemic (mutant analysis, BBB-permeable AAV delivery) and anatomically targeted (e.g. AAV-based) manipulations of cell function. This really has the potential to dramatically alter the design of these experiments, shifting effort towards analysis of cell types that can be defined by relatively small numbers of genes.

We did provide the most discriminating marker genes for each subclass, supertype and cluster in the cell type annotation table, Supplementary Table 7. Given the numerous clusters across the whole brain, it is challenging to identify a very small number of markers that uniquely define each type, especially as we require each marker to be expressed in at least 40% cells in the cluster and have high contrast with the background. For every cluster, we examine the DE genes between the given cluster with all the other clusters and choose the one that ranks the highest based on all the pairwise comparison. We then eliminate the pairs that are already discriminated by the selected marker, and search for the next best gene for the remaining pairs, and we continue this process until all the pairs have been resolved. There is no guarantee that the gene set chosen this way is minimal, but the top genes chosen this way tend to be very specific. We also generated similar marker gene lists restricted to TFs or genes present on the MERFISH gene panel, which can be used for validation. We have now updated the marker gene lists in Supplementary Table 7 corresponding to the revised WMB cell type taxonomy.

Other questions:

A. In Fig. S3a, it appears that most neurons seem to have 2-2.5x UMI/cell relative to CBX/MOB cells and astrocytes. This doesn't seem to reflect a difference in cell volume, at least according to Fig. S2d, so what is the significance of this?

We believe this is indeed due to difference in cell volumes, as CBX and MOB are dominated by small granule cells. The quantification shown in Extended Data Figure 2d is based on each coronal section in its entirety. We could see that cell volumes in the most anterior sections (corresponding to MOB) are somewhat smaller compared to other sections, whereas cell volumes in the most posterior sections (containing both CBX and medulla) appear to show a broader range which could include the smaller cerebellar granule cells.

Clarifications:

A. Please define hypendymal cells and spell out choroid plexus (CHOR).

Hypendymal cells are specialized ependymal-like cells of the subcommissural organ (SCO) that secrete SCO-spondin (encoded by *Sspo* gene) into the cerebrospinal fluid leading to the formation of Reissner's fiber during development^{32,33}. We have added this information to the text (text lines 596-598).

References

1. Puelles, L. Survey of Midbrain, Diencephalon, and Hypothalamus Neuroanatomic Terms Whose Prosomeric Definition Conflicts With Columnar Tradition. *Front. Neuroanat.* **13**, 20 (2019).
2. Peukert, D., Weber, S., Lumsden, A. & Scholpp, S. Lhx2 and Lhx9 determine neuronal differentiation and compartment in the caudal forebrain by regulating Wnt signaling. *PLoS Biol.* **9**, e1001218 (2011).
3. Matho, K. S. *et al.* Genetic dissection of the glutamatergic neuron system in cerebral cortex. *Nature* **598**, 182–187 (2021).
4. Nakagawa, Y., Johnson, J. E. & O'Leary, D. D. Graded and areal expression patterns of regulatory genes and cadherins in embryonic neocortex independent of thalamocortical input. *J. Neurosci.* **19**, 10877–10885 (1999).
5. Okamura-Oho, Y. *et al.* Broad integration of expression maps and co-expression networks compassing novel gene functions in the brain. *Sci. Rep.* **4**, 6969 (2014).
6. Flandin, P. *et al.* Lhx6 and Lhx8 coordinately induce neuronal expression of Shh that controls the generation of interneuron progenitors. *Neuron* **70**, 939–950 (2011).
7. Fragkouli, A., van Wijk, N. V., Lopes, R., Kessaris, N. & Pachnis, V. LIM homeodomain transcription factor-dependent specification of bipotential MGE progenitors into cholinergic and GABAergic striatal interneurons. *Development* **136**, 3841–3851 (2009).
8. Doucet-Beaupré, H. *et al.* Lmx1a and Lmx1b regulate mitochondrial functions and survival of adult midbrain dopaminergic neurons. *Proc. Natl. Acad. Sci. U. S. A.* **113**, E4387-96 (2016).
9. Yan, C. H., Levesque, M., Claxton, S., Johnson, R. L. & Ang, S.-L. Lmx1a and Lmx1b function cooperatively to regulate proliferation, specification, and differentiation of midbrain dopaminergic progenitors. *J. Neurosci.* **31**, 12413–12425 (2011).
10. Ding, Y.-Q. *et al.* Lmx1b is essential for the development of serotonergic neurons. *Nat. Neurosci.* **6**, 933–938 (2003).
11. Cheng, L. *et al.* Lmx1b, Pet-1, and Nkx2.2 coordinately specify serotonergic neurotransmitter phenotype. *J. Neurosci.* **23**, 9961–9967 (2003).
12. Ehrman, L. A. *et al.* The LIM homeobox gene *Is1* is required for the correct development of the striatonigral pathway in the mouse. *Proc. Natl. Acad. Sci. U. S. A.* **110**, E4026-35 (2013).
13. Lipiec, M. A. *et al.* TCF7L2 regulates postmitotic differentiation programmes and excitability patterns in the thalamus. *Development* **147**, (2020).
14. Chen, L., Guo, Q. & Li, J. Y. H. Transcription factor Gbx2 acts cell-nonautonomously to regulate the formation of lineage-restriction boundaries of the thalamus. *Development* **136**, 1317–1326 (2009).
15. Mallika, C., Guo, Q. & Li, J. Y. H. Gbx2 is essential for maintaining thalamic neuron identity and repressing habenular characters in the developing thalamus. *Dev. Biol.* **407**, 26–39 (2015).
16. Lee, M. *et al.* Tcf7l2 plays crucial roles in forebrain development through regulation of thalamic and habenular neuron identity and connectivity. *Dev. Biol.* **424**, 62–76 (2017).
17. Yao, Z. *et al.* A taxonomy of transcriptomic cell types across the isocortex and hippocampal formation. *Cell* **184**, 3222-3241.e26 (2021).
18. Tasic, B. *et al.* Shared and distinct transcriptomic cell types across neocortical areas. *Nature* **563**, 72–78 (2018).
19. Yao, Z. *et al.* A transcriptomic and epigenomic cell atlas of the mouse primary motor cortex. *Nature* **598**, 103–110 (2021).
20. Wallace, M. L. & Sabatini, B. L. Synaptic and circuit functions of multitransmitter neurons in the mammalian brain. *Neuron* (2023) doi:10.1016/j.neuron.2023.06.003.

21. Shimogori, T. *et al.* A genomic atlas of mouse hypothalamic development. *Nat. Neurosci.* **13**, 767–775 (2010).
22. Delaunay, D. *et al.* Genetic tracing of subpopulation neurons in the prethalamus of mice (*Mus musculus*). *J. Comp. Neurol.* **512**, 74–83 (2009).
23. Vue, T. Y. *et al.* Characterization of progenitor domains in the developing mouse thalamus. *J. Comp. Neurol.* **505**, 73–91 (2007).
24. Wong, S. Z. H. *et al.* In vivo clonal analysis reveals spatiotemporal regulation of thalamic nucleogenesis. *PLoS Biol.* **16**, e2005211 (2018).
25. Kiecker, C. & Lumsden, A. Compartments and their boundaries in vertebrate brain development. *Nat. Rev. Neurosci.* **6**, 553–564 (2005).
26. Chatterjee, M. & Li, J. Y. H. Patterning and compartment formation in the diencephalon. *Front. Neurosci.* **6**, 66 (2012).
27. Govek, K. W. *et al.* Developmental trajectories of thalamic progenitors revealed by single-cell transcriptome profiling and Shh perturbation. *Cell Rep.* **41**, 111768 (2022).
28. Nakamura, H., Katahira, T., Matsunaga, E. & Sato, T. Isthmus organizer for midbrain and hindbrain development. *Brain Res. Brain Res. Rev.* **49**, 120–126 (2005).
29. Hidalgo-Sánchez, M., Andreu-Cervera, A., Villa-Carballar, S. & Echevarria, D. An Update on the Molecular Mechanism of the Vertebrate Isthmic Organizer Development in the Context of the Neuromeric Model. *Front. Neuroanat.* **16**, 826976 (2022).
30. Wurst, W. & Bally-Cuif, L. Neural plate patterning: upstream and downstream of the isthmus organizer. *Nat. Rev. Neurosci.* **2**, 99–108 (2001).
31. Kiecker, C. & Lumsden, A. The role of organizers in patterning the nervous system. *Annu. Rev. Neurosci.* **35**, 347–367 (2012).
32. Guerra, M. M. *et al.* Understanding How the Subcommissural Organ and Other Periventricular Secretory Structures Contribute via the Cerebrospinal Fluid to Neurogenesis. *Front. Cell. Neurosci.* **9**, 480 (2015).
33. Zeisel, A. *et al.* Molecular Architecture of the Mouse Nervous System. *Cell* **174**, 999–1014 e22 (2018).

Reviewer Reports on the First Revision:

Referees' comments:

Referee #1 (Remarks to the Author):

The authors conducted an extensive revision of the original manuscript. I am glad to see that the data have been made accessible through a new easy-to-use web interface, the ABC Atlas. The authors replied extensively to this reviewer's questions, comments, and concerns. The use of transcription factors (TFs) to build the cell type taxonomy is motivated extremely well, with an additional figure (extended figure 3) that visually illustrates why TFs are so well suited for classifying neuron types. The authors also expanded on the analysis of the spatial distribution of cell types, revealing interesting biological insights. Overall, the manuscript is improved on all fronts, and I wholeheartedly recommend its publication in the present form.

Referee #2 (Remarks to the Author):

The authors have thoughtfully and thoroughly addressed to my satisfaction each point raised in review. With the addition of an user-friendly interactive web browser for viewing the data, a refined hierarchical taxonomy of brain cell types, and strengthened support of key claims such as the central role of transcription factor expression in defining cell identity, this is an extremely rigorous and monumental study which will both likely both serve as the standard neuroanatomical reference for the mouse brain for at least the next decade, and truly sets the standard for future studies. They are to be congratulated for the successful cumulation of a massive two-decade long effort.

I currently have only one very minor correction to the manuscript:

On line 377, the authors state that neurons of the thalamic reticular nucleus and zona incerta derive from the ZLI. While this may be true for a small subset of neurons in the these structures, the preponderance of evidence (e.g. PMID: 20436479, 21865661, 32420617, 35045288), suggests that both structures primarily derive from the prethalamus (p3 in current prosomere model). This statement should be amended accordingly.

Author Rebuttals to First Revision:

Response to referees

Referee #1 (Remarks to the Author):

The authors conducted an extensive revision of the original manuscript. I am glad to see that the data have been made accessible through a new easy-to-use web interface, the ABC Atlas. The authors replied extensively to this reviewer's questions, comments, and concerns. The use of transcription factors (TFs) to build the cell type taxonomy is motivated extremely well, with an additional figure (extended figure 3) that visually illustrates why TFs are so well suited for classifying neuron types. The authors also expanded on the analysis of the spatial distribution of cell types, revealing interesting biological insights. Overall, the manuscript is improved on all fronts, and I wholeheartedly recommend its publication in the present form.

We thank this referee's very positive comments and recommendation of publication in the present form.

Referee #2 (Remarks to the Author):

The authors have thoughtfully and thoroughly addressed to my satisfaction each point raised in review. With the addition of an user-friendly interactive web browser for viewing the data, a refined hierarchical taxonomy of brain cell types, and strengthened support of key claims such as the central role of transcription factor expression in defining cell identity, this is an extremely rigorous and monumental study which will both likely both serve as the standard neuroanatomical reference for the mouse brain for at least the next decade, and truly sets the standard for future studies. They are to be congratulated for the successful culmination of a massive two-decade long effort.

I currently have only one very minor correction to the manuscript:

On line 377, the authors state that neurons of the thalamic reticular nucleus and zona incerta derive from the ZLI. While this may be true for a small subset of neurons in the these structures, the preponderance of evidence (e.g. PMID: 20436479, 21865661, 32420617, 35045288), suggests that both structures primarily derive from the prethalamus (p3 in current prosomere model). This statement should be amended accordingly.

We thank this referee's very positive comments. Regarding the minor point, we think that there are previous publications supporting both prethalamus and ZLI as the developmental origin(s) of thalamic reticular nucleus and zona incerta neurons. Therefore, we have amended the relevant sentence in our manuscript to: "Both RT and ZI neurons may have originated from the prethalamus or the zona limitans intrathalamica (ZLI)³³⁻³⁷." And we cited relevant references, including some suggested by the referee (not all refs due to lack of space).